# Kaolinite induces rapid authigenic mineralisation in unburied shrimps
Nora Corthésy ✉, Farid Saleh ⬡ ✉, Jonathan B. Antcliffe ⬡ & Allison C. Daley ⬡

Fossils preserving soft tissues and lightly biomineralized structures are essential for the reconstruction of past ecosystems and their evolution. Understanding fossilization processes, including decay and mineralisation, is crucial for accurately interpreting ancient morphologies. Here we investigate the decay of marine and freshwater shrimps deposited on the surface of three different clay beds. In experimental set ups containing kaolinite, cryogenic scanning electron microscopy shows a black film comprised of newly formed anhedral and cryptocrystalline aluminosilicates on marine shrimp cuticles, which stabilise the overall morphology. This is the first experimental evidence for the replication of arthropod lightly biomineralized structures in aluminosilicates shortly after death, while carcasses are not buried by sediments. The preservation of morphology through aluminosilicates could result in carcasses persisting on the seafloor for weeks without losing much external anatomical information. In this context, instantaneous burial capturing animals alive may not be a prerequisite for exceptional preservation as usually thought.

Under typical environmental conditions, soft, non-mineralized organic tissues and lightly biomineralized structures are rapidly destroyed through cell autolysis, heterotrophy by micro- and macro-organisms, and denaturing of structural elements under the changing chemical conditions of decay. Yet, some delicate structures survive as fossils for hundreds of millions of years. For example, the exquisite fossils of the Burgess Shale (~508 million years), and Burgess Shale-type preservation more broadly, record the early diversification of animals during the Cambrian Explosion[1–5]. In sites with Burgess Shale-type preservation, diverse taxa such as annelids, arthropods, lobopodians, and molluscs are preserved with their soft tissues[6]. These fossils are often found as carbonaceous compressions[7,8] associated with aluminosilicate minerals[3,9]. Despite the importance of the Burgess Shale for understanding major evolutionary events, the processes behind its exceptional preservation remain shrouded in mystery.

Geological processes must have interrupted biological decay to transform labile anatomies into endurant rock. In other words, exceptional preservation requires structural stabilization of soft tissues and lightly biomineralized structures by permineralisation or templated mineral growth to outpace decay processes such as microbial metabolism and cell autolysis. The spatial association of labile anatomies with aluminosilicate sheets has been observed in fossils with Burgess Shale-type preservation[3,9–14], however, the timing of the replication in aluminosilicate sheets is controversial, with some suggesting that it occurs during early diagenesis[3,9,10,12], and others suggesting it occurs during maturation, a much later stage of

fossilization[11,13,14]. If the preservation of labile morphologies is truly governed by early diagenetic interactions between organic structures and clay minerals, then one would expect to see them associated in the earliest stages of decay. If the association does not develop until later, then other mechanisms must be responsible for the stabilization of organic structures. It is also unclear whether the aluminosilicates result from the simple attachment of pre-existing clays, such as kaolinite, in the matrix[9,15] or from the precipitation and formation of new authigenic minerals[9,16].

Experiments investigating the interaction between clay minerals and decaying carcasses help to provide meaningful controls on the relevant conditions for the replication of morphologies in aluminosilicate sheets. Decay experiments have proven highly valuable in cataloguing the loss and retention of anatomical information under various environmental conditions, some in the presence of sediments[16–21] and some without[22–27]. Excluding sediments is a sensible and coherent approach to understanding the baseline of the processes of decay by limiting the variables in a specific system. Recent years have seen extensive mapping of the patterns of decay sequences for a range of animals[28,29], and it is now possible to examine more process-orientated questions with more interactions and complexity introduced to the system.

In order to test the effects of clay minerals on decaying carcasses without burial, euthanised specimens of the marine shrimp *Palaemon varians* and the freshwater shrimp *Neocaridina davidi* were placed atop beds of three different clay compositions and under water to simulate conditions at the sediment-water interface. These experiments show that

Institute of Earth Sciences, University of Lausanne, Géopolis, Lausanne, CH-1015, Switzerland. ✉e-mail: nora.corthesy@unil.ch; farid.nassim.saleh@gmail.com

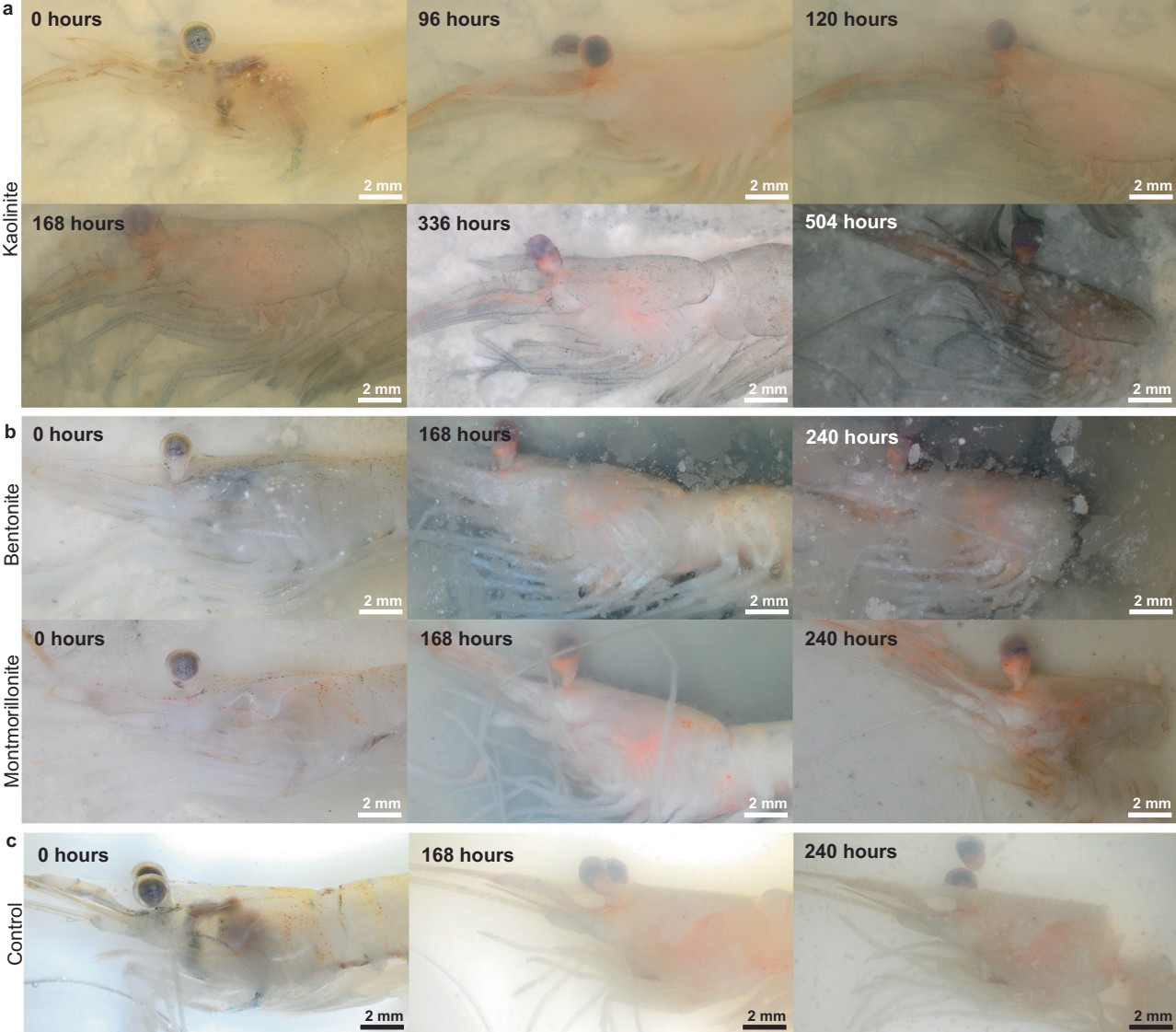

**Fig. 1 | Examples of decay stages of marine shrimps.** Deposited on kaolinite (**a**), on bentonite and montmorillonite (**b**), and without any sediment (control) (**c**). At 0 h, the shrimps are all intact and transparent (Taphonomic score 0). **a** In the presence of kaolinite, the shrimps remain opaque and intact until 96 h (Taphonomic score 1). At 120 h, a black film is observed on the cuticle of the shrimps on kaolinite (Taphonomic score 2), and they remain well preserved until 336 h when the carapace detaches from the cephalothorax (Taphonomic score 3). The shrimps on kaolinite are separated into two pieces at 504 h of decay (Taphonomic score 5). **b** For shrimps placed on bentonite and montmorillonite, the carapace detaches from the cephalothorax at 168 h (Taphonomic score 3), and the shrimps are separated into two

pieces at 240 h (Taphonomic score 5). **c** For marine controls, at 168 h, the carapace detaches from the cephalothorax, and the cephalothorax starts to separate from the abdomen (Taphonomic score 4). At 240 h, the shrimps are separated into two pieces in the control condition (Taphonomic score 5). Note that marine shrimps with bentonite and montmorillonite and without sediment are more decayed at 240 h than marine shrimps with kaolinite at 336 h of decay. Also, occasionally shrimps decaying on bentonite show black material around the decaying organism (**b**). This black material, unlike under kaolinite set-up, does not replicate the anatomy and does not limit the decay of the carcass.

the decay of marine shrimps deposited on the surface of kaolinite is slower than that of marine shrimps deposited on bentonite and montmorillonite, or without clay. Slower degradation is accompanied by the precipitation of newly formed aluminosilicates on marine shrimp cuticles, providing the first experimental evidence that this mode of authigenic mineralisation can occur on arthropod carcasses in the absence of burial. When the replication in aluminosilicates occurs in the presence of kaolinite clays, external morphology remains undistorted for several weeks even when exposed to the chemical gradients of the water column. This observation holds particular significance in the context of the Cambrian Explosion, as kaolinite is frequently observed in association with exceptional fossil preservation[30] in marine rock from the Cambrian Period.

## Results and discussion

### Kaolinite slows down marine shrimp decay

Immediately following euthanasia, marine shrimps are transparent and intact (Fig. 1, Supplementary Fig. 1a). At 24 h, under all conditions, marine shrimps become opaque and pink/white, and the eyes start to turn black (Taphonomic score 1; Supplementary Table 1, Supplementary Fig. 1b). Over the next three days (at 96 h of decay), minimal external anatomical change is observed in the presence of any of the three clays (Fig. 1a). However, starting at 120 h, a fine black film appears on the cuticle of marine shrimps on kaolinite (Taphonomic score 2; Figs. 1a and 2), outlining the exoskeleton in exquisite detail (Figs. 1a and 2). This film is absent from all our other experimental conditions (Fig. 1b, c), which are instead covered by a white filamentous layer (white film) as early as 24 h after the start of decay

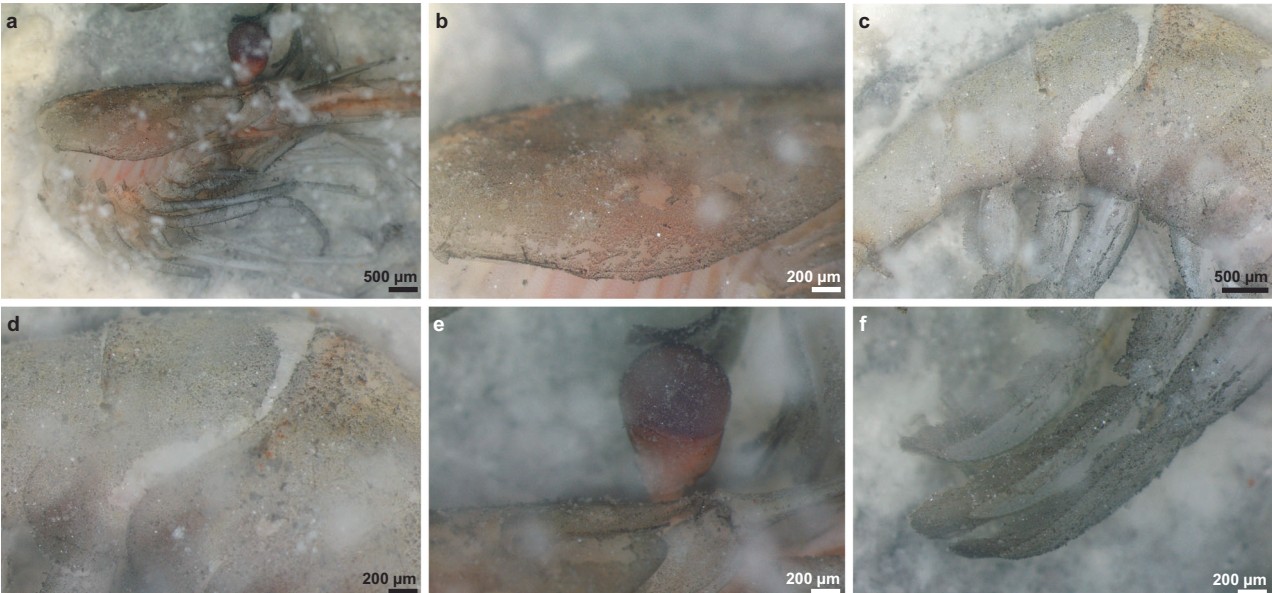

**Fig. 2 | Observation of the black film on a marine shrimp placed on a kaolinite bed after 504 h of decay. a** Cephalothorax, **b** carapace, **c**, **d** abdomen, **e** eyes, and **f** pleopods covered by the black film. Although the black film in the kaolinite experiments originally formed on the shrimps, after some time it also appeared in the surrounding matrix, likely following the diffusion of organic material from the carcass during degradation.

(Taphonomic score 2; Fig. 1b, c, Supplementary Fig. 1c, d; Supplementary Data). For marine controls, the carapace starts to detach from the cephalothorax at 96 h (Taphonomic score 3; Fig. 1c). The carapace of marine shrimps decaying on bentonite and montmorillonite begins to detach from the cephalothorax between 120 to 168 h into the experiment (Taphonomic score 3; Fig. 1b), whereas, during the same period for the controls, the carapace is completely detached from the cephalothorax, which starts to separate from the abdomen (Taphonomic scores 4–5; Fig. 1c). The carcasses of marine shrimps remain intact in the presence of kaolinite until 336 h of degradation (Fig. 1a). With bentonite and montmorillonite, the carapace is detached from the cephalothorax at 240 h of decay (Taphonomic score 4), and for some specimens, the cephalothorax separates from the abdomen (Taphonomic Score 5; Fig. 1b). In the presence of kaolinite, the black film persists on the exoskeleton of marine shrimps, and the cephalothorax only separates from the abdomen at 504 h (Fig. 1a). Overall, in marine settings, decay is less pronounced in the presence of kaolinite compared to other conditions (Figs. 1 and 2), as the morphology of the shrimps placed on kaolinite beds (Fig. 1a) appears to be preserved for a longer period than in the absence of sediments or in the presence of bentonite and montmorillonite (Fig. 1b, c). This result is consistent with previous studies that suggest that kaolinite slows down decay. Earlier work showed that kaolinite can limit bacterial growth and reduce bacterial diversity surrounding a carcass, thus, limiting its degradation and recycling[18,31]. It has also been shown that carcasses buried in kaolinite decay slower than when buried in other minerals[16,17,20]. The implication of our study is different from previous works because it involves unburied shrimp carcasses deposited on the surface of three mineralogical beds.

When comparing individual shrimps in each marine experimental condition, decay proceeded more uniformly for samples decaying on kaolinite (Fig. 3a) than for bentonite (Fig. 3b), montmorillonite (Fig. 3c), and the control (Fig. 3d). For example, all marine shrimps undergoing decay on the surface of kaolinite maintain the same stage of degradation for up to 120 h (Fig. 3a). This uniformity is not observed in the three other marine conditions, where individuals decay more rapidly and have more heterogeneous decay scores (Fig. 3b–d). After 120 h, decay rates also become more pronounced and variable in the absence of kaolinite (Fig. 3b–d) compared to when it is present in the experimental setup (Fig. 3a). Importantly, a plateau in taphonomic scores is evident in marine shrimps decaying on kaolinite starting at 120 h (Fig. 3e). This plateau is not observed in the other experimental conditions and marks the

point in the experiment when the decay progress between the different marine conditions strongly departs from each other (Fig. 3e). Starting at this timepoint, the taphonomic scores of marine shrimps decaying on kaolinite are significantly lower than those of marine shrimps decaying on bentonite and without clay (Contrast analysis, $p_{Time=120[Bentonite-Kaolinite]} = 0.021$, $z\text{-}ratio_{Time=120[B-K]} = 2.882$, $p_{Time=120[Control-Kaolinite]} = 0.005$, $z\text{-}ratio_{Time=120[C-K]} = 3.289$, Supplementary Table 2). Between 144 and 216 h of decay, the taphonomic scores of marine shrimps decaying on kaolinite are lower than those of marine shrimps decaying in the three other experimental conditions (bentonite, montmorillonite, without clay) (Contrast analysis, $p_{Time=144[Bentonite-Kaolinite]} = 0.005$, $z\text{-}ratio_{Time=144[B-K]} = 3.295$; $p_{Time=144[Kaolinite-Montmorillonite]} = 0.004$, $z\text{-}ratio_{Time=144[K-M]} = -3.380$, $p_{Time=144[Control-Kaolinite]} = 0.001$, $z\text{-}ratio_{Time=144[C-K]} = 3.741$, Supplementary Table 2). Taphonomic scores are not significantly different between marine shrimps decaying on bentonite, montmorillonite, and without clay during the entire time of the experiment (Supplementary Table 2). Despite the small sample size, the decay rate of marine shrimps placed on kaolinite is significantly different from the decay rate of the other specimens (Fig. 3). The start of the statistically significant decay plateau at 120 h for marine shrimps placed on kaolinite (Fig. 3e) coincides with the onset of the black film formation (Figs. 1a and 2).

The black film (Fig. 2) was formed uniquely on all marine shrimps decaying on kaolinite at 1.024 psu. The black film did not form in the single marine shrimp left to decay on kaolinite at a slightly lower salinity (1.019 psu, prepared for Cryo-SEM). Occasionally marine shrimps decaying on bentonite show a black material around the decaying organism (Fig. 1b) but this black material does not replicate the anatomy and does not limit the decay of the carcass unlike the black film forming on shrimps in the presence of kaolinite. Moreover, the black film was not observed in any of the freshwater shrimps, regardless of whether they were decaying in the presence of kaolinite, bentonite, montmorillonite, or without sediments (Supplementary Fig. 2). In general, freshwater shrimps exhibit a faster rate of decay compared to their saltwater counterparts (Supplementary Fig. 3). Initially, at 0 h, all freshwater shrimps appear intact and transparent (Supplementary Fig. 2). After 24 h of decay, the cuticle becomes opaque and pink (Supplementary Fig. 2). Between 48 and 96 h, the cuticle starts detaching from the thorax and abdomen, exposing the gills (Supplementary Fig. 2). Complete detachment of the cuticle occurs at 96–144 h (Supplementary Fig. 2). By 144–168 h, the thorax and abdomen are split, the body anatomy is destroyed, and internal organs are displaced (Supplementary Fig. 2).

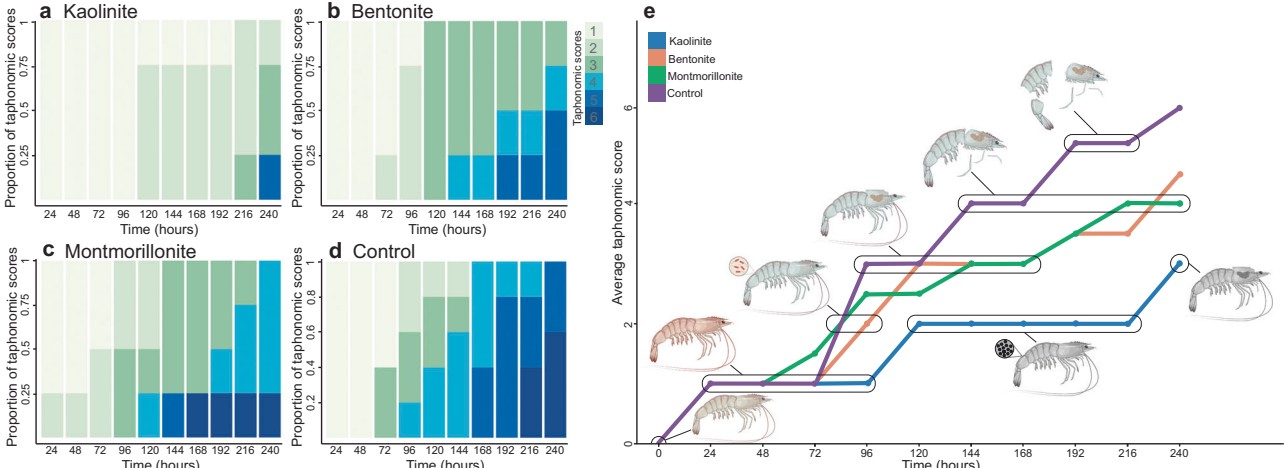

**Fig. 3 | Marine shrimp preservation in different experimental conditions.** Proportion of the taphonomic score according to time (**a**) in the presence of kaolinite (*n* = 4), **b** in the presence of bentonite (*n* = 4), **c** in the presence of montmorillonite (*n* = 4), and **d** in the absence of sediment (*n* = 5). Taphonomic scores, quantifying the decay state, are represented by different colours. Darker blue colours indicate more advanced decay. **e** Taphonomic scores of marine shrimps (*n* = 17) decaying under different environmental conditions over time. The lines follow the median of the taphonomic scores of each experimental condition at each timepoint of the experiment. Each taphonomic score is illustrated by representations of shrimp throughout the experiment as described above. For the shrimps placed on bentonite, montmorillonite and in the absence of clay, the decay pattern is the same, with the formation of a white filamentous layer that is likely to correspond to a biofilm represented with the white colour and the little pink bacteria in the circle. The black film only forms in the presence of kaolinite hence the representation of black shrimps in this specific experimental condition and the poorly mineralized aluminosilicate crystals are represented in the circle.

Generally, no significant differences are observed between freshwater shrimps decaying on different substrates (Supplementary Table 3), other than a single significant difference occurring at 48 h when the carapace detaches faster for the shrimps placed on montmorillonite than in the other conditions (Supplementary Fig. 3).

## Kaolinite promotes the mineralisation of the cuticle of marine shrimps

For specimens on which no black film formed, the shrimp analysed under Cryo-SEM, show an organic surface (Fig. 4a), covered by bacteria, mainly coccoids, ranging in size from 0.6 to 1.5 µm (Fig. 4b). These bacteria are likely to correspond to the white filamentous layer that was observed on all the samples except for the shrimps placed on kaolinite (Fig. 1b, c). For the two marine shrimps on which a black film formed (Fig. 4c), the cuticle is irregular (Fig. 4d), showing more relief than in the sample where a black film did not form (Fig. 4a). At high magnification, areas of higher relief often lack clearly defined structures (Fig. 4e) and resemble poorly crystallized aluminosilicates (Fig. 4e) characterized by a limited presence of their typical sheets (Fig. 4f–h).

Interpretations of higher textural relief in backscatter detector imagery (Fig. 4e–h) are corroborated by elemental analyses obtained with Energy-Dispersive Spectroscopy, highlighting cuticle mineralisation in aluminosilicates when the black film is present (Fig. 4i). The main differences between the two spectra of energy dispersive elements are the large aluminium and silicon peaks that were observed in the areas with the black film and that were not detected in the areas with no black film (Fig. 4i). For the other detected elements, the peaks are more or less similar between the two types of areas, although more carbon is detected in the areas with no black film probably because the organic carbon was replaced/covered by aluminosilicates in areas with a black film. Calcium can also be detected in the analysed shrimps because it is one of the constituent elements of the cuticle (Fig. 4i, n). Some of this calcium might have been mobilized away from the cuticle owing to low pH values that form around a decaying carcass[32]. Oxygen is present everywhere even in the microscope chamber. Platinum is present in all samples because these were coated with this element before analyses. Sodium and chlorine also are ubiquitous because they are the main constituents of salt in ASW.

Elemental mapping reveals more details about the black film and its associated mineralisation (Fig. 4l–t). In addition to the observed association between aluminium and silicon, some aluminium and silicon are associated with potassium in specific regions highlighting the existence of more than one aluminosilicate type mineralising the cuticles (Fig. 4r–t). Importantly, the distribution of aluminosilicates in this film is non-uniform, as within the samples exhibiting the black film, aluminium, silicon, and potassium are not distributed on the entire surface of the samples (Fig. 4r–t). The absence of these elements within the samples could mean that aluminosilicate nuclei are too small to be detected. It could also mean that specific regions of the shrimp are more conducive to mineralisation than others, although no heterogeneity of aluminosilicates is visible at a macroscale. This black film forming on decaying marine arthropod has not been reported in previous publications on decay experiments although, possibly similar aluminosilicate veneers have been observed on scallop muscle tissues buried in kaolinite[20] and on anemone decaying on kaolinite substrates[21]. However, directly comparing the results of Newman et al.[20], Slagter et al.[21], and this work is a challenging task considering major differences in their respective experimental design.

Considering that the black film formed uniquely on marine shrimps decaying on kaolinite at 1.024 psu, the precipitation of clay minerals on the cuticle is likely dictated by complex conditions such as the limited oxygen around a decaying carcass, salt that can act as a catalyser, changes in pH during the decay of the carcass, and the nature of the organic matter in question. It has been previously observed that high salinities can drive the precipitation of certain clays in natural systems[33], which could explain why this film did not precipitate at low salinities and in freshwater. The formation of the black film could also be microbially mediated, which might lead to the favourable dissolution of kaolinite and its subsequent precipitation on shrimp carcasses.

## Black film formation happens through the precipitation of new aluminosilicates

These results have significant implications for understanding the fossil record. The mineralisation of labile structures in aluminosilicates has been a long-debated topic. Initially, Orr et al.[3] suggested that the enrichment of labile structures in aluminosilicates begins early during diagenesis, stabilizing morphologies over geological times. This proposition was contested in

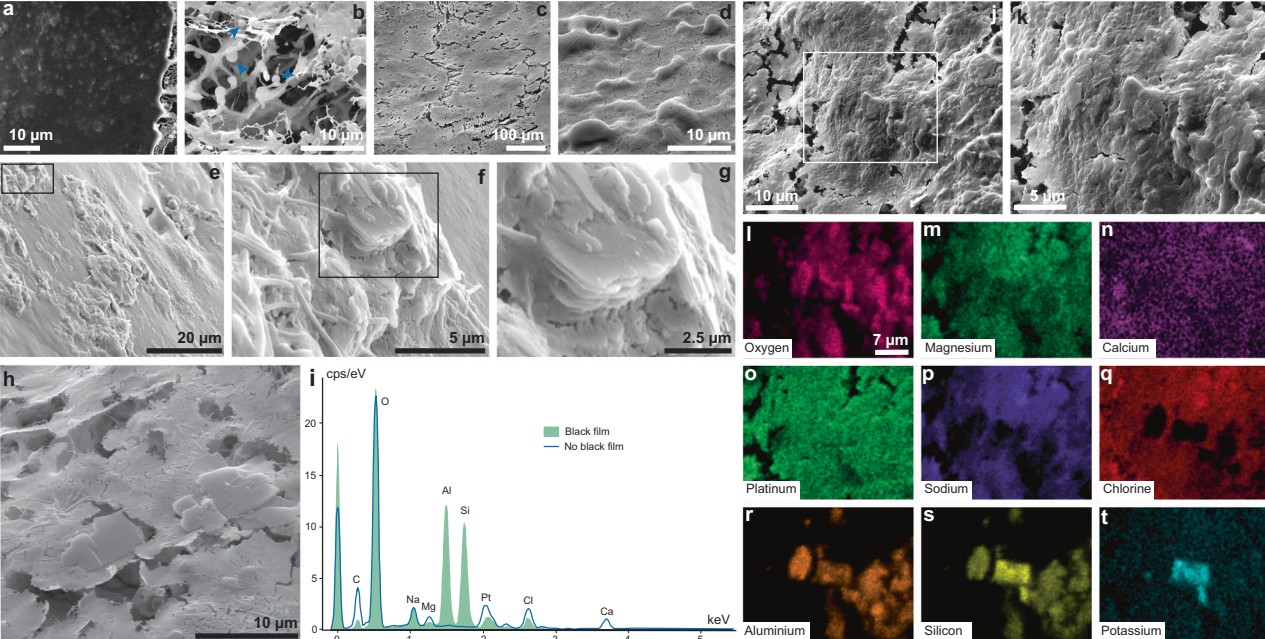

**Fig. 4 | Cryogenic scanning electron microscopy analyses.** Analyses performed at 10 keV with a backscatter detector (**a**, **c**–**h**) and a secondary electron detector (**b**), and elemental mapping of an area with black film (**l**–**t**). **a**, **b** Marine shrimp on which no black film formed. **a** The cuticle is dark under backscattered electron detectors as it is rich in carbon and possibly some other light elements. **b** The cuticle is organic with no aluminosilicates and shows bacteria indicated with blue arrows. **c**–**h** Marine shrimps on which a black film formed. **c** The cuticle of shrimps with a black film shows minimal signs of decay and (**d**, **e**) is irregular due to the deposition of (**f**–**h**) some poorly crystallized aluminosilicates that can very occasionally develop their typical sheet structures. **f** Zoom in on the framed area in (**e**). **g** Zoom in on the framed area in (**f**). **h** Another area with some aluminosilicate sheets that start to form on the cuticle. **i** Spectra of energy dispersive elements of areas with no black films (*n* = 8; blue spectrum) and an area with black films (*n* = 8; green spectrum). The predominant elements of the 16 analysed areas are represented: carbon (C), oxygen (O), sodium (Na), magnesium (Mg), aluminium (Al), silicon (Si), platinum (Pt), chlorine (Cl), and calcium (Ca). **j** Cryo-SEM imaging of the cephalothorax of a marine shrimp on which the black film formed. **k** Zoom in on the framed area in (**j**), corresponding to the analysed area. **l**–**t** The distribution of chemical elements in the analysed area: oxygen, magnesium, calcium, platinum, sodium, chlorine, aluminium, silicon, and potassium.

subsequent studies that proposed mineralisation in aluminosilicates likely occurs later during maturation. This type of mineralisation would therefore play a minimal role in anatomical information retention[11,13,14], and as a result labile non-heavily biomineralized structures are originally preserved as organic material and later replicated by silicates under high temperatures and pressures. Based on fossils, Anderson et al.[9] demonstrated that this type of mineralisation likely happens during early diagenesis, supporting the theory of Orr et al.[3]. However, the exact timing and whether this process happens within days, weeks, or months, could not be elucidated based only on fossils. The results of this study provide the first direct experimental evidence that aluminosilicates start to precipitate five days after death (120 h) in specimens left to decay on kaolinite (Figs. 1–4).

Further, understanding the origin of the aluminosilicates associated with fossils is contentious. It is unclear if, during early diagenesis, the stabilization of morphology through mineralisation happens through the attachment of pre-existing minerals from the sediment matrix surrounding the decaying carcass (i.e., the kaolinite powder that was added at the beginning of the experiment[15]) or if newly formed minerals precipitate in the decay sites (i.e., minerals formed through a physicochemical gradient from surrounding elements[9]). Morphological data of the shrimp cuticle and its associated minerals favour the latter scenario. The observed minerals (Fig. 4e–h) are poorly crystallized and very often lack organization in sheets (Fig. 4f, g), which contrasts with traditional aluminosilicate sedimentary matrices that are characterized by large sheets that are almost perfectly aligned[34]. Also, the newly formed film is black, in contrast to the original white colour of the surrounding clay matrix. These colour differences, and the scarcity of large, ordered aluminosilicate sheets covering the shrimp cuticle (Fig. 4f–h) suggest rapid in-situ nucleation[35]. Although these nuclei are not easily observed due to their small sizes, their presence could be inferred by the aluminosilicate enrichment of the cuticle (Fig. 4). When

rapid nuclei formation outpaces mineral growth, it results in an enrichment of aluminium and silicon in small nuclei rather than well-structured sheets, which may develop later over time[36]. Since kaolinite is a highly reactive mineral[37], its dissolution provides a steep chemical gradient in the water column, which, together with the chemistry of the decaying carcass and the ASW (containing potassium), leads to the precipitation of newly formed cryptocrystalline aluminosilicates (Fig. 4g, k, r–t). The exact chemical cascade leading to the precipitation of these authigenic minerals could be explored in future experimental work, which could also investigate whether minerals other than kaolinite (that were not studied herein) would favour the same mineralisation pattern. Yet, the current observations provide direct experimental evidence for the precipitation of new aluminosilicates on decaying carcasses stabilizing their morphologies. Given that the black film only formed in the presence of kaolinite, and the same methodology was applied to all samples, the film cannot be the result of suspended clay minerals in the water that passively landed on the surface of the shrimp. If this were the case, then all shrimps in the presence of the three clay minerals would show their surfaces covered by clays.

**Taphonomic and ecological implications**

Since the specimens studied in these experiments were not fully buried, but were deposited at the water-sediment interface, the results provide insight into the possible duration of the biostratinomic phase of the taphonomic process. The experiments revealed that marine arthropods deposited on kaolinite have the potential to retain all external anatomical features for many days after death (Figs. 1a and 2). This can be due to the inhibition of certain bacterial strains by kaolinite[18,31] and by the precipitation of authigenic aluminosilicates on the carcass in the presence of kaolinite. This stabilization indicates that a prolonged biostratinomic window for several days, and possibly weeks, postmortem was possible from the point of view of

the structural integrity of the carcass. Of course, many aspects of the seafloor were not replicated in the experimental setup, and in a natural setting, an unburied carcass could potentially be subjected to destructive influences such as the action of scavengers or high wave energy. However, in a calm environment with little perturbation, there is the potential for a carcass to be rapidly stabilized such that it could withstand several days of lying on the seafloor with minimal destruction of its cuticular structures. As such, instantaneous burial may not always have been necessary in marine Konservat-Lagerstätten containing decapod crustaceans, and perhaps other arthropods.

Eventually, burial will be needed to preserve animals over geological time frames, but its role does not seem to be essential in the first days, and possibly weeks postmortem, whenever kaolinite is present. It is also possible that the time frame under which external anatomical information can be retained in the presence of kaolinite can be considerably expanded when combined with other conditions that may be favourable for the preservation of non-heavily biomineralized anatomies (low oxygen concentrations, low pH, high sediment input, low temperature) that were not tested herein[38]. Note that the suggested lack of instantaneous burial does not contradict observations of rapid obrution events in Burgess Shale-type deposits[39–44]. Obrution events are fast, but they do not always occur regularly in depositional environments and they depend on numerous parameters such as seasonality[45]. As such, animals could be decaying for days or weeks on the seafloor before their burial by a rapid sedimentary event.

The results of this experiment also highlight that the association of aluminosilicates with exceptional fossil preservation in sites such as the Burgess Shale is indeed the result of complex processes involving early mineralisation[9]. However, this mechanism is not exclusive, and more aluminosilicates certainly precipitated during later diagenesis and metamorphism, during which time some weathering of previously precipitated aluminosilicates could also have occurred[11]. These processes result in organisms preserved in aluminosilicates[3,9] in association with organic material[8,42]. Due to the flattening of this organic material over geological times, the resulting fossils would consist of carbonaceous compressions with accessory minerals. As such, all these modes and mechanisms of preservation are not contradictory but complementary, and they contribute altogether to our understanding of some of the most iconic fossil deposits recording the Cambrian Explosion.

In these deposits, arthropods are often the most abundant and diverse animal phylum[6]. The obtained results could suggest that the abundance of arthropods in the Cambrian may not solely indicate ecological dominance but could also be the result of a taphonomic process like the black film formation that stabilized their carcasses shortly after death and favoured their preservation over other taxa. However, since possibly similar authigenic clays were also observed on unburied decaying anemones[21], this hypothesis is yet to be fully proven by doing similar experiments to the one made herein on other animal groups. These experiments would investigate whether aluminosilicate precipitation on decaying carcasses other than arthropods can happen without burial or whether non-arthropod carcasses need to be buried for this process to occur (see for instance Newman et al.[20]). In addition, it is worth noting that the black film only forms with standard marine salinity (1.024 psu) and not with a lower salinity (1.019 psu). Therefore, the authigenic mineralisation of the arthropod carcasses may be limited to the marine realm though the range of viable salinities has not been constrained in detail and no upper bound was defined.

This finding has broader implications for understanding community preservation in Konservat-Lagerstätten recording the Cambrian Explosion. In these sites, a separation is usually made between carcasses decaying on the seafloor, termed "time-averaged assemblage," and those killed during the burial event, termed "census assemblage"[46,47]. Our experimental observations suggest that census assemblages might represent previously dead communities that were stabilized by aluminosilicates instead of freshly killed ones. This process might be operational in hundreds of Cambrian fossiliferous sedimentary layers where kaolinite has been identified in the sedimentary matrix[30], and the high preservation potential of arthropods in these sites does not necessarily imply the exclusive preservation of freshly killed organisms.

## Conclusion

The decay of freshwater and marine shrimps deposited on the surface of three different clays was investigated. Decay proceeds more slowly in the presence of kaolinite than in the presence of other clay minerals in marine settings, as has been suggested previously[17,31]. Moreover, a black film of poorly mineralized aluminosilicates forms shortly after death, only on shrimp carcasses in the presence of kaolinite and in the absence of burial. These aluminosilicates stabilize the overall morphology of the carcass for days, even weeks, after death, even when exposed to the chemical gradient of the water column. In this context, marine shrimps, and potentially marine arthropods, may not need to be captured alive by obrution events for the preservation of lightly biomineralized structures to occur. These results constitute a paradigm shift in our understanding of the processes driving exceptional preservation in deep time, particularly during the Cambrian, which saw the appearance and diversification of many modern animal groups.

## Materials and methods

Adult freshwater shrimps (*Neocaridina davidi*; 1.5 cm long) and marine shrimps (*Palaemon varians*; 3 cm long), that were raised at the Institute of Earth Sciences of the University of Lausanne, were used in the experiment. These species were chosen because they are phylogenetically constrained and well-studied anatomically[48–52].

Adult shrimps were euthanized using clove oil [$C_7H_{12}ClN_3O_2$] which was chosen to avoid mechanically damaging the animals and to induce their rapid death. One drop of clove oil was added to the head of each shrimp. After the death of the shrimp, the head was rinsed repeatedly with reverse osmosis deionized water until no remaining oil could be seen on the shrimp or the surface of the water when submerged. Shrimps were individually left to decay in sterilized acrylic boxes (5 × 3 × 2 cm), closed but not sealed, containing 5 g of sediment with one of three compositions: kaolinite, bentonite, or montmorillonite. The clays used have a purity between 80–90%, with the remaining 10–20% phase consisting dominantly of quartz minerals. Each shrimp deposited on the surface of the sediment was covered with 35 g of water: reverse osmosis deionized water for freshwater shrimps and artificial seawater (ASW) for marine shrimps, prepared to 1.024 psu with reverse osmosis deionized water and Aquarium Systems Reef Crystals. For the control samples, freshwater shrimps were left to decay in reverse osmosis deionized water and marine shrimps in ASW without introducing clays to the system. We opted to use pure reverse osmosis deionized water to limit the number of variables in the experiment by excluding bacteria, plankton, and other organisms found in aquaria and natural aquatic environments in highly variable proportions.

In total, twenty-three freshwater shrimps (seven replicates for each of the three clay minerals, and two replicates for the controls) and seventeen marine shrimps (four individual replicates for each of the three clay minerals, and five replicates for the controls) were used in this study. All samples were kept at room temperature in the dark to avoid bias in microbial growth following a similar protocol to Sansom (2014). For ten days, each specimen was imaged every 24 h using a SC50 5-megapixel colour camera (Olympus Life Science Solutions) with Olympus Stream Basic software (version 2.2; Olympus Soft Imaging Solutions). Following the ten days, specimens were allowed to continue to decay for up to 20 days, during which time they were checked daily, and additional photos were taken in the event of major taphonomic state changes.

The state of decay of each specimen was assessed every 24 h by assigning a taphonomic score (Supplementary Data). Three different taphonomic scores, numbered from 0 to 6 (0 = intact, 6 = completely degraded), were defined based on the degradation aspect of the cephalothoracic carapace, the appendages and the eyes of the animals in this study (see Supplementary Table 1 and Fig. 1 in Supplementary Material). To determine the global taphonomic score of each individual, the median of the

carapace, appendages and eyes taphonomic scores was calculated (see Supplementary Table 1 and Fig. 1 in Supplementary Material). For score 0, the shrimp is intact and transparent. Score 1 corresponds to the carapace and appendages becoming pink and opaque, and the eyes turning black. When biofilms and/or other coatings are observed on the decaying carcass, score 2 is attributed. Score 3 is observed when the carapace detaches from the cephalothorax, at least 50% of the appendages are disarticulated, and the eyes detach from the cephalothorax. Score 4 is attributed when the carapace is completely detached from the cephalothorax so the gills and internal organs are exposed, at least 50% of the appendages are detached, and the eyes are detached. Score 5 is assigned when the cephalothorax and carapace are separated from the abdomen, at least 50% of the appendages are broken into several small pieces, and the eyes show advanced deterioration. Finally, for score 6, the individual is completely degraded. For each shrimp species independently, differences in taphonomic scores through time for each clay system were assessed with an ordinal logistic regression using the software R 4.1.1 (R Core Team, 2021). When the interaction between the clay and time variables was significant, a contrast analysis was performed with *emmeans* package (version 1.8-2) in R studio.

Three additional marine shrimp replicates were investigated using Cryogenic Scanning Electron Microscopy (Cryo-SEM, $-140\,°C$) at the University of Lausanne. Two of the three marine shrimps were left to decay in the presence of 5 g of kaolinite and 35 g of ASW at 1.024 psu in an experimental setup similar to the previous experiments on marine shrimps. The third marine shrimp was left to decay in the presence of 5 g of kaolinite and 35 g of ASW at a slightly lower salinity of 1.019 psu, to help investigate why certain patterns are only observed in marine settings and were not observed in freshwater conditions. The three samples were coated with a 3 nm platinum layer. For each shrimp, SEM images, elemental spectra, and an elemental map of eight different random areas of the cephalothorax and the abdomen were acquired using Quanta FEG-250 Scanning Electron Microscope at 10 keV. For further information regarding the choice of model organisms, experimental design, and chemical investigations, please refer to the Supplementary Material.

## Data availability
All data necessary to replicate this work are available in the main text and in the Supplementary Material and Supplementary Data files.

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

## Acknowledgements
The authors thank A. Mucciolo for his assistance during SEM analyses and all members of the Anom Lab at the University of Lausanne, Switzerland, and numerous attendees of the PalAss Annual Meeting in Cambridge, UK for fruitful discussions. NC thanks the Master of Science in Behavior, Evolution, and Conservation of the University of Lausanne during which this work was designed. No ethical approval was required to do these experiments, which were funded by the Swiss National Science Foundation. The taxa used in these experiments are not protected in Switzerland. Individuals were euthanised ethically, and the experiments adhered to the 3Rs principle of research (reduce, reuse and recycle). NC and FS are funded by an SNF Ambizione Grant (PZ00P2_209102). JBA is supported by an SNF Sinergia Grant (198691) awarded to ACD and three other PIs.

## Author contributions
All authors designed the research. NC and JBA did the shrimp experiments and performed statistical analyses. NC, FS, JBA, and ACD did the initial elemental investigations, followed by more detailed elemental work by NC and FS. NC and FS interpreted and discussed the results with JBA and ACD. NC made the figures and wrote the initial version of the text with the help of all co-authors.

## Competing interests
The authors declare no competing interests.
