## [Transparent Peer Review file · Communications Earth & Environment]

Kaolinite induces rapid authigenic mineralisation in unburied shrimps

Corresponding Author: Dr Farid Saleh

Version 0:

Decision Letter:

Dear Dr Saleh,

Your manuscript titled "Kaolinite induces rapid authigenic mineralization of soft tissues within days of marine shrimp decay" has now been seen by 3 reviewers, whose comments are appended below. You will see that they find your work of some potential interest. However, they have raised quite substantial concerns that must be addressed. In light of these comments, we cannot accept the manuscript for publication, but would be interested in considering a revised version that fully addresses these serious concerns.

In this revision, we are setting a few editorial thresholds that should be met. These are:

- 1] Please fully justify the definition and use of 'soft tissue' as a descriptor of the studied exoskeletons, in response to the criticisms raised by reviewer #1.
- 2] The revision should include a more extended presentation and discussion of the impact of the results on considerations of time averaging, including a more thorough referencing of previous work and explanation of how this current work demonstrates new findings.
- 3] The specific requests concerning improvements in the description of methods (e.g., Reviewer 3 comments) should be met.

In addition, please check and confirm that you have adhered to all required ethical oversight regarding the use of these shrimp and include a statement to that effect in the Acknowledgements, even if it is just to say that no ethical approval was required.

We hope you will find the reviewers' comments useful as you decide how to proceed. Should additional work allow you to address these criticisms, we would be happy to look at a substantially revised manuscript. If you choose to take up this option, please either highlight all changes in the manuscript text file, or provide a list of the changes to the manuscript with your responses to the reviewers.

When resubmitting, please provide a point-by-point response to the reviewers' comments. Please submit your responses as a separate file, distinct from your cover letter where you can add responses to the Editors' comments that you do not want to be made available to the reviewers. Word files are preferred.

Important: The response to reviewers must not include any figures, tables or graphs. If you wish to respond to the reviewer reports with additional data in one of these formats, please add them to the main article or Supplementary Information, and refer to them in the rebuttal. Due to current technical limitations, any figures, tables, or graphs embedded in your rebuttal will not be included in the peer review file, if published.

If the revision process takes significantly longer than three months, we will be happy to reconsider your paper at a later date, as long as nothing similar has been accepted for publication at Communications Earth & Environment or published elsewhere in the meantime.

Please use the following link to submit your revised manuscript, point-by-point response to the reviewers' comments with a list of your changes to the manuscript text (which should be in a separate document to any cover letter), a tracked-changes version of the manuscript (as a PDF file) and any completed checklist:

Link Redacted

Please do not hesitate to contact us if you have any questions or would like to discuss the required revisions further. Thank you for the opportunity to review your work.

Best regards,

D'Arcy Meyer-Dombard, PhD
Editorial Board Member
Communications Earth & Environment
orcid.org/0000-0001-9862-4839

Carolina Ortiz Guerrero, PhD
Associate Editor
Communications Earth & Environment

EDITORIAL POLICIES AND FORMAT

If you decide to resubmit your paper, please ensure that your manuscript complies with our editorial policies and complete and upload the checklist below as a Related Manuscript file type with the revised article:

Editorial Policy Policy requirements
(Download the link to your computer as a PDF.)

- Behavioural and social science
- Ecological, evolutionary & environmental sciences
- Life sciences

<https://www.nature.com/documents/nr-reporting-summary.zip>

For your information, you can find some guidance regarding format requirements summarized on the following checklist: (<https://www.nature.com/documents/commsj-phys-style-formatting-checklist-article.pdf>) and formatting guide (<https://www.nature.com/documents/commsj-phys-style-formatting-guide-accept.pdf>).

REVIEWER COMMENTS:

Reviewer #1 (Remarks to the Author):

This manuscript deals with the results of a taphonomic experiment where the carcasses of two species of caridean shrimp, one marine, one fresh water, were left to decay on substrates composed of various clay minerals. The results of these experiments can be summarized as follows:

- Carcasses associated with a kaolinite substrate decayed more slowly than on other types of clay; while interesting, this finding is not new (Wilson & Butterfield 2014; McMahon et al. 2016; Naimark et al. 2016), as per the authors own admission.
- In marine shrimp decomposing in artificial sea water, growth of likely authigenic aluminosilicates occurs on the exoskeletal parts of the carcass. This finding is of significant interest regarding the debate on whether aluminosilicates in Burgess Shale-type preservation are early diagenetic, and aided in preservation, or are merely a late diagenetic feature, and hence of no preservational consequence. The current results suggest that the former is the case.
- No aluminosilicates form on carcasses in fresh water, or under reduced salinities. This, again, is not quite unexpected, as the role of salinity in the formation and its influence on the type of authigenic clays that are formed is well established (see e.g. Calvo et al. 1995).

Unfortunately, while some of the experimental results are of interest, the paper over-inflates the significance of these findings

through several mischaracterizations and misrepresentations.

First, the authors' work focusses solely on the exoskeletal elements of their caridean test subjects. They incorrectly claim that these exoskeletal elements equate to "soft tissues", from which they continue to extrapolate their findings to present a major advance of our understanding of "soft tissue preservation". Euarthropod exoskeletons are cuticles secreted by specialized epithelial cells in the epidermis – hence, they are cellular secretions, not tissues themselves. Euarthropod cuticle is composed of a matrix of intertwined polysaccharides (predominantly chitin) and proteins. Through localized de-watering, and to a lesser extent through cross-linking, these originally soft and flexible cuticles become hard and rigid in a process that is referred to as tanning. In addition, in decapods, like the shrimp used in these experiments, these cuticles are further reinforced through biomineralization with calcium salts. While the extent of biomineralization differs between different decapod taxa, all are biomineralizing to at least some degree. This is actually attested by the authors' own EDS data which show the presence of Ca in the cuticle – something they avoid commenting on (in fact, the apparent absence of Ca in the spectrum for the specimen covered in aluminosilicates suggests that the limited biomineralization may have dissolved as a consequence of acidification associated with decay). Consequently, while shrimp exoskeletons are relatively delicate and poorly biomineralized compared to e.g. crabs or lobsters, they nevertheless are quite definitely neither "soft" nor "tissues", but in fact fall on the more recalcitrant side of the spectrum of preservation. So, the authors' description of the exoskeletal elements of the shrimp as "soft tissues" is doubly wrong, and as a result their characterization of the significance of their findings for our understanding of real soft tissue preservation is misleading. In fact, the authors implicitly admit as much themselves when they write (p. 11, lines 328 – 331):

"These results could suggest that the abundance of arthropods in the Cambrian may not solely indicate ecological dominance but could also be the result of a taphonomic process like the black film formation that stabilized their carcasses shortly after death and favored their preservation over other taxa."

So, with this sentence, the authors actually admit themselves that their results are not applicable to "soft-bodied" animals or "soft-tissues", but are only relevant to euarthropods which by definition possess rigid, hardened exoskeletons. And it is not exactly a novel idea that even a poorly sclerotized euarthropod generally still has a better preservation potential than a genuinely soft-bodied organism like e.g. a polychaete or a chaetognath.

Next, the authors set up a classical straw man argument, by implying that previous workers assumed "exclusive preservation of freshly killed organisms" (p. 12, lines 372-373) which they then use to construct the claim they have now identified a previously-ignored potential source of time averaging in obrution deposits which should be taken into consideration in future ecological studies of exceptionally preserved faunas – thereby falsely implying that this is a novel insight with major previously unrealized implications for the study of the ecology of exceptionally preserved faunas. In fact, the late Harry Whittington in 1971 already commented on a significant presence of decayed specimens in the Burgess Shale. Back in 1977, Conway Morris noted the decay sequence of *Ottoia prolifica* as observed in the specimens, observed decay in other Burgess Shale priapulids, and further explicitly remarked that a considerable number of specimens of *Selkirkia columbia* must have been dead before transportation and burial. The same author (Conway Morris 1986) in his famous paper on the community structure of the Burgess Shale Phyllopod Bed again commented on the topic of carcasses being present among the preserved biota, and the need to distinguish between specimens that were alive and dead at time of burial for community studies. Caron & Jackson (2006) specifically looked at high resolution at the presence of carcasses vs individuals transported and buried alive in the Greater Phyllopod Bed and explicitly stated (Caron & Jackson 2006, p. 458):

"The relative ratio of species in different states of preservation varies through time, and in some instances the proportion of the time-averaged assemblage is much higher than the proportion of the census assemblage. This demonstrates that the event of burial is not always the main cause of death for many organisms."

The same authors followed up this study of the taphonomy of the Greater Phyllopod Bed with a work on the ecology (Caron & Jackson 2008). Here, they stated (Caron & Jackson 2008, p. 224):

"Rarefaction curves suggest that preburial decay had no effect on species richness (Caron, 2005; Caron and Jackson, 2006). Therefore the effect of time-averaging through preburial decay is not an important controlling factor in the community."

These are only a couple of very well-known references that immediately come to mind, but this list is far from exhaustive, and it is difficult to understand how the authors could not have been aware of these works. Suffice to say that researchers working on exceptional preservation generally have been cognizant of the presence of carcasses and the potential for time averaging in exceptionally preserved faunas for some 40+ years at the very least, and have taken this into consideration when studying the ecology of ancient biotas; indeed, it stands to reason that it is almost inevitable that some carcasses will be included in an exceptionally preserved assemblage, considering that even extremely labile groups like e.g. polychaetes will not instantaneously vanish upon death.

These two very major issues, which entirely undercut the premises of the manuscript, should in my opinion be enough to disqualify it from publication in its current form. However, there are also further shortcomings that, although of a lesser magnitude, still need to be noted.

In their methods section (p. 3, line 98-100, and supplementary text p. 1), the authors state:

"We opted to use pure ASW and deionized water to limit the number of variables in the experiment by excluding bacteria, plankton, and other organisms found in aquaria and natural aquatic environments in highly variable proportions."

So, apparently, they believe that de-ionized water is essentially sterile. This is incorrect. De-ionized water is exactly what the name says – it is water from which the ions have been removed through adsorption on a resin matrix. However, this process does nothing to remove neutrally charged molecules or microbiota. To assure that the water is sterile, it would need to be distilled, or possibly filtered through a dedicated nanofiltration systems. Given this, the authors' assertion that the majority of the bacteria present in their experimental system originated from the carcasses is tenuous at best.

In the text, the authors also constantly refer to "lagerstätten". First, this is a German noun, so it needs to be capitalized. Second, Seilacher originally distinguished two types of Lagerstätten – Konservat-Lagerstätten (exceptionally preserved biotas) and Konzentrat Lagerstätten (concentration deposits with an abundance of material); later authors sometimes added further types of Lagerstätten, e.g. "Echinoderm-Lagerstätten". So, for clarity, it is important to indicate which type of Lagerstätte one is talking about – and in this case, the authors should use "Konservat-Lagerstätten".

Further remarks and comments are included on the attached annotated manuscript.

Finally, there is an ethical dimension to this entire paper. Under European legislation, the authors were not required to obtain any approval for their research from an ethical committee. However, there is an increasing body of research that suggests that decapods are sentient and capable of feeling pain (e.g. Elwood 2012; Passantino et al. 2021), leading countries like Switzerland (where the authors are based) and New Zealand to outlaw the live boiling of lobsters. As a consequence, there have also been increasing calls to regulate the use of crustaceans in research and put their use under ethical oversight (e.g. Mather 2019; Passantino et al. 2021; Rowe 2018). Considering that none of the work presented here has any practical value whatsoever, but was carried out simply to satisfy the intellectual curiosity of the authors, I would like them (and, indeed, the editor) to ponder the following question: do you feel that it is ethically and morally justifiable to kill 43 potentially sentient individuals merely to quell your curiosity?

Ethics aside, this paper does present some interesting data that merit publication. However, the manuscript suffers from such significant inaccuracies and misrepresentations, which are used to construct an inflated narrative of the importance of the authors' work, that it really cannot be considered for publication anywhere in its current form.

Peter Van Roy
04 July 2024

References

- Calvo, J.P., Blanc-Valleron, M.M., Rodríguez-Arandía, J.P., Rouchy, J.M. & Sanz, M.E. 1995. Authigenic clay minerals in continental evaporitic environments. In: Thiry, M. & Simon-Coinçon, R., *Palaeoweathering, Palaeosurfaces and Related Continental Deposits*, pp. 129-151. Wiley-Blackwell & The International Association of Sedimentologists.
- Caron, J.B. & Jackson, D.A. 2008. Paleoeology of the Greater Phyllopod Bed community, Burgess Shale. *Palaeogeography, Palaeoclimatology, Palaeoecology* 258, 222-256.
- Caron, J.B. & Jackson, D.A. 2006. Taphonomy of the Greater Phyllopod Bed Community, Burgess Shale. *Palaios* 21 (5), 451–465.
- Conway Morris, S. 1977. Fossil priapulid worms. *Special Papers in Palaeontology* 20, 1–95.
- Conway Morris, S. 1986. The community structure of the Middle Cambrian phyllopod bed (Burgess Shale). *Palaeontology* 29 (3), 423–467.
- Elwood, R.W. 2012. Evidence for pain in decapod crustaceans. *Animal Welfare* 2012, 21, 23-27.
- Mather, J.A. 2019. Ethics and care: for animals, not just mammals. *Animals* 2019, 9(12), 1018.
- McMahon, S., Anderson, R.P., Saupe, E.E. & Briggs, D.E.G. 2016. Experimental evidence that clay inhibits bacterial decomposers: implications for preservation of organic fossils. *Geology* 44, 867–870.
- Naimark, E., Kalinia, M., Shokurov, A., Boeva, N., Markov, A. & Zaytseva, L. 2016. Decaying in different clays: implications for soft-tissue preservation. *Palaeontology* 59(4), 583-595.
- Passantino, A., Elwood, R.W. & Coluccio, P. 2021. Why protect decapod crustaceans used as models in biomedical research and in ecotoxicology? Ethical and legislative considerations. *Animals* 2021, 11(1), 73.
- Rowe, A. 2018. Should scientific research involving crustaceans require ethical review? *Journal of Agricultural and Environmental Ethics* 31, 625-634.
- Whittington, H.B. 1971. The Burgess Shale: history of research and preservation of fossils. *Symposium of the North American Paleontological Convention 1969, Part 1*, 1170-1201.
- Wilson, L.A. & Butterfield, N.J. 2014. Sediment effects on the preservation of Burgess Shale-type compression fossils.

Reviewer #2 (Remarks to the Author):

This manuscript by Nora Corthésy and colleagues firmly demonstrates, from experiments, that decay of arthropod carcasses is retarded to different degrees when lying on three investigated clay types (kaolinite, bentonite and montmorillonite), and argue how this may influence exceptional preservation in mainly Burgess Shale-type Lagerstätten.

The experiments are notable because they show that kaolinite is the most effective clay to retard decay, and stabilise tissues, even without burial, and that this stabilisation is associated with aluminosilicate clay templating on the carcasses surfaces. While tissue stabilisation and aluminosilicate clay templating has previously been demonstrated to occur in association with kaolinite in the decay experiments by e.g. Naimark et al. (2016a <https://doi.org/10.1111/pala.12246> and 2016b <https://doi.org/10.1017/jpa.2016.23>) and Newman et al. (2019 <http://dx.doi.org/10.2110/palo.2019.030>), the core strength of the manuscript is that it, through a large and well-documented dataset, adds compelling evidence that these processes start rapidly after death and without burial. Therefore, the main implication of this manuscript is that it convincingly shows that the fossilisation process already starts shortly after death and irrespective of burial. The manuscript is timely since there currently is a lot of work on understanding the preservational biases in fossil Lagerstätten and how these biases shape our interpretation of the fossil record. The chosen methods are well-argued and well-described, and the results are generally well-presented and support these conclusions.

As an ecological implication, the authors use this to argue that arthropod death assemblages on kaolinite-rich seabeds (purportedly the Burgess Shale Lagerstätten) may be subjected to a more extended time-averaging if carcasses can lie seemingly undisturbed for ~weeks. This section is the manuscript's weakest part. First, in my opinion, time-averaging at this temporal scale is not overly important for ecological interpretations as populations and communities mostly shift at a higher temporal scale (months/years). In fact, at the monthly scale, another decay experiment with shrimp larvae on kaolinite-surfaces show that the carcasses had dissolved after 3 to 4 months (Naimark et al., 2016b <https://doi.org/10.1017/jpa.2016.23>). Second, the authors implies that it is generally assumed that fossil assemblages in single-bed burial events only contain freshly killed animals, such as in the Burgess Shale. However, it is well established that e.g. Burgess Shale beds contain mixtures between freshly killed (census) assemblages as well as the accumulating death assemblages (Caron & Jackson 2006 <https://doi.org/10.2110/palo.2003.P05-070R> and others since them). However, the authors could still use their findings to make the argument that the completely preserved specimens interpreted to represent census assemblages in Caron & Jackson (2006) could alternatively represent recently deceased death assemblages.

Overall, the manuscript is good and deserves publication. The main results will predictably shape future experiments to investigate the timing and processes of exceptional preservation in clay systems. However, the text could be clarified in some sections and certain auxiliary implications scaled back a bit. The authors will be able to do this without changing anything to their experiments or data presentation.

Below are some specific comments and questions to the text:

ABSTRACT

Line 16-18: "This is the first experimental evidence for the replication of soft tissues in aluminosilicates shortly after death in Burgess Shale-type fossil lagerstätten recording the Cambrian Explosion." Unless I misunderstand the sentence, it should be clarified a bit since a) the experiments do not mimic the environmental conditions of Burgess Shale beds, and b) is not the first experimental evidence of aluminosilicate replication of soft tissues (see references above).

Line 19-20: "The preservation of morphology through aluminosilicates could frequently result in carcasses persisting on the seafloor for extended durations [...]" In my opinion, this sentence should define the length of "extended durations" since these can vary from days to years, depending on the chosen scale of investigation.

Line 20-21: "meaning that instantaneous or near-instantaneous burial may not be a prerequisite for soft tissue preservation as usually thought." Again, "instantaneous" and "near-instantaneous" would benefit from being defined to avoid confusion since researchers may work at very different time scales. I think it is commonly assumed that many fossils in exceptionally preserved beds may represent decayed death assemblages. Alternatively, in my opinion (and therefore just a subjective suggestion), the authors could strengthen the manuscript if they instead highlight that their results indicate that the fossilisation process starts before burial.

Line 22-25: As I argued above, I think it is uncertain how much this cryptic time-averaging (albeit convincingly demonstrated) will affect palaeoecological interpretations.

INTRODUCTION

Line 70-71: "Slower degradation is accompanied by the precipitation of newly formed aluminosilicates on the marine shrimp carcasses, providing the first experimental evidence for this mode of authigenic mineralization." It should be clarified that while aluminosilicate clay templating/mineralization has been shown in other experiments (see refs for Naimark et al., 2016a,b and Newman 2019 above), the manuscript provides the first convincing evidence that the mineralization is authigenic.

Line 76-78: "This observation holds particular significance in the context of the Cambrian Explosion, as kaolinite is frequently observed in association with soft tissues in marine rock from the Cambrian Period." This argument would be strengthened if the authors provided references showing the frequent observation of soft tissues associated with kaolinite-rich sediments.

MATERIAL & METHODS

Line 114: "Seven different taphonomic scores, numbered from 0 to 6 (0 = intact, 6 = completely degraded), were defined based on the global degradation aspect of the animal (see Tab. S1 & 114 Fig. S1 in Supplementary Materials)." I am uncertain what "global degradation aspect" refers to; does it refer to the general degradation aspects observed in this study only or in all decay experiments collectively?

RESULTS & DISCUSSION

Line 214-215: "The black film (Fig. 2) was formed uniquely in the experimental condition of marine shrimp decaying on kaolinite at 1.024 psu." Did the black film form on all carcasses decaying on kaolinite at 1.024 psu or only some?

Line 224-225: "This black film has not been reported in previous publications on decay experiments." Could the black films ("clay veneers") in Newman et al. (2019 <http://dx.doi.org/10.2110/palo.2019.030>) possibly be similar? Albeit from a different experimental setup.

Line 236-239: "For the specimens on which no black film formed, the remains of the cuticle, observed with the Cryo-SEM, show an organic surface (Fig. 4A), covered by bacteria, mainly coccoids, ranging in size from 0.6 to 238 1.5 μ m (Fig. 4B)." This is a bit unclear to me. According to the methods (lines 120-130), three carcasses decaying on kaolinite were prepared for Cryo-SEM. How many of these did not form black films? The text on lines 236-239 indicates that it did not form on multiple carcasses and not just the single carcass on kaolinite at 1.019 psu.

Line 271: "Importantly, the distribution of aluminosilicates in this film is non-uniform" Would it be possible to provide some kind of meaningful general density/frequency intervals of the aluminosilicate sheets to estimate their prevalence within the black films?

Line 273-276: "The absence of these elements in some areas could mean that aluminosilicate nuclei are too small to be detected and that specific regions of the shrimp are more conducive to mineralization than others, although no clear differences could be seen while investigating the shrimp photographs at a macro scale." Have these nuclei been observed or are they assumed to be present? If they have not been observed, other possibilities could be discussed. Is it, for example, a possibility that the aluminosilicates just have not precipitated everywhere despite the black film being present at macro scale?

Line 306-310: "These color differences, and the scarcity of large, ordered aluminosilicate sheets covering the shrimp cuticle (Fig. 4F-H) suggest rapid in-situ nucleation (32). When rapid nuclei formation outpaces mineral growth, it results in an enrichment of aluminum and silicon in small nuclei rather than well-structured sheets, which may develop later over time (33)" Again, it is unclear if these nuclei have been observed. If not, I think it should be clearly stated in the text.

Line 327-322: "The formation of a black film has, thus far, been exclusively observed on the carcasses of marine arthropods. These results could suggest that the abundance of arthropods in the Cambrian may not solely indicate ecological dominance but could also be the result of a taphonomic process like the black film formation that stabilized their carcasses shortly after death and favored their preservation over other taxa." It is my understanding that these experiments have only been performed on arthropods. Unless I am mistaken, I think it is a stretch to attribute the rich arthropod record in Cambrian Lagerstätten to this process, especially since kaolinite has not, to my knowledge, been observed in all other Lagerstätten. Additionally, if the black clay veneers in the scallops of Newman et al. (2019 <http://dx.doi.org/10.2110/palo.2019.030>) is similar to the black films shown in this manuscript, the taphonomic process may extend to other phyla than just arthropods.

Line 336: "[...] driven by the precipitation of authigenic aluminosilicates on the carcass (Fig. 6 A, B)." Has it been fully established that the driver of preservation at this timing is the precipitation of authigenic aluminosilicates rather than kaolinites inhibition of decay-inducing bacteria with authigenic precipitation as a side product?

Line 344-345: "As such, instantaneous burial may not always have been necessary in marine fossil lagerstätten containing decapod crustaceans, and perhaps other arthropods." and Line 351-256: "Note that the suggested lack of instantaneous burial does not contradict observations of rapid obrution events in Burgess Shale-type deposits (37–42). Obrution events are fast, but they do not always occur regularly in depositional environments and they depend on numerous parameters such as seasonality (43). As such, animals could be decaying for days or weeks on the seafloor before their burial by a rapid sedimentary event." As stated above, Caron & Jackson (2006 <https://doi.org/10.2110/palo.2003.P05-070R>) already argued that Burgess Shale-type burial events contain both census assemblages as well as accumulated death assemblages, implying that instantaneous burial is not necessary for exceptional preservation and that some fossils may have accumulated (and decayed) on the seafloor for prolonged periods. However, Caron & Jackson do assume that completely preserved fossils represent freshly killed census assemblages.

Line 357-377: This section argues for cryptic time-averaging since the black film on kaolinite beds can retard decay for weeks. However, since the composition of communities and populations in a given environment does not vary significantly

at the daily/weekly-scale, but rather at a monthly/yearly scale, it is uncertain what the impact of this geologically very short time-averaging is. Perhaps it could be speculated that the black films, in combination with other processes, could potentially induce time-averaging at a larger scale. Like for the section above, Caron & Jackson (2006) (and several other publications since) have already argued that Burgess Shales-type burial events already consist of both census and death assemblages.

Line 383-386: "Any part of the organism that has not been replicated by aluminosilicates would, in turn, be preserved in carbon (8, 40)(Fig. 6D), which would result in a complex fossil preserved in both carbonaceous compressions and aluminosilicates owing to their flattening over geological times (3, 9)(Fig. 6D)." It could be clarified whether aluminosilicate coats or replaces the carbon. As I read it right now, the text suggests, to me, that the aluminosilicates replace the carbon and that carbon only occurs where there are no aluminosilicate sheets, rather than aluminosilicates co-occurring with the carbon. Additionally, the text currently appears to suggest that the flattening is the driver of the complexity and not the authigenic aluminosilification.

Congratulations on the manuscript, which I hope to see published.

Best wishes,
Morten Lunde Nielsen

Reviewer #3 (Remarks to the Author):

Comments for Author

This is a really interesting manuscript that explores the interactions and effects of substrate on decay in shrimps and investigates Burgess Shale-type preservation pathways. The work is well-written, and the figures complement the text. The approach and methods used are sensible and highly appropriate for this field of work, and they provide a nice framework for future workers.

The resulting novel evidence of the rapid formation of an aluminosilicate covering and the disruption of the decay process on a Kaolinite substrate is very exciting. I agree with the authors that, as well as answering a long-standing question surrounding the timing of Burgess Shale-type preservation, the findings have wide-ranging implications for the interpretation of fossils and fossil-bearing sites and subsequent ecological reconstructions. I think that as this work aims to quantify taphonomic processes and identify pathways that underpin the fossil record, it will be of interest to a broad range of palaeontologists and also to adjacent fields such as archaeology and forensics.

While I think this is an important piece of work, I feel that some minor revisions are needed to improve the repeatability of the methods and clarify some areas of the manuscript. These do not require additional analysis or data collection and are largely minor edits, so I am confident the authors can quickly address them.

Overall, this is an excellent piece of work, and I recommend it for publication once the revisions have been made.

General comments:

- My primary concern is that I don't think the methods are detailed enough to support the reader, especially with the taphonomic scoring, but also with the general setup. As it is, the manuscript gives a good general explanation, but I think it would be difficult to replicate. I think the addition of extra text, largely in the extended methods, would quickly rectify this.
- Taphonomic Scoring: I think it is a sensible system, and I can broadly understand what was done, but as scoring is the key method for quantifying the decay data for statistical comparison, more explicate information on how the system was established and applied would be appropriate. The methods provide a short statement that the scoring system was derived through observation, but I don't think it is clear enough to understand how or why the boundaries were established. The addition of information such as what was observed and for how long along with why those markers were chosen as a boundary (I assume it was due to major or distinct morphological changes, but I don't know) would help the reader understand and also help future workers adapt the method. Similarly, while Fig S1 is very helpful for communicating morphological states, I would like more information on how scores are assigned and the thresholds for scores. As it is, I think I would struggle to replicate the scoring system confidently and get the same results for each specimen. For example, in the case of appendages, how many are needed to be disarticulated to score a 3? Is it all, or is just one enough? How do I score intermediate or mixes of categories? If I had a shrimp that had a carapace cuticle detaching from internal organs (3) but no disarticulated appendages (2) or detached eyes (2), would I score that as a 3 or a 2? I think this is especially important to help support the finding of a decay plateau/delay in the Kaolinite specimens – is it truly a pause, or do they decay in a different order, or is it an artefact of the scoring system? It is hard to distinguish this as a reader, so including the details of the system in the methods would provide more evident support for the findings.
- Related to this, I think it would also be useful to have a breakdown of the scores or morphology per specimen, but I appreciate that Fig 3 & S3 do give a sense of this information so it is less important than the detail of the system itself.
- There could also be some more detail given to the physical set-up (albeit it constrained in the supplementary extended section rather than in the main body of the manuscript). Inclusion of a figure or description of the setup of the acrylic boxes, including the dimensions, number of shrimps per box etc. would be very helpful. It would help give a sense of the size of the animals in relation to the surrounding clay surfaces, and help future workers who wish to use your protocols.
- Additionally, with the cryo-SEM and the EDS, indication of where on the shrimp the scans were taken from would help the

reader understand if differences in the distribution of the aluminosilicate considered in the discussion are related to the location or not.

- Black film vs White biofilm – This is a slightly pedantic question but how have you established that the white film is a biofilm and not something else? Perhaps reword to white film?

- Some consideration of the sample size would be appropriate in the discussion.

- Finding that the black film does not form in lower salinity is really interesting, and I think it is worth a few sentences of consideration in the Taphonomic and Ecological implications section. Do you have a sense of what this could present like over salinity gradients in coastal/inlet regions? Alongside the temporal complexity that this preservation style might create, it seems like it might capture some additional localised environmental biases too, adding to the difficulties of understanding lagerstätten.

- Fig 3 E - This is a lovely figure that really complements the descriptive sections and helps communicate the variation in decay seen. I assume the pink objects and black blobs in the small circles are meant to represent bacteria or the films forming? A key or explanation in the caption would help to clarify this.

- Fig 4 B – Highlighting an example of a bacteria on this would help orientate the reader.

- There is some inconsistency throughout on the use of shrimp/shrimps as a plural, e.g., L134 marine shrimps is used, then on L134-5 marine shrimp is used. It's largely not a problem for understanding the work, but is worth checking. The only place I think it causes some ambiguity is L216 where I could read the "shrimps" as marine or fresh water shrimp which does change the narrative of the paragraph.

Comments on the manuscript:

Lines 22-24 – Consider rewording this sentence. I feel it gives the impression that heavily decayed morphology would be similar to 'fresh' morphology, rather than there being a delay in the decay process so the older specimens are less decayed than expected impacting temporal reconstructions (which I think is what this section is indicating?)

Lines 51-53 - "If their association...", "...unclear whether these aluminosilicates..." - In this section I find the use of them/these slightly hides the subject of the sentences and disrupts the readability. It could be replaced both with "the" and keep the meaning.

Line 63 – "... for a range of animals,..." please include some reference examples here, such as Sansom et al 2011 and Briggs & Kear 1993 (or similar)

Line 65-78 – I think a small direct sentence on why this work was done (e.g., in order to test/investigate...etc) would create a stronger link between the introduction and this paragraph.

Line 94-96 – This is an important point but in this form reads like a discussion point. Consider editing it to highlight that consistent use of clove oil is the control a variable as done with the following sentences about ASW (and maybe moved to the area above about euthanasia)

Lines 100-101 – Consider removing this sentence, I think it is repeating the previous sentence in a less informative way. It might be that some of the nuance of it has been lost in the transition from extended method to main body?

Lines 109-111 – I understand this sentence to mean that after 10 days photos were only taken if there were major taphonomic changes – is that correct? – or was a photo taken every week while also checking for major changes regardless of whether or not any changes had happened? I could read it both ways so is worth checking to remove ambiguity.

Additionally, this is slightly different from the method in the supplementary where the check is "periodically" not weekly.

Line 110 – weekly checked → checked weekly

L164&166 Fig 1 Caption - consider rewording "is detaching" maybe to 'detaches' or 'beings to detach'

Line 211 – "...organs are removed.." Consider rewording, 'removed' sounds intentional, 'displaced' maybe?

Line 213 – Can you elaborate on the one difference? I don't think it needs a lot of discussion, just stating what it is would be fine.

Lines 271&273 – "certain areas" and "in some areas" – Where are the certain areas and are they consistent through the shrimp? The elemental variation within the body maybe linked to analysis artefacts as suggested, but it's quite hard to judge that as a reader without some understanding of the areas mentioned. If it's not random then something else might be happening that is related to the organism. This might be fixed by changing the word choice – deleting certain and some – but I think an idea of the distribution of these areas would be helpful to support the point, even if constrained to the supplementary material.

Communications Earth & Environment is committed to improving transparency in authorship. As part of our efforts in this direction, we are now requesting that all authors identified as 'corresponding author' create and link their Open Researcher

and Contributor Identifier (ORCID) with their account on the Manuscript Tracking System prior to acceptance. ORCID helps the scientific community achieve unambiguous attribution of all scholarly contributions. You can create and link your ORCID from the home page of the Manuscript Tracking System by clicking on 'Modify my Springer Nature account' and following the instructions in the link below. Please also inform all co-authors that they can add their ORCIDs to their accounts and that they must do so prior to acceptance.

Version 1:

Decision Letter:

Dear Dr Saleh,

Your revised manuscript titled "Kaolinite induces rapid authigenic mineralisation in unburied shrimps" has now been seen by our original reviewers #2 and #3, whose comments appear below. In light of their advice we are delighted to say that we are happy, in principle, to publish a suitably revised version in Communications Earth & Environment.

We therefore invite you to revise your paper one last time to address the remaining concerns of our reviewers. At the same time we ask that you edit your manuscript to comply with our format requirements and to maximise the accessibility and therefore the impact of your work.

EDITORIAL REQUESTS:

****Please take care to match our formatting and policy requirements. We will check revised manuscript and return manuscripts that do not comply. Such requests will lead to delays. ****

SUBMISSION INFORMATION:

OPEN ACCESS:

Communications Earth & Environment is a fully open access journal. Articles are made freely accessible on publication. For further information about article processing charges, open access funding, and advice and support from Nature Research, please visit <https://www.nature.com/commsenv/open-access>

Link Redacted

Best regards,

Carolina Ortiz Guerrero, Ph.D.
Associate Editor
Communications Earth & Environment

D'Arcy Meyer-Dombard, PhD
Editorial Board Member
Communications Earth & Environment
orcid.org/0000-0001-9862-4839

REVIEWERS' COMMENTS:

Reviewer #2 (Remarks to the Author):

The authors' replies to my own and reviewer 1's comments are satisfactory. The authors also satisfactorily handled the three outlined editorial thresholds.

I therefore consider the manuscript ready for publication.

Best wishes,
Morten Lunde Nielsen

Reviewer #3 (Remarks to the Author):

I am grateful for the authors' thoughtful consideration and thorough approach to revisions. I am happy that the authors have fully addressed my comments and concerns from the original review. I think the changes have improved the work's clarity, removed ambiguity and strengthened the overall message of the paper.

For the specific editor comments:

1] Please fully justify the definition and use of 'soft tissue' as a descriptor of the studied exoskeletons, in response to the criticisms raised by reviewer #1.

I agree with reviewer #1 that shrimp cuticle is technically not a "soft-tissue". However, I agree with the authors that as shrimp cuticle largely exhibits the same preservation patterns as soft-tissues the use of "soft-tissue" did indicate the inherent preservation potential of the cuticle. While it is just a terminology change and largely doesn't impact the message of the paper, I think the authors' response and related changes are appropriate and increase the precision of the work.

2] The revision should include a more extended presentation and discussion of the impact of the results on considerations of time averaging, including a more thorough referencing of previous work and explanation of how this current work demonstrates new findings.

I think the changes to the discussion, including expansion and revision of the text throughout have addressed these concerns. There is an increased consideration of the place of the work within a wider context, and relation to other works, along with more explicate description of the novelty of the presented work.

3] The specific requests concerning improvements in the description of methods (e.g., Reviewer 3 comments) should be met.

I think this response to this has been detailed and well thought through. I think the additional information has improved the repeatability of the protocols and clarified what exactly was done. Improvements to the taphonomic scoring descriptions greatly improve the readers' understanding and subsequent interpretation of the results

Overall I think the revisions have improved the message and the manuscript greatly, I have no hesitation in recommending this work for publication.

**Kaolinite induces rapid authigenic mineralization of soft tissues within days**
**of marine shrimp decay**

Nora Corthésy^{1*}, Farid Saleh^{1*}, Jonathan B. Antcliff¹, and Allison C. Daley¹

¹Institute of Earth Sciences, University of Lausanne, Géopolis, CH-1015 Lausanne,
Switzerland

Corresponding authors: N. Corthésy (nora.corthesy@unil.ch)

F. Saleh (farid.nassim.saleh@gmail.com)

**ABSTRACT**

The fossil record is essential for the reconstruction of past ecosystems and their evolution,
especially fossil **lagerstätten** where soft tissues are preserved. Understanding fossilization
processes, including decay and mineralization, is crucial for accurately interpreting ancient
morphologies. Here, we investigate the decay of marine and freshwater shrimps deposited on
the surface of three different clay beds. In kaolinite systems, cryogenic scanning electron
microscopy shows a black film comprised of newly formed anhedral and cryptocrystalline
aluminosilicates on marine shrimp cuticles, stabilizing their overall morphology. This is the
first experimental evidence for the replication of **soft tissues** in aluminosilicates shortly after
death in Burgess Shale-type fossil **lagerstätten** recording the Cambrian Explosion. The
preservation of morphology through aluminosilicates could frequently result in carcasses
persisting on the seafloor for extended durations, meaning that instantaneous or near-
instantaneous burial may not be a prerequisite for **soft tissue** preservation as usually thought.
Given the extremely detailed replication of cuticular anatomy by the black film
aluminosilicates, it would be difficult to distinguish fossilized organisms that had decayed for
24 weeks from those that were freshly killed, **indicating cryptic time averaging may exist in**
**Cambrian communities discovered in kaolinite-rich levels.**

**KEYWORDS**

exceptional fossil preservation, experimental taphonomy, *Palaemon varians*, kaolinite,
silicates, authigenic mineralization

INTRODUCTION

Under typical environmental conditions, soft, non-mineralized organic tissues are rapidly
destroyed through cell autolysis, heterotrophy by micro- and macro-organisms, and denaturing
of structural elements under the changing chemical conditions of decay. Yet, some incredibly
delicate tissues survive as fossils for hundreds of millions of years. For example, the exquisite
fossils of the Burgess Shale (~508 million years), and Burgess Shale-type preservation more
broadly, record the early diversification of animals during the Cambrian Explosion (1–5). In
sites with Burgess Shale-type preservation, diverse taxa such as annelids, arthropods,
lobopodians, and mollusks are preserved with their soft tissues (6). These fossils are often found
as carbonaceous compressions (7, 8) associated with aluminosilicate minerals (3, 9). Despite
the importance of the Burgess Shale for understanding major evolutionary events, the exact
processes behind its preservation remain shrouded in mystery to some extent.

Geological processes must have interrupted biological decay to transform labile soft tissues into
enduring rock. In other words, soft tissue preservation requires structural stabilization of tissues
by permineralization or templated mineral growth to outpace decay processes such as microbial
metabolism and cell autolysis. The spatial association of soft tissues and aluminosilicate sheets
has been observed in fossils with Burgess Shale-type preservation (3, 9–14), however, the
timing of soft tissue replication in aluminosilicate sheets is controversial, with some suggesting
that it occurs during early diagenesis (3, 9–11), and others suggesting it occurs during
maturation, a much later stage of fossilization (12–14). If soft tissue preservation is truly
governed by early diagenetic interactions between organic structures and clay minerals, then
one would expect to see them associated in the earliest stages of decay. If their association does
not develop until later, then other mechanisms must be responsible for the stabilization of
organic structures. It is also unclear whether these aluminosilicates result from the simple
attachment of pre-existing clays, such as kaolinite, in the matrix (9, 15) or from the precipitation
and formation of new authigenic minerals (9, 16).

Experiments investigating the interaction between clay minerals and decaying carcasses help
to provide meaningful controls on the relevant conditions for soft tissue replication in
aluminosilicate sheets. Decay experiments have proven highly valuable in cataloging the loss
and retention of anatomical information under various environmental conditions, some in the
presence of sediments (16–19) and some without (20–25). Excluding sediments is a sensible
and coherent approach to understand the baseline of the processes of decay by limiting the
variables in a specific system. Recent years have seen extensive mapping of the patterns of
decay sequences for a range of animals, and it is now possible to examine more process-
orientated questions with more interactions and complexity introduced to the system.

Herein, deceased specimens of marine shrimp *Palaemon varians* and freshwater shrimp
*Neocaridina davidi* were placed atop beds of three different clay compositions under seawater
medium to simulate conditions at the sediment-water interface without burial. These
experiments show that the decay of marine shrimps deposited on the surface of kaolinite is
slower than that of marine shrimps deposited on bentonite and montmorillonite, or without clay.
Slower degradation is accompanied by the precipitation of newly formed aluminosilicates on
the marine shrimp carcasses, providing the first experimental evidence for this mode of
authigenic mineralization. These findings provide novel perspectives on the origin and timing
of the replication of soft tissues in aluminosilicates, suggesting that instantaneous or near-
instantaneous burial may not be a prerequisite for soft-tissue preservation as the morphology of
soft tissues remained stable/undistorted for several weeks in the presence of kaolinite clays,
without burial. This observation holds particular significance in the context of the Cambrian
Explosion, as kaolinite is frequently observed in association with soft tissues in marine rock
from the Cambrian Period.

MATERIAL & METHODS

Adult freshwater shrimps (*Neocaridina davidi*) and marine shrimps (*Palaemon varians*), that
were raised at the Institute of Earth Sciences of the University of Lausanne, were used in the
experiment. These species were chosen because they are phylogenetically constrained and well-
studied anatomically (26–30).

Adult shrimps were euthanized using clove oil [$C_7H_{12}ClN_3O_2$] which was chosen to avoid
mechanically damaging the animals and to induce their rapid death within minutes. One drop
of clove oil was added to the head of the shrimp. After the death of the shrimp, the head was
rinsed repeatedly with deionized water until no remaining oil could be seen on the shrimp or
the surface of the water when submerged. Shrimps were left to decay in sterilized acrylic boxes,
closed but not sealed, containing 5g of sediment with one of three compositions: kaolinite,
bentonite, or montmorillonite. Sediment and shrimp were covered with 35g of water: artificial
seawater (ASW, 1.024 psu) for marine shrimps and deionized water for freshwater shrimps.
The clays used have a purity between 80-90%, with the remaining 10-20% phase consisting
dominantly of quartz minerals. Considering that all shrimps were euthanized using clove oil,
using this oil cannot explain why some decay patterns occurred only in the presence of specific
clays. For the control samples, freshwater shrimps were left to decay in deionized water and
marine shrimps in ASW without introducing clays to the system. We opted to use pure ASW
and deionized water to limit the number of variables in the experiment by excluding bacteria,
plankton, and other organisms found in aquaria and natural aquatic environments in highly
variable proportions. Using pure ASW and deionized water is the simplest way to see how clays
impact the decay of freshwater and marine shrimps.

In total, twenty-three freshwater shrimps (seven replicates for each of the three clay minerals,
and two replicates for the controls) and seventeen marine shrimps (four individual replicates
for each of the three clay minerals, and five replicates for the controls) were used in this study.
All samples were kept at room temperature in the dark to avoid bias in microbial growth
following a similar protocol to Sansom (2014). For ten days, each specimen was imaged every
24 hours using a SC50 5-megapixel color camera (Olympus Life Science Solutions) with
Olympus Stream Basic software (version 2.2; Olympus Soft Imaging Solutions). Following the
ten days, specimens were allowed to continue to decay for up to 21 days, during which time
they were weekly checked for major taphonomic state changes, at which point photos were
taken.

The state of decay of each specimen was assessed every 24 hours by assigning a taphonomic
score. Seven different taphonomic scores, numbered from 0 to 6 (0 = intact, 6 = completely
degraded), were defined based on the global degradation aspect of the animal (see Tab. S1 &
Fig. S1 in *Supplementary Materials*). For each shrimp species independently, differences in
taphonomic scores through time for each clay system were assessed with an ordinal logistic
regression using the software R 4.1.1 (R Core Team, 2021). When the interaction between the
clay and time variables was significant, a contrast analysis was performed with *emmeans*
package (version 1.8-2) in R studio.

Three additional marine shrimp replicates were investigated using Cryogenic Scanning
Electron Microscopy (Cryo-SEM, -140°C) at the University of Lausanne. Two of the three
marine shrimps were left to decay in the presence of 5g of kaolinite and 35g of ASW at 1.024
123 psu in an experimental setup similar to the previous experiments on marine shrimps. The third
marine shrimp was left to decay in the presence of 5g of kaolinite and 35g of ASW at a slightly
lower salinity of 1.019 psu, to help investigate why certain patterns are only observed in marine
settings, and were not observed in freshwater conditions. The three samples were coated with a

3nm platinum layer. SEM images, sixteen elemental spectra, and an elemental map of a specific
area were acquired using Quanta FEG-250 Scanning Electron Microscope at 10 keV. For further
information regarding the choice of model organisms, the experimental design, chemical
investigations, and the statistical analyses, please refer to the *Supplementary Materials*.

**RESULTS & DISCUSSION**

**Kaolinite slows down marine shrimp decay.** Immediately following euthanasia, marine
shrimps are transparent and intact (Figs 1, S1A). At 24 hours, under all conditions, marine
shrimp become opaque and pink/white, and the eyes start to turn black (Taphonomic score 1;
Tab. S1, Fig. S1B). Over the next three days (at 96 hours of decay), minimal anatomical change
is observed in the presence of any of the three clays (Fig. 1A). However, starting at 120 hours,
a fine black film appears on the cuticle of marine shrimps on kaolinite (Taphonomic score 2;
Figs. 1A, 2), outlining the exoskeleton in exquisite detail (Figs. 1A, 2). This film is absent from
all other experimental conditions (Fig. 1B, C), which are instead covered by a white biofilm as
early as 24 hours after the start of decay (Taphonomic score 2; Figs. 1B, C, S1C, D). For marine
controls, the cuticle starts to detach from the abdomen at 96 hours (Taphonomic score 3; Fig.
1C). The cuticles of marine shrimps decaying on bentonite and montmorillonite begin to detach
from the abdomen between 120 to 168 hours into the experiment (Taphonomic score 3; Fig. 1B),
whereas, during the same period for the controls, the cuticle is completely detached from the
cephalothorax, which starts to separate from the abdomen (Taphonomic scores 4-5; Fig. 1C).
The carcasses of marine shrimps remain intact in the presence of kaolinite until 336 hours of
degradation (Fig. 1A). With bentonite and montmorillonite, the cuticle is detached from the
cephalothorax at 240 hours of decay (Taphonomic score 4) and, for some specimens, the
cephalothorax separates from the abdomen (Taphonomic Score 5; Fig. 1B). In the presence of
kaolinite, the black film persists on the exoskeleton of marine shrimps, and the cephalothorax
only separates from the abdomen at 504 hours (Fig. 1A). Overall, in marine settings, decay is
less pronounced in the presence of kaolinite compared to other conditions (Figs. 1, 2), as the
morphology of the shrimps placed on kaolinite beds (Fig. 1A) appears to be preserved for a
longer period than in the absence of sediments or in the presence of bentonite and
montmorillonite (Fig. 1B, C). This result is consistent with previous studies that have shown a
higher fidelity of soft tissue preservation in the presence of kaolinite than in other clay minerals
(16, 17).

**Figure 1.** Examples of decay stages of marine shrimps deposited (A) on kaolinite, (B) on bentonite and
 montmorillonite, and (C) without any sediment (control). At 0 hours, the shrimps are all intact and transparent
 (Taphonomic score 0). (A) In the presence of kaolinite, the shrimps remain opaque and intact until 96 hours
 (Taphonomic score 1). At 120 hours, a black film is observed on the cuticle of the shrimps on kaolinite
 (Taphonomic score 2), and they remain well preserved until 336 hours when the cuticle is detaching from the
 thorax (Taphonomic score 3). The shrimps on kaolinite are separated into two pieces at 504 hours of decay
 (Taphonomic score 5). (B) For shrimps placed on bentonite and montmorillonite, the cuticle is detaching from the
 thorax at 168 hours (Taphonomic score 3), and the shrimps are separated into two pieces at 240 hours (Taphonomic
 score 5). (C) For marine controls, at 168 hours, the cuticle is detached from the thorax and the thorax starts to
 separate from the abdomen (Taphonomic score 4). At 240 hours, the shrimps are separated into two pieces in the
 control condition (Taphonomic score 5). Note that marine shrimps with bentonite and montmorillonite and without
 sediment are more decayed at 240 hours than marine shrimps with kaolinite at 336 hours of decay. Also,
 occasionally shrimps decaying on bentonite show a black halo around the decaying organism (B). This black halo,
 unlike under kaolinite, does not replicate the anatomy and does not limit the decay of the carcass.

**Figure 2.** Observation of the black film on a marine shrimp placed on a kaolinite bed after 504 hours of decay. (A)
 Cephalothorax, (B) carapace, (C-D) abdomen, (E) eyes, and (F) pleopods covered by the black film. Although the
 black film in the kaolinite experiments originally formed on the shrimps, after some time it also appeared in the
 surrounding matrix, likely following the diffusion of organic material from the carcass during degradation.

When comparing individual shrimp in each marine experimental condition, decay proceeded
 more uniformly for samples decaying on kaolinite (Fig. 3A) than for bentonite (Fig. 3B),
 montmorillonite (Fig. 3C), and the control (Fig. 3D). For example, all marine shrimps
 undergoing decay on the surface of kaolinite maintain the same stage of degradation for up to
 120 hours (Fig. 3A). This uniformity is not observed in the three other marine conditions, where
 individuals decay more rapidly and have more heterogeneous decay scores (Fig. 3B-D). After
 120 hours, decay rates also become more pronounced and variable in the absence of kaolinite
 (Fig. 3B-D) compared to when it is present in the experimental setup (Fig. 3A). Importantly, a
 plateau in taphonomic scores is evident in marine shrimps decaying on kaolinite starting at 120
 188 hours (Fig. 3E). This plateau is not observed in the other experimental conditions and marks
 the point in the experiment when the decay progress between the different marine conditions
 strongly departs from each other (Fig. 3E). Starting at this timepoint, the taphonomic scores of
 marine shrimps decaying on kaolinite are significantly lower than those of marine shrimps
 decaying on bentonite and without clay (Contrast analysis, $p_{Time=120[Bentonite-Kaolinite]} = 0.021$, $z-$
 $ratio_{Time=120[B-K]} = 2.882$, $p_{Time=120[Control-Kaolinite]} = 0.005$, $z-ratio_{Time=120[C-K]} = 3.289$, Tab. S2).
 Between 144 and 216 hours of decay, the taphonomic scores of marine shrimps decaying on
 kaolinite are lower than those of marine shrimps decaying in the three other experimental
 conditions (bentonite, montmorillonite, without clay) (Contrast analysis, $p_{Time=144[Bentonite-$

$Kaolinite] = 0.005$, $z\text{-ratio}_{Time=144[B-K]} = 3.295$; $p_{Time=144[Kaolinite-Montmorillonite]} = 0.004$, $z\text{-ratio}_{Time=144[K-}$
$M] = -3.380$, $p_{Time=144[Control-Kaolinite]} = 0.001$, $z\text{-ratio}_{Time=144[C-K]} = 3.741$, Tab. S2). Taphonomic
scores are not significantly different between marine shrimps decaying on bentonite,
montmorillonite, and without clay during the entire time of the experiment (Tab. S2). The start
of the statistically significant decay plateau at 120 hours for marine shrimps placed on kaolinite
(Fig. 3E) coincides with the onset of the black film formation (Figs. 1A, 2).

The black film was not observed in any of the freshwater shrimps, regardless of whether they
were decaying in the presence of kaolinite, bentonite, montmorillonite, or without sediments
(Fig. S2). In general, freshwater shrimps exhibit a faster rate of decay compared to their
saltwater counterparts (Fig. S3). Initially, at 0 hours, all freshwater shrimps appear intact and
transparent (Fig. S2). After 24 hours of decay, the cuticle becomes opaque and pink (Fig. S2).
Between 48 and 96 hours, the cuticle starts detaching from the thorax and abdomen, exposing
the gills (Fig. S2). Complete detachment of the cuticle occurs at 96-144 hours (Fig. S2). By
144-168 hours, the thorax and abdomen are split, the body anatomy is destroyed, and internal
organs are removed (Fig. S2). Generally, no significant differences are observed between
freshwater shrimps decaying on different substrates, beside a single significant difference
occurring at 48 hours (Tab. S3).

The black film (Fig. 2) was formed uniquely in the experimental condition of marine shrimp
decaying on kaolinite at 1.024 psu. The black film did not form in the single marine shrimp left
to decay on kaolinite at a slightly lower salinity (1.019 psu). Occasionally shrimps decaying on
bentonite show a black halo around the decaying organism (Fig. 1B) but this black halo does
not replicate the anatomy and does not limit the decay of the carcass unlike the black film
forming on shrimps in the presence of kaolinite. The precipitation of clay minerals on the
surface of the shrimp is likely to be dictated by complex conditions such as the limited oxygen
around a decaying carcass, salt that can act as a catalyzer, changes in pH during the decay of
the carcass, and the nature of the organic matter in question. The formation of the black film
could also be the result of complex microbial processes that might be leading to the favorable
dissolution of kaolinite and its subsequent precipitation on shrimp carcasses. This black film
has not been reported in previous publications on decay experiments.

**Figure 3.** Representation of the tissue preservation of decaying marine shrimps in each experimental condition.
 Proportion of the taphonomic score according to time (A) in the presence of kaolinite (n = 4), (B) in the presence
 of bentonite (n = 4), (C) in the presence of montmorillonite (n = 4), and (D) in the absence of sediment (n = 5).
 Taphonomic scores, quantifying the decay state, are represented by different colors. Darker blue colors indicate
 more advanced decay. (E) Taphonomic scores of marine shrimps (n = 17) decaying under different environmental
 conditions over time. The lines follow the median of the taphonomic scores of each experimental condition at each
 timepoint of the experiment. Each taphonomic score is illustrated by representations of shrimp throughout the
 experiment as described above. The black film only forms in the presence of kaolinite hence the representation of
 black shrimps in this specific experimental condition.

**Kaolinite promotes the mineralization of the cuticle of marine shrimps.** For the specimens
 on which no black film formed, the remains of the cuticle, observed with the Cryo-SEM, show
 an **organic** surface (Fig. 4A), covered by bacteria, mainly coccoids, ranging in size from 0.6 to
 1.5 μ m (Fig. 4B). For marine shrimps on which a black film formed (Fig. 4C), the cuticle is
 irregular (Fig. 4D), showing more relief than in the sample where a black film did not form
 (Fig. 4A). At a high magnification, areas of higher relief often lack clearly defined structures
 (Fig. 4E) and resemble poorly crystallized aluminosilicates (Fig. 4E) characterized by a limited
 presence of their typical sheets (Fig. 4F-H).

Interpretations of higher textural relief in BSE imagery (Fig. 4E-H) are corroborated by
 elemental analyses obtained with Energy-Dispersive Spectroscopy, highlighting cuticle
 mineralization in aluminosilicates when the black film is present (Fig. 4I). The main differences
 between the two spectra of energy dispersive elements are the large aluminum and silicon peaks
 that were observed in the areas with the black film and that are not detected in the areas with
 no black film (Fig. 4I). For the other detected elements, the peaks are similar between the two
 types of areas, although more carbon is detected in the areas with no black film probably
 because the organic carbon was replaced/covered by aluminosilicates. Oxygen is present
 everywhere even in the microscope chamber. Platinum is present in all samples because these
 were coated with this element before analyses. Sodium and chlorine also are ubiquitous because
 they are the main constituents of salt in ASW.

**Figure 4.** Cryogenic Scanning Electron Microscopy (Cryo-SEM) imaging of decaying marine shrimps, performed
 at 10keV with a backscattered detector (A, C-H) and a secondary electron detector (B). (A, B) Marine shrimp on
 which no black film formed. (A) The cuticle is dark under backscattered detectors as it is rich in carbon. (B) the
 cuticle in (A) is organic with no minerals and shows bacteria. (C-H) Marine shrimp on which a black film formed.
 The cuticle of shrimps with a black film shows minimal signs of decay (C) and is irregular (D, E) due to the
 deposition of some poorly crystallized aluminosilicates (F-H) that can very occasionally develop their typical sheet
 structures. (F) Zoom in on the framed area in E. (G) Zoom in on the framed area in F. (H) Other area with some
 aluminosilicate sheets that start to form on the cuticle. (I) Spectra of energy dispersive elements of areas with no
 black films (n = 8; pink spectrum) and an area with black films (n = 8; black spectrum). The predominant elements
 of the 16 analyzed areas are represented: carbon (C), oxygen (O), sodium (Na), magnesium (Mg), aluminum (Al),
 silicon (Si), platinum (Pt), chlorine (Cl), and calcium (Ca).

Elemental mapping reveals additional details about the black film and its associated
 mineralization (Fig. 5). In addition to the observed association between aluminum and silicon,
 some aluminum and silicon are associated with potassium in specific regions highlighting the
 existence of more than one aluminosilicate type mineralizing the cuticles (Fig. 5I-K).
 Importantly, the distribution of aluminosilicates in this film is non-uniform, as certain areas
 within the samples exhibiting the black film lack detectable amounts of aluminum, silicon, and
 potassium (Fig. 5I-K). The absence of these elements in some areas could mean that
 aluminosilicate nuclei are too small to be detected and that specific regions of the shrimp are

more conducive to mineralization than others, although no clear differences could be seen while
investigating the shrimp photographs at a macro scale.

**Figure 5.** Elemental mapping of an area with black film. (A) Cryo-SEM imaging of the cephalothorax of a marine
shrimp on which the black film formed. (B) Zoom in on the framed area in A, corresponding to the analyzed area.
(C-K) The distribution of chemical elements in the analyzed area: oxygen (O), magnesium (Mg), calcium (Ca),
platinum (Pt), sodium (Na), chlorine (Cl), aluminum (Al), silicon (Si), and potassium (K).

**Black film formation happens through the precipitation of new aluminosilicates.** These
results have significant implications for understanding the fossil record. The mineralization of
**soft structures** in aluminosilicates has been a long-debated topic. Initially, Orr et al. (1998)
suggested that the enrichment of soft tissues in aluminosilicates begins early during diagenesis,
stabilizing the tissue over geological times. This proposition was contested in subsequent
studies, which proposed that the mineralization of soft tissues in aluminosilicates likely occurs
later during maturation, thus playing a minimal role in anatomical information retention (12–
14), and that soft tissues are originally preserved as organic material, that were later replicated
by silicates under high temperatures and pressures. Based on fossils, Anderson et al. (2021)
demonstrated that this type of mineralization likely happens during early diagenesis, supporting
the theory of Orr et al. (1998). However, the exact timing, whether this process happens within
293 days, weeks, or months, could not be identified based only on fossils. **The results of this study**
**provide the first direct experimental evidence that aluminosilicates start to precipitate five days**
**after death (120 hours) in specimens left to decay on kaolinite (Figs. 1-4).**

[revised manuscript text omitted]

A corollary of this finding is that assemblages of arthropod fossils found in a single sedimentary
layer of a fossil lagerstätte could contain a mix of specimens that were freshly killed
immediately before (or during) a burial event, and those that were dead and undergoing decay
for days or weeks on the seafloor but had their morphology rapidly stabilized by the
precipitation of aluminosilicates on their carcasses (Fig. 6B, C). Distinguishing between these
two elements of any given fossil lagerstätte assemblage may not be straightforward, given the
detailed and nearly pristine structural anatomy that was stabilized by the black film during the
experiments of this study. Some degree of hidden time-averaging may have been affecting some
assemblages typically considered to be a single community snapshot where all specimens had
been killed by the burial event at the same time (Fig. 6A-C). Prolonged temporal averaging has
been proposed for the Fezouata Biota (Early Ordovician, Morocco), where consecutive animal
communities underwent visible decomposition on the seafloor before preservation (44). Some
degree of time averaging may also be affecting sites without such visible evidence of
decomposition, such as at the Walcott Quarry in the Burgess Shale (Cambrian, Canada), where
kaolinite has been identified in the sedimentary matrix (9, 45). In essence, the high preservation
potential of arthropods from the Walcott Quarry (37, 38) does not necessarily imply the
exclusive preservation of freshly killed organisms. Whenever obrution events occur (39),
preservation would involve not only freshly killed animals but also those that had previously
died and decayed, yet became stabilized through aluminosilicate precipitation (Fig. 6 A, C).
This reasoning may be projected to hundreds of Cambrian kaolinite-rich fossiliferous levels
(45) which may be capturing more temporally averaged communities than previously thought.

The results of this experiment also highlight that the association of aluminosilicates with soft
tissue preservation in sites such as the Burgess Shale is indeed the result of complex processes
involving early mineralization (9)(Fig. 6B, C). However, this mechanism is not exclusive, and
more aluminosilicates certainly precipitated during metamorphism, during which time some
weathering of previously precipitated aluminosilicates could also have occurred (12)(Fig. 6D).
Any part of the organism that has not been replicated by aluminosilicates would, in turn, be
preserved in carbon (8, 40)(Fig. 6D), which would result in a complex fossil preserved in both
carbonaceous compressions and aluminosilicates owing to their flattening over geological times
(3, 9)(Fig. 6D). As such, all these taphonomic processes are not contradictory but
complementary, and they contribute altogether to our understanding of some of the most iconic
fossil deposits recording the Cambrian Explosion.

**Figure 6.** Preservation process of decaying shrimps. (A) A marine shrimp, from a first community, dies and falls
 on a kaolinite-rich seafloor. (B) Kaolinite liberates aluminum and silicon ions which deposit on the shrimp cuticle
 stabilizing its morphology for weeks in the absence of burial. In the meantime, a marine shrimp from a second
 community dies and falls on the kaolinite bed. (C) The mineralization of the shrimp from the first community
 carries on, while the shrimp from the second community starts to mineralize. (D) Burial eventually occurs,
 followed by maturation, during which time early aluminosilicates can be altered and new aluminosilicates can
 form, resulting in an organism characterized by complex aluminosilicate phases in addition to the presence of
 carbon in regions that did not mineralize. This mechanism means that some delay in burial can occur without much
 information loss when kaolinite is present leading the mineralization of the cuticle. It could also mean that subtle
 time averaging of communities could be expected in sites where kaolinite is present. Burgess Shale-type
 preservation is the result of a complex cascade of mineral-organic matter interaction during biostratinomy, early
 diagenesis, and maturation.

CONCLUSION

The decay of freshwater and marine shrimps deposited on the surface of three different clays is
 investigated. Results show that decay proceeds more slowly in the presence of kaolinite than in
 the presence of other clay minerals in marine settings, as has been suggested previously (17,
 46). However, the current work differs from previous experiments involving sediments because
 carcasses were decaying on the surface of three clay beds rather than being completely buried.
 A novel observation is that a black film of newly formed aluminosilicates forms on shrimp
 carcasses in the presence of kaolinite shortly after death. This film was not observed in any
 previously published decay experiment. Kaolinite slows down decay by driving the
 precipitation of authigenic aluminosilicates on the cuticle of marine shrimps, which is visible
 as a black film starting at 120 hours. These results provide the first experimental evidence that
 the replication of soft tissues by aluminosilicates starts just a few days after death. This study
 also suggests that, in the presence of kaolinite and absence of burial, marine arthropods could
 retain most of their anatomical information for days, possibly weeks after death. Marine shrimp,
 and potentially any marine arthropod, may not need to be instantaneously buried for exceptional
 fossil preservation to occur. The stabilization of anatomy by aluminosilicates extends the
 duration of the potential biostratinomic window for fossil specimens described from
 lagerstätten with sediments rich in kaolinite, with the corollary that paleoecological analyses of

such sites should recognize that time-averaging of these communities may have occurred,
amalgamating specimens that died over a period of days to weeks. Taken together, these results
constitute a paradigm shift in our understanding of the processes driving exceptional
preservation in deep time, particularly during the Cambrian, which saw the appearance and
diversification of many modern animal groups.

**Acknowledgments.** The authors thank A. Mucciolo for his assistance during SEM analyses and
all members of the Anom Lab at the University of Lausanne, Switzerland, and numerous
attendees of the PalAss Annual Meeting in Cambridge, UK for fruitful discussions. NC thanks
the Master of Science in Behavior, Evolution, and Conservation of the University of Lausanne
during which this work was designed.

**Funding.** NC and FS are funded by an SNF Ambizione Grant (PZ00P2_209102). JBA is
supported by an SNF Sinergia Grant (198691) awarded to ACD and three other PIs.

**Conflict of interests.** The authors declare no competing interests.

**Data availability.** All data necessary to replicate this work are available in the main text and in
the supplementary material files.

**Author contributions.** All authors designed the research. NC and JBA did the shrimp
experiments and performed statistical analyses. All authors did the initial elemental
investigations, followed by more detailed elemental work by NC and FS. NC and FS interpreted
and discussed the results with all co-authors. NC made the figures and wrote the initial version
of the text with the help of all co-authors.

**REFERENCES**

- 1. D. E. G. Briggs, A. J. Kear, D. M. Martill, P. R. Wilby, Phosphatization of soft-tissue in
experiments and fossils. *JGS* **150**, 1035–1038 (1993).
- 2. D. E. G. Briggs, A. J. Kear, Decay and Mineralization of Shrimps. *PALAIOS* **9**, 431–456
(1994).
- 3. P. J. Orr, D. E. G. Briggs, S. L. Kearns, Cambrian Burgess Shale Animals Replicated in
Clay Minerals. *Science* **281**, 1173–1175 (1998).
- 4. R. Raiswell, K. Whaler, S. Dean, M. L. Coleman, D. E. G. Briggs, A simple three-
dimensional model of diffusion-with-precipitation applied to localised pyrite formation in
framboids, fossils and detrital iron minerals. *Marine Geology* **113**, 89–100 (1993).
- 5. P. R. Wilby, D. E. G. Briggs, B. Riou, Mineralization of soft-bodied invertebrates in a
Jurassic metalliferous deposit. *Geology* **24**, 847–850 (1996).
- 6. A. C. Daley, J. B. Antcliffe, H. B. Drage, S. Pates, Early fossil record of Euarthropoda and
the Cambrian Explosion. *Proc. Natl. Acad. Sci. U.S.A.* **115**, 5323–5331 (2018).
- 7. N. J. Butterfield, Secular distribution of Burgess-Shale-type preservation. *Lethaia* **28**, 1–
13 (1995).
- 8. R. R. Gaines, D. E. G. Briggs, Z. Yuanlong, Cambrian Burgess Shale–type deposits share
a common mode of fossilization. *Geology* **36**, 755–758 (2008).
- 9. R. P. Anderson, N. J. Tosca, E. E. Saupe, J. Wade, D. E. G. Briggs, Early formation and
taphonomic significance of kaolinite associated with Burgess Shale fossils. *Geology* **49**,
355–359 (2021).
- 10. N. J. Butterfield, Organic preservation of non-mineralizing organisms and the taphonomy
of the Burgess Shale. *Paleobiology* **16**, 272–286 (1990).
- 11. C. R. Woltz, R. P. Anderson, N. J. Tosca, S. M. Porter, The role of clay minerals in the
preservation of Precambrian organic-walled microfossils. *Geobiology* **n/a** (2023).
- 12. N. J. Butterfield, U. Balthasar, L. A. Wilson, Fossil Diagenesis in the Burgess Shale.
*Palaeontology* **50**, 537–543 (2007).
- 13. A. Page, S. E. Gabbott, P. R. Wilby, J. A. Zalasiewicz, Ubiquitous Burgess Shale–style
“clay templates” in low-grade metamorphic mudrocks. *Geol* **36**, 855 (2008).
- 14. W. Powell, Greenschist-facies metamorphism of the Burgess Shale and its implications
for models of fossil formation and preservation. *Canadian Journal of Earth Sciences* **40**,
13–25 (2003).
- 15. D. Martin, D. E. G. Briggs, R. J. Parkes, Experimental attachment of sediment particles to
invertebrate eggs and the preservation of soft-bodied fossils. *JGS* **161**, 735–738 (2004).
- 16. E. Naimark, *et al.*, Decaying in different clays: implications for soft-tissue preservation.
*Palaeontology* **59**, 583–595 (2016).

- 17. L. A. Wilson, N. J. Butterfield, Sediment Effects on the Preservation of Burgess Shale-
Type Compression Fossils. *PALAIOS* **29**, 145–154 (2014).
- 18. N. Corthésy, F. Saleh, C. Thomas, J. B. Antcliffé, A. C. Daley, The effects of clay minerals
on bacterial community composition during arthropod decay. [Preprint] (2024). Available
at: <https://www.biorxiv.org/content/10.1101/2024.02.19.580992v1> [Accessed 23
February 2024].
- 19. J. Sagemann, S. J. Bale, D. E. G. Briggs, R. J. Parkes, Controls on the formation of
authigenic minerals in association with decaying organic matter: an experimental
approach. *Geochimica et Cosmochimica Acta* **63**, 1083–1095 (1999).
- 20. A. D. Butler, J. A. Cunningham, G. E. Budd, P. C. J. Donoghue, Experimental taphonomy
of *Artemia* reveals the role of endogenous microbes in mediating decay and fossilization.
*Proceedings of the Royal Society B: Biological Sciences* **282**, 20150476 (2015).
- 21. A. D. Hancy, J. B. Antcliffé, Anoxia can increase the rate of decay for cnidarian tissue:
Using *Actinia equina* to understand the early fossil record. *Geobiology* **18**, 167–184
(2020).
- 22. D. J. Murdock, S. E. Gabbott, G. Mayer, M. A. Purnell, Decay of velvet worms
(*Onychophora*), and bias in the fossil record of lobopodians. *BMC Evolutionary Biology*
**14**, 222 (2014).
- 23. P. A. Allison, Soft-bodied animals in the fossil record: The role of decay in fragmentation
during transport. *Geol* **14**, 979 (1986).
- 24. R. S. Sansom, Experimental Decay of Soft Tissues. *Paleontol. Soc. pap.* **20**, 259–274
(2014).
- 25. R. S. Sansom, Preservation and phylogeny of Cambrian ecdysozoans tested by
experimental decay of *Priapululus*. *Sci Rep* **6**, 32817 (2016).
- 26. I. González-Castellano, J. Pons, E. González-Ortegón, A. Martínez-Lage, Mitogenome
phylogenetics in the genus *Palaemon* (Crustacea: Decapoda) sheds light on species
crypticism in the rockpool shrimp *P. elegans*. *PLOS ONE* **15**, e0237037 (2020).
- 27. A. Jabłońska, T. Mamos, P. Gruszka, A. Szlauer-Łukaszewska, M. Grabowski, First record
and DNA barcodes of the aquarium shrimp, *Neocaridina davidi*, in Central Europe from
thermally polluted River Oder canal, Poland. *Knowl. Manag. Aquat. Ecosyst.* **14** (2018).
<https://doi.org/10.1051/kmae/2018004>.
- 28. K. Ovenbeck, A. Dürr, H. Meenke, D. Brandis, C. Ewers, A shrimp between two worlds:
the genetic differentiation of the brackish water shrimp *Palaemon varians* Leach, 1813 in
the Baltic and the North Sea. *Hydrobiologia* **850**, 97–108 (2023).
- 29. J. A. F. Pantaleão, R. A. Gregati, R. C. da Costa, L. S. López-Greco, M. L. Negreiros-
Fransozo, Post-hatching development of the ornamental ‘Red Cherry Shrimp’ *Neocaridina*
*davidi* (Bouvier, 1904) (Crustacea, Caridea, Atyidae) under laboratorial conditions.
*Aquaculture Research* **48**, 553–569 (2017).

- 30. J. Park, Y. Kim, W. Kwon, H. Xi, J. Park, The complete mitochondrial genome of
Neocaridina heteropoda koreana Kubo, 1938 (Decapoda: Atyidae). *Mitochondrial DNA*
*Part B* **4**, 2332–2334 (2019).
- 31. M. Ivanić, N. Vdović, de Baretto, V. Bermanec, I. Sondi, Mineralogy, surface properties
and electrokinetic behaviour of kaolin clays from the naturally occurring pegmatite
deposits. *Geologia Croatica* **68**, 139–145 (2015).
- 32. J. Du, R. A. Pushkarova, R. St. C. Smart, A cryo-SEM study of aggregate and floc structure
changes during clay settling and raking processes. *International Journal of Mineral*
*Processing* **93**, 66–72 (2009).
- 33. S. B. Bullen, D. F. Sibley, Dolomite selectivity and mimic replacement. *Geol* **12**, 655
(1984).
- 34. C. Detellier, Functional Kaolinite. *The Chemical Record* **18**, 868–877 (2018).
- 35. D. E. G. Briggs, Extraordinary Fossils. *American Scientist* **79**, 130–141 (1991).
- 36. M. A. Purnell, *et al.*, Experimental analysis of soft-tissue fossilization: opening the black
box. *Palaeontology* **61**, 317–323 (2018).
- 37. F. Saleh, *et al.*, Taphonomic bias in exceptionally preserved biotas. *Earth and Planetary*
*Science Letters* **529**, 115873 (2020).
- 38. F. Saleh, *et al.*, A novel tool to untangle the ecology and fossil preservation knot in
exceptionally preserved biotas. *Earth and Planetary Science Letters* **569**, 117061 (2021).
- 39. R. R. Gaines, Burgess Shale-type Preservation and its Distribution in Space and Time. *The*
*Paleontological Society Papers* **20**, 123–146 (2014).
- 40. R. R. Gaines, *et al.*, Mechanism for Burgess Shale-type preservation. *Proceedings of the*
*National Academy of Sciences* **109**, 5180–5184 (2012).
- 41. F. Saleh, *et al.*, Insights into soft-part preservation from the Early Ordovician Fezouata
Biota. *Earth-Science Reviews* **213**, 103464 (2021).
- 42. F. Saleh, *et al.*, The Chengjiang Biota inhabited a deltaic environment. *Nat Commun* **13**,
1569 (2022).
- 43. F. Saleh, B. Pittet, J.-P. Perrillat, B. Lefebvre, Orbital control on exceptional fossil
preservation. *Geology* **47**, 103–106 (2019).
- 44. F. Saleh, *et al.*, Skeletal elements controlled soft-tissue preservation in echinoderms from
the Early Ordovician Fezouata Biota. *Geobios* (2023).
<https://doi.org/10.1016/j.geobios.2023.08.001>.
- 45. R. P. Anderson, N. J. Tosca, R. R. Gaines, N. Mongiardino Koch, D. E. G. Briggs, A
mineralogical signature for Burgess Shale-type fossilization. *Geology* **46**, 347–350
(2018).

46. S. McMahon, R. P. Anderson, E. E. Saupe, D. E. G. Briggs, Experimental evidence that
clay inhibits bacterial decomposers: Implications for preservation of organic fossils.
*Geology* **44**, 867–870 (2016).

**Kaolinite induces rapid authigenic mineralization of soft tissues within days**
**of marine shrimp decay**

Nora Corthésy^{1*}, Farid Saleh^{1*}, Jonathan B. Antcliff¹, and Allison C. Daley¹

¹Institute of Earth Sciences, University of Lausanne, Géopolis, CH-1015 Lausanne,
Switzerland

Corresponding authors: N. Corthésy (nora.corthesy@unil.ch)

F. Saleh (farid.nassim.saleh@gmail.com)

**ABSTRACT**

The fossil record is essential for the reconstruction of past ecosystems and their evolution,
especially fossil lagerstätten where soft tissues are preserved. Understanding fossilization
processes, including decay and mineralization, is crucial for accurately interpreting ancient
morphologies. Here, we investigate the decay of marine and freshwater shrimps deposited on
the surface of three different clay beds. In kaolinite systems, cryogenic scanning electron
microscopy shows a black film comprised of newly formed anhedral and cryptocrystalline
aluminosilicates on marine shrimp cuticles, stabilizing their overall morphology. This is the
first experimental evidence for the replication of soft tissues in aluminosilicates shortly after
death in Burgess Shale-type fossil lagerstätten recording the Cambrian Explosion. The
preservation of morphology through aluminosilicates could frequently result in carcasses
persisting on the seafloor for extended durations, meaning that instantaneous or near-
instantaneous burial may not be a prerequisite for soft tissue preservation as usually thought.
Given the extremely detailed replication of cuticular anatomy by the black film
aluminosilicates, it would be difficult to distinguish fossilized organisms that had decayed for
24 weeks from those that were freshly killed, indicating cryptic time averaging may exist in
Cambrian communities discovered in kaolinite-rich levels.

**KEYWORDS**

exceptional fossil preservation, experimental taphonomy, *Palaemon varians*, kaolinite,
silicates, authigenic mineralization

INTRODUCTION

Under typical environmental conditions, soft, non-mineralized organic tissues are rapidly
destroyed through cell autolysis, heterotrophy by micro- and macro-organisms, and denaturing
of structural elements under the changing chemical conditions of decay. Yet, some incredibly
delicate tissues survive as fossils for hundreds of millions of years. For example, the exquisite
fossils of the Burgess Shale (~508 million years), and Burgess Shale-type preservation more
broadly, record the early diversification of animals during the Cambrian Explosion (1–5). In
sites with Burgess Shale-type preservation, diverse taxa such as annelids, arthropods,
lobopodians, and mollusks are preserved with their soft tissues (6). These fossils are often found
as carbonaceous compressions (7, 8) associated with aluminosilicate minerals (3, 9). Despite
the importance of the Burgess Shale for understanding major evolutionary events, the exact
processes behind its preservation remain shrouded in mystery to some extent.

Geological processes must have interrupted biological decay to transform labile soft tissues into
enduring rock. In other words, soft tissue preservation requires structural stabilization of tissues
by permineralization or templated mineral growth to outpace decay processes such as microbial
metabolism and cell autolysis. The spatial association of soft tissues and aluminosilicate sheets
has been observed in fossils with Burgess Shale-type preservation (3, 9–14), however, the
timing of soft tissue replication in aluminosilicate sheets is controversial, with some suggesting
that it occurs during early diagenesis (3, 9–11), and others suggesting it occurs during
maturation, a much later stage of fossilization (12–14). If soft tissue preservation is truly
governed by early diagenetic interactions between organic structures and clay minerals, then
one would expect to see them associated in the earliest stages of decay. **If their association does
not develop until later, then other mechanisms must be responsible for the stabilization of
organic structures. It is also unclear whether these aluminosilicates result from the simple
attachment of pre-existing clays, such as kaolinite, in the matrix (9, 15) or from the precipitation
and formation of new authigenic minerals (9, 16).**

Experiments investigating the interaction between clay minerals and decaying carcasses help
to provide meaningful controls on the relevant conditions for soft tissue replication in
aluminosilicate sheets. Decay experiments have proven highly valuable in cataloging the loss
and retention of anatomical information under various environmental conditions, some in the
presence of sediments (16–19) and some without (20–25). Excluding sediments is a sensible
and coherent approach to understand the baseline of the processes of decay by limiting the
variables in a specific system. Recent years have seen extensive mapping of the patterns of
**decay sequences for a range of animals**, and it is now possible to examine more process-
orientated questions with more interactions and complexity introduced to the system.

Herein, deceased specimens of marine shrimp *Palaemon varians* and freshwater shrimp
*Neocaridina davidi* were placed atop beds of three different clay compositions under seawater
medium to simulate conditions at the sediment-water interface without burial. These
experiments show that the decay of marine shrimps deposited on the surface of kaolinite is
slower than that of marine shrimps deposited on bentonite and montmorillonite, or without clay.
Slower degradation is accompanied by the precipitation of newly formed aluminosilicates on
the marine shrimp carcasses, providing the first experimental evidence for this mode of
authigenic mineralization. These findings provide novel perspectives on the origin and timing
of the replication of soft tissues in aluminosilicates, suggesting that instantaneous or near-
instantaneous burial may not be a prerequisite for soft-tissue preservation as the morphology of
soft tissues remained stable/undistorted for several weeks in the presence of kaolinite clays,
without burial. This observation holds particular significance in the context of the Cambrian
Explosion, as kaolinite is frequently observed in association with soft tissues in marine rock
from the Cambrian Period.

MATERIAL & METHODS

Adult freshwater shrimps (*Neocaridina davidi*) and marine shrimps (*Palaemon varians*), that
were raised at the Institute of Earth Sciences of the University of Lausanne, were used in the
experiment. These species were chosen because they are phylogenetically constrained and well-
studied anatomically (26–30).

Adult shrimps were euthanized using clove oil [$C_7H_{12}ClN_3O_2$] which was chosen to avoid
mechanically damaging the animals and to induce their rapid death within minutes. One drop
of clove oil was added to the head of the shrimp. After the death of the shrimp, the head was
rinsed repeatedly with deionized water until no remaining oil could be seen on the shrimp or
the surface of the water when submerged. Shrimps were left to decay in sterilized acrylic boxes,
closed but not sealed, containing 5g of sediment with one of three compositions: kaolinite,
bentonite, or montmorillonite. Sediment and shrimp were covered with 35g of water: artificial
seawater (ASW, 1.024 psu) for marine shrimps and deionized water for freshwater shrimps.
The clays used have a purity between 80-90%, with the remaining 10-20% phase consisting
dominantly of quartz minerals. Considering that all shrimps were euthanized using clove oil,
using this oil cannot explain why some decay patterns occurred only in the presence of specific
clays. For the control samples, freshwater shrimps were left to decay in deionized water and
marine shrimps in ASW without introducing clays to the system. We opted to use pure ASW
and deionized water to limit the number of variables in the experiment by excluding bacteria,
plankton, and other organisms found in aquaria and natural aquatic environments in highly
variable proportions. Using pure ASW and deionized water is the simplest way to see how clays
impact the decay of freshwater and marine shrimps.

In total, twenty-three freshwater shrimps (seven replicates for each of the three clay minerals,
and two replicates for the controls) and seventeen marine shrimps (four individual replicates
for each of the three clay minerals, and five replicates for the controls) were used in this study.
All samples were kept at room temperature in the dark to avoid bias in microbial growth
following a similar protocol to Sansom (2014). For ten days, each specimen was imaged every
24 hours using a SC50 5-megapixel color camera (Olympus Life Science Solutions) with
Olympus Stream Basic software (version 2.2; Olympus Soft Imaging Solutions). Following the
ten days, specimens were allowed to continue to decay for up to 21 days, during which time
they were weekly checked for major taphonomic state changes, at which point photos were
taken.

The state of decay of each specimen was assessed every 24 hours by assigning a taphonomic
score. Seven different taphonomic scores, numbered from 0 to 6 (0 = intact, 6 = completely
degraded), were defined based on the global degradation aspect of the animal (see Tab. S1 &
Fig. S1 in *Supplementary Materials*). For each shrimp species independently, differences in
taphonomic scores through time for each clay system were assessed with an ordinal logistic
regression using the software R 4.1.1 (R Core Team, 2021). When the interaction between the
clay and time variables was significant, a contrast analysis was performed with *emmeans*
package (version 1.8-2) in R studio.

Three additional marine shrimp replicates were investigated using Cryogenic Scanning
Electron Microscopy (Cryo-SEM, -140°C) at the University of Lausanne. Two of the three
marine shrimps were left to decay in the presence of 5g of kaolinite and 35g of ASW at 1.024
123 psu in an experimental setup similar to the previous experiments on marine shrimps. The third
marine shrimp was left to decay in the presence of 5g of kaolinite and 35g of ASW at a slightly
lower salinity of 1.019 psu, to help investigate why certain patterns are only observed in marine
settings, and were not observed in freshwater conditions. The three samples were coated with a

3nm platinum layer. SEM images, sixteen elemental spectra, and an elemental map of a specific
area were acquired using Quanta FEG-250 Scanning Electron Microscope at 10 keV. For further
information regarding the choice of model organisms, the experimental design, chemical
investigations, and the statistical analyses, please refer to the *Supplementary Materials*.

**RESULTS & DISCUSSION**

**Kaolinite slows down marine shrimp decay.** Immediately following euthanasia, marine
shrimps are transparent and intact (Figs 1, S1A). At 24 hours, under all conditions, marine
shrimp become opaque and pink/white, and the eyes start to turn black (Taphonomic score 1;
Tab. S1, Fig. S1B). Over the next three days (at 96 hours of decay), minimal anatomical change
is observed in the presence of any of the three clays (Fig. 1A). However, starting at 120 hours,
a fine black film appears on the cuticle of marine shrimps on kaolinite (Taphonomic score 2;
Figs. 1A, 2), outlining the exoskeleton in exquisite detail (Figs. 1A, 2). This film is absent from
all other experimental conditions (Fig. 1B, C), which are instead covered by a white biofilm as
early as 24 hours after the start of decay (Taphonomic score 2; Figs. 1B, C, S1C, D). For marine
controls, the cuticle starts to detach from the abdomen at 96 hours (Taphonomic score 3; Fig.
1C). The cuticles of marine shrimps decaying on bentonite and montmorillonite begin to detach
from the abdomen between 120 to 168 hours into the experiment (Taphonomic score 3; Fig. 1B),
whereas, during the same period for the controls, the cuticle is completely detached from the
cephalothorax, which starts to separate from the abdomen (Taphonomic scores 4-5; Fig. 1C).
The carcasses of marine shrimps remain intact in the presence of kaolinite until 336 hours of
degradation (Fig. 1A). With bentonite and montmorillonite, the cuticle is detached from the
cephalothorax at 240 hours of decay (Taphonomic score 4) and, for some specimens, the
cephalothorax separates from the abdomen (Taphonomic Score 5; Fig. 1B). In the presence of
kaolinite, the black film persists on the exoskeleton of marine shrimps, and the cephalothorax
only separates from the abdomen at 504 hours (Fig. 1A). Overall, in marine settings, decay is
less pronounced in the presence of kaolinite compared to other conditions (Figs. 1, 2), as the
morphology of the shrimps placed on kaolinite beds (Fig. 1A) appears to be preserved for a
longer period than in the absence of sediments or in the presence of bentonite and
montmorillonite (Fig. 1B, C). This result is consistent with previous studies that have shown a
higher fidelity of soft tissue preservation in the presence of kaolinite than in other clay minerals
(16, 17).

**Figure 1.** Examples of decay stages of marine shrimps deposited (A) on kaolinite, (B) on bentonite and
 montmorillonite, and (C) without any sediment (control). At 0 hours, the shrimps are all intact and transparent
 (Taphonomic score 0). (A) In the presence of kaolinite, the shrimps remain opaque and intact until 96 hours
 (Taphonomic score 1). At 120 hours, a black film is observed on the cuticle of the shrimps on kaolinite
 (Taphonomic score 2), and they remain well preserved until 336 hours when the cuticle is detaching from the
 thorax (Taphonomic score 3). The shrimps on kaolinite are separated into two pieces at 504 hours of decay
 (Taphonomic score 5). (B) For shrimps placed on bentonite and montmorillonite, the cuticle is detaching from the
 thorax at 168 hours (Taphonomic score 3), and the shrimps are separated into two pieces at 240 hours (Taphonomic
 score 5). (C) For marine controls, at 168 hours, the cuticle is detached from the thorax and the thorax starts to
 separate from the abdomen (Taphonomic score 4). At 240 hours, the shrimps are separated into two pieces in the
 control condition (Taphonomic score 5). Note that marine shrimps with bentonite and montmorillonite and without
 sediment are more decayed at 240 hours than marine shrimps with kaolinite at 336 hours of decay. Also,
 occasionally shrimps decaying on bentonite show a black halo around the decaying organism (B). This black halo,
 unlike under kaolinite, does not replicate the anatomy and does not limit the decay of the carcass.

**Figure 2.** Observation of the black film on a marine shrimp placed on a kaolinite bed after 504 hours of decay. (A)
 Cephalothorax, (B) carapace, (C-D) abdomen, (E) eyes, and (F) pleopods covered by the black film. Although the
 black film in the kaolinite experiments originally formed on the shrimps, after some time it also appeared in the
 surrounding matrix, likely following the diffusion of organic material from the carcass during degradation.

When comparing individual shrimp in each marine experimental condition, decay proceeded
 more uniformly for samples decaying on kaolinite (Fig. 3A) than for bentonite (Fig. 3B),
 montmorillonite (Fig. 3C), and the control (Fig. 3D). For example, all marine shrimps
 undergoing decay on the surface of kaolinite maintain the same stage of degradation for up to
 120 hours (Fig. 3A). This uniformity is not observed in the three other marine conditions, where
 individuals decay more rapidly and have more heterogeneous decay scores (Fig. 3B-D). After
 120 hours, decay rates also become more pronounced and variable in the absence of kaolinite
 (Fig. 3B-D) compared to when it is present in the experimental setup (Fig. 3A). Importantly, a
 plateau in taphonomic scores is evident in marine shrimps decaying on kaolinite starting at 120
 188 hours (Fig. 3E). This plateau is not observed in the other experimental conditions and marks
 the point in the experiment when the decay progress between the different marine conditions
 strongly departs from each other (Fig. 3E). Starting at this timepoint, the taphonomic scores of
 marine shrimps decaying on kaolinite are significantly lower than those of marine shrimps
 decaying on bentonite and without clay (Contrast analysis, $p_{Time=120[Bentonite-Kaolinite]} = 0.021$, $z-$
 $ratio_{Time=120[B-K]} = 2.882$, $p_{Time=120[Control-Kaolinite]} = 0.005$, $z-ratio_{Time=120[C-K]} = 3.289$, Tab. S2).
 Between 144 and 216 hours of decay, the taphonomic scores of marine shrimps decaying on
 kaolinite are lower than those of marine shrimps decaying in the three other experimental
 conditions (bentonite, montmorillonite, without clay) (Contrast analysis, $p_{Time=144[Bentonite-$

$Kaolinite] = 0.005$, $z\text{-ratio}_{Time=144[B-K]} = 3.295$; $p_{Time=144[Kaolinite-Montmorillonite]} = 0.004$, $z\text{-ratio}_{Time=144[K-}$
$M] = -3.380$, $p_{Time=144[Control-Kaolinite]} = 0.001$, $z\text{-ratio}_{Time=144[C-K]} = 3.741$, Tab. S2). Taphonomic
scores are not significantly different between marine shrimps decaying on bentonite,
montmorillonite, and without clay during the entire time of the experiment (Tab. S2). The start
of the statistically significant decay plateau at 120 hours for marine shrimps placed on kaolinite
(Fig. 3E) coincides with the onset of the black film formation (Figs. 1A, 2).

The black film was not observed in any of the freshwater shrimps, regardless of whether they
were decaying in the presence of kaolinite, bentonite, montmorillonite, or without sediments
(Fig. S2). In general, freshwater shrimps exhibit a faster rate of decay compared to their
saltwater counterparts (Fig. S3). Initially, at 0 hours, all freshwater shrimps appear intact and
transparent (Fig. S2). After 24 hours of decay, the cuticle becomes opaque and pink (Fig. S2).
Between 48 and 96 hours, the cuticle starts detaching from the thorax and abdomen, exposing
the gills (Fig. S2). Complete detachment of the cuticle occurs at 96-144 hours (Fig. S2). By
144-168 hours, the thorax and abdomen are split, the body anatomy is destroyed, and internal
organs are removed (Fig. S2). Generally, no significant differences are observed between
freshwater shrimps decaying on different substrates, beside a single significant difference
occurring at 48 hours (Tab. S3).

The black film (Fig. 2) was formed uniquely in the experimental condition of marine shrimp
decaying on kaolinite at 1.024 psu. The black film did not form in the single marine shrimp left
to decay on kaolinite at a slightly lower salinity (1.019 psu). Occasionally shrimps decaying on
bentonite show a black halo around the decaying organism (Fig. 1B) but this black halo does
not replicate the anatomy and does not limit the decay of the carcass unlike the black film
forming on shrimps in the presence of kaolinite. The precipitation of clay minerals on the
surface of the shrimp is likely to be dictated by complex conditions such as the limited oxygen
around a decaying carcass, salt that can act as a catalyzer, changes in pH during the decay of
the carcass, and the nature of the organic matter in question. The formation of the black film
could also be the result of complex microbial processes that might be leading to the favorable
dissolution of kaolinite and its subsequent precipitation on shrimp carcasses. This black film
has not been reported in previous publications on decay experiments.

**Figure 3.** Representation of the tissue preservation of decaying marine shrimps in each experimental condition.
 Proportion of the taphonomic score according to time (A) in the presence of kaolinite (n = 4), (B) in the presence
 of bentonite (n = 4), (C) in the presence of montmorillonite (n = 4), and (D) in the absence of sediment (n = 5).
 Taphonomic scores, quantifying the decay state, are represented by different colors. Darker blue colors indicate
 more advanced decay. (E) Taphonomic scores of marine shrimps (n = 17) decaying under different environmental
 conditions over time. The lines follow the median of the taphonomic scores of each experimental condition at each
 timepoint of the experiment. Each taphonomic score is illustrated by representations of shrimp throughout the
 experiment as described above. The black film only forms in the presence of kaolinite hence the representation of
 black shrimps in this specific experimental condition.

**Kaolinite promotes the mineralization of the cuticle of marine shrimps.** For the specimens
 on which no black film formed, the remains of the cuticle, observed with the Cryo-SEM, show
 an organic surface (Fig. 4A), covered by bacteria, mainly coccoids, ranging in size from 0.6 to
 1.5µm (Fig. 4B). For marine shrimps on which a black film formed (Fig. 4C), the cuticle is
 irregular (Fig. 4D), showing more relief than in the sample where a black film did not form
 (Fig. 4A). At a high magnification, areas of higher relief often lack clearly defined structures
 (Fig. 4E) and resemble poorly crystallized aluminosilicates (Fig. 4E) characterized by a limited
 presence of their typical sheets (Fig. 4F-H).

Interpretations of higher textural relief in BSE imagery (Fig. 4E-H) are corroborated by
 elemental analyses obtained with Energy-Dispersive Spectroscopy, highlighting cuticle
 mineralization in aluminosilicates when the black film is present (Fig. 4I). The main differences
 between the two spectra of energy dispersive elements are the large aluminum and silicon peaks
 that were observed in the areas with the black film and that are not detected in the areas with
 no black film (Fig. 4I). For the other detected elements, the peaks are similar between the two
 types of areas, although more carbon is detected in the areas with no black film probably
 because the organic carbon was replaced/covered by aluminosilicates. Oxygen is present
 everywhere even in the microscope chamber. Platinum is present in all samples because these
 were coated with this element before analyses. Sodium and chlorine also are ubiquitous because
 they are the main constituents of salt in ASW.

**Figure 4.** Cryogenic Scanning Electron Microscopy (Cryo-SEM) imaging of decaying marine shrimps, performed
 at 10keV with a backscattered detector (A, C-H) and a secondary electron detector (B). (A, B) Marine shrimp on
 which no black film formed. (A) The cuticle is dark under backscattered detectors as it is rich in carbon. (B) the
 cuticle in (A) is organic with no minerals and shows bacteria. (C-H) Marine shrimp on which a black film formed.
 The cuticle of shrimps with a black film shows minimal signs of decay (C) and is irregular (D, E) due to the
 deposition of some poorly crystallized aluminosilicates (F-H) that can very occasionally develop their typical sheet
 structures. (F) Zoom in on the framed area in E. (G) Zoom in on the framed area in F. (H) Other area with some
 aluminosilicate sheets that start to form on the cuticle. (I) Spectra of energy dispersive elements of areas with no
 black films (n = 8; pink spectrum) and an area with black films (n = 8; black spectrum). The predominant elements
 of the 16 analyzed areas are represented: carbon (C), oxygen (O), sodium (Na), magnesium (Mg), aluminum (Al),
 silicon (Si), platinum (Pt), chlorine (Cl), and calcium (Ca).

Elemental mapping reveals additional details about the black film and its associated
 mineralization (Fig. 5). In addition to the observed association between aluminum and silicon,
 some aluminum and silicon are associated with potassium in specific regions highlighting the
 existence of more than one aluminosilicate type mineralizing the cuticles (Fig. 5I-K).
 Importantly, the distribution of aluminosilicates in this film is non-uniform, as certain areas
 within the samples exhibiting the black film lack detectable amounts of aluminum, silicon, and
 potassium (Fig. 5I-K). The absence of these elements in some areas could mean that
 aluminosilicate nuclei are too small to be detected and that specific regions of the shrimp are

more conducive to mineralization than others, although no clear differences could be seen while
investigating the shrimp photographs at a macro scale.

**Figure 5.** Elemental mapping of an area with black film. (A) Cryo-SEM imaging of the cephalothorax of a marine
shrimp on which the black film formed. (B) Zoom in on the framed area in A, corresponding to the analyzed area.
(C-K) The distribution of chemical elements in the analyzed area: oxygen (O), magnesium (Mg), calcium (Ca),
platinum (Pt), sodium (Na), chlorine (Cl), aluminum (Al), silicon (Si), and potassium (K).

**Black film formation happens through the precipitation of new aluminosilicates.** These
results have significant implications for understanding the fossil record. The mineralization of
soft structures in aluminosilicates has been a long-debated topic. Initially, Orr et al. (1998)
suggested that the enrichment of soft tissues in aluminosilicates begins early during diagenesis,
stabilizing the tissue over geological times. This proposition was contested in subsequent
studies, which proposed that the mineralization of soft tissues in aluminosilicates likely occurs
later during maturation, thus playing a minimal role in anatomical information retention (12–
14), and that soft tissues are originally preserved as organic material, that were later replicated
by silicates under high temperatures and pressures. Based on fossils, Anderson et al. (2021)
demonstrated that this type of mineralization likely happens during early diagenesis, supporting
the theory of Orr et al. (1998). However, the exact timing, whether this process happens within
293 days, weeks, or months, could not be identified based only on fossils. The results of this study
provide the first direct experimental evidence that aluminosilicates start to precipitate five days
after death (120 hours) in specimens left to decay on kaolinite (Figs. 1-4).

[revised manuscript text omitted]

A corollary of this finding is that assemblages of arthropod fossils found in a single sedimentary
layer of a fossil lagerstätte could contain a mix of specimens that were freshly killed
immediately before (or during) a burial event, and those that were dead and undergoing decay
for days or weeks on the seafloor but had their morphology rapidly stabilized by the
precipitation of aluminosilicates on their carcasses (Fig. 6B, C). Distinguishing between these
two elements of any given fossil lagerstätte assemblage may not be straightforward, given the
detailed and nearly pristine structural anatomy that was stabilized by the black film during the
experiments of this study. Some degree of hidden time-averaging may have been affecting some
assemblages typically considered to be a single community snapshot where all specimens had
been killed by the burial event at the same time (Fig. 6A-C). Prolonged temporal averaging has
been proposed for the Fezouata Biota (Early Ordovician, Morocco), where consecutive animal
communities underwent visible decomposition on the seafloor before preservation (44). Some
degree of time averaging may also be affecting sites without such visible evidence of
decomposition, such as at the Walcott Quarry in the Burgess Shale (Cambrian, Canada), where
kaolinite has been identified in the sedimentary matrix (9, 45). In essence, the high preservation
potential of arthropods from the Walcott Quarry (37, 38) does not necessarily imply the
exclusive preservation of freshly killed organisms. Whenever obrution events occur (39),
preservation would involve not only freshly killed animals but also those that had previously
died and decayed, yet became stabilized through aluminosilicate precipitation (Fig. 6 A, C).
This reasoning may be projected to hundreds of Cambrian kaolinite-rich fossiliferous levels
(45) which may be capturing more temporally averaged communities than previously thought.

The results of this experiment also highlight that the association of aluminosilicates with soft
tissue preservation in sites such as the Burgess Shale is indeed the result of complex processes
involving early mineralization (9)(Fig. 6B, C). However, this mechanism is not exclusive, and
more aluminosilicates certainly precipitated during metamorphism, during which time some
weathering of previously precipitated aluminosilicates could also have occurred (12)(Fig. 6D).
Any part of the organism that has not been replicated by aluminosilicates would, in turn, be
preserved in carbon (8, 40)(Fig. 6D), which would result in a complex fossil preserved in both
carbonaceous compressions and aluminosilicates owing to their flattening over geological times
(3, 9)(Fig. 6D). As such, all these taphonomic processes are not contradictory but
complementary, and they contribute altogether to our understanding of some of the most iconic
fossil deposits recording the Cambrian Explosion.

**Figure 6.** Preservation process of decaying shrimps. (A) A marine shrimp, from a first community, dies and falls
 on a kaolinite-rich seafloor. (B) Kaolinite liberates aluminum and silicon ions which deposit on the shrimp cuticle
 stabilizing its morphology for weeks in the absence of burial. In the meantime, a marine shrimp from a second
 community dies and falls on the kaolinite bed. (C) The mineralization of the shrimp from the first community
 carries on, while the shrimp from the second community starts to mineralize. (D) Burial eventually occurs,
 followed by maturation, during which time early aluminosilicates can be altered and new aluminosilicates can
 form, resulting in an organism characterized by complex aluminosilicate phases in addition to the presence of
 carbon in regions that did not mineralize. This mechanism means that some delay in burial can occur without much
 information loss when kaolinite is present leading the mineralization of the cuticle. It could also mean that subtle
 time averaging of communities could be expected in sites where kaolinite is present. Burgess Shale-type
 preservation is the result of a complex cascade of mineral-organic matter interaction during biostratinomy, early
 diagenesis, and maturation.

CONCLUSION

The decay of freshwater and marine shrimps deposited on the surface of three different clays is
 investigated. Results show that decay proceeds more slowly in the presence of kaolinite than in
 the presence of other clay minerals in marine settings, as has been suggested previously (17,
 46). However, the current work differs from previous experiments involving sediments because
 carcasses were decaying on the surface of three clay beds rather than being completely buried.
 A novel observation is that a black film of newly formed aluminosilicates forms on shrimp
 carcasses in the presence of kaolinite shortly after death. This film was not observed in any
 previously published decay experiment. Kaolinite slows down decay by driving the
 precipitation of authigenic aluminosilicates on the cuticle of marine shrimps, which is visible
 as a black film starting at 120 hours. These results provide the first experimental evidence that
 the replication of soft tissues by aluminosilicates starts just a few days after death. This study
 also suggests that, in the presence of kaolinite and absence of burial, marine arthropods could
 retain most of their anatomical information for days, possibly weeks after death. Marine shrimp,
 and potentially any marine arthropod, may not need to be instantaneously buried for exceptional
 fossil preservation to occur. The stabilization of anatomy by aluminosilicates extends the
 duration of the potential biostratinomic window for fossil specimens described from
 lagerstätten with sediments rich in kaolinite, with the corollary that paleoecological analyses of

such sites should recognize that time-averaging of these communities may have occurred,
amalgamating specimens that died over a period of days to weeks. Taken together, these results
constitute a paradigm shift in our understanding of the processes driving exceptional
preservation in deep time, particularly during the Cambrian, which saw the appearance and
diversification of many modern animal groups.

**Acknowledgments.** The authors thank A. Mucciolo for his assistance during SEM analyses and
all members of the Anom Lab at the University of Lausanne, Switzerland, and numerous
attendees of the PalAss Annual Meeting in Cambridge, UK for fruitful discussions. NC thanks
the Master of Science in Behavior, Evolution, and Conservation of the University of Lausanne
during which this work was designed.

**Funding.** NC and FS are funded by an SNF Ambizione Grant (PZ00P2_209102). JBA is
supported by an SNF Sinergia Grant (198691) awarded to ACD and three other PIs.

**Conflict of interests.** The authors declare no competing interests.

**Data availability.** All data necessary to replicate this work are available in the main text and in
the supplementary material files.

**Author contributions.** All authors designed the research. NC and JBA did the shrimp
experiments and performed statistical analyses. All authors did the initial elemental
investigations, followed by more detailed elemental work by NC and FS. NC and FS interpreted
and discussed the results with all co-authors. NC made the figures and wrote the initial version
of the text with the help of all co-authors.

**REFERENCES**

- 1. D. E. G. Briggs, A. J. Kear, D. M. Martill, P. R. Wilby, Phosphatization of soft-tissue in
experiments and fossils. *JGS* **150**, 1035–1038 (1993).
- 2. D. E. G. Briggs, A. J. Kear, Decay and Mineralization of Shrimps. *PALAIOS* **9**, 431–456
(1994).
- 3. P. J. Orr, D. E. G. Briggs, S. L. Kearns, Cambrian Burgess Shale Animals Replicated in
Clay Minerals. *Science* **281**, 1173–1175 (1998).
- 4. R. Raiswell, K. Whaler, S. Dean, M. L. Coleman, D. E. G. Briggs, A simple three-
dimensional model of diffusion-with-precipitation applied to localised pyrite formation in
framboids, fossils and detrital iron minerals. *Marine Geology* **113**, 89–100 (1993).
- 5. P. R. Wilby, D. E. G. Briggs, B. Riou, Mineralization of soft-bodied invertebrates in a
Jurassic metalliferous deposit. *Geology* **24**, 847–850 (1996).
- 6. A. C. Daley, J. B. Antcliffe, H. B. Drage, S. Pates, Early fossil record of Euarthropoda and
the Cambrian Explosion. *Proc. Natl. Acad. Sci. U.S.A.* **115**, 5323–5331 (2018).
- 7. N. J. Butterfield, Secular distribution of Burgess-Shale-type preservation. *Lethaia* **28**, 1–
13 (1995).
- 8. R. R. Gaines, D. E. G. Briggs, Z. Yuanlong, Cambrian Burgess Shale–type deposits share
a common mode of fossilization. *Geology* **36**, 755–758 (2008).
- 9. R. P. Anderson, N. J. Tosca, E. E. Saupe, J. Wade, D. E. G. Briggs, Early formation and
taphonomic significance of kaolinite associated with Burgess Shale fossils. *Geology* **49**,
355–359 (2021).
- 10. N. J. Butterfield, Organic preservation of non-mineralizing organisms and the taphonomy
of the Burgess Shale. *Paleobiology* **16**, 272–286 (1990).
- 11. C. R. Woltz, R. P. Anderson, N. J. Tosca, S. M. Porter, The role of clay minerals in the
preservation of Precambrian organic-walled microfossils. *Geobiology* **n/a** (2023).
- 12. N. J. Butterfield, U. Balthasar, L. A. Wilson, Fossil Diagenesis in the Burgess Shale.
*Palaeontology* **50**, 537–543 (2007).
- 13. A. Page, S. E. Gabbott, P. R. Wilby, J. A. Zalasiewicz, Ubiquitous Burgess Shale–style
“clay templates” in low-grade metamorphic mudrocks. *Geol* **36**, 855 (2008).
- 14. W. Powell, Greenschist-facies metamorphism of the Burgess Shale and its implications
for models of fossil formation and preservation. *Canadian Journal of Earth Sciences* **40**,
13–25 (2003).
- 15. D. Martin, D. E. G. Briggs, R. J. Parkes, Experimental attachment of sediment particles to
invertebrate eggs and the preservation of soft-bodied fossils. *JGS* **161**, 735–738 (2004).
- 16. E. Naimark, *et al.*, Decaying in different clays: implications for soft-tissue preservation.
*Palaeontology* **59**, 583–595 (2016).

- 17. L. A. Wilson, N. J. Butterfield, Sediment Effects on the Preservation of Burgess Shale-
Type Compression Fossils. *PALAIOS* **29**, 145–154 (2014).
- 18. N. Corthésy, F. Saleh, C. Thomas, J. B. Antcliffé, A. C. Daley, The effects of clay minerals
on bacterial community composition during arthropod decay. [Preprint] (2024). Available
at: <https://www.biorxiv.org/content/10.1101/2024.02.19.580992v1> [Accessed 23
February 2024].
- 19. J. Sagemann, S. J. Bale, D. E. G. Briggs, R. J. Parkes, Controls on the formation of
authigenic minerals in association with decaying organic matter: an experimental
approach. *Geochimica et Cosmochimica Acta* **63**, 1083–1095 (1999).
- 20. A. D. Butler, J. A. Cunningham, G. E. Budd, P. C. J. Donoghue, Experimental taphonomy
of *Artemia* reveals the role of endogenous microbes in mediating decay and fossilization.
*Proceedings of the Royal Society B: Biological Sciences* **282**, 20150476 (2015).
- 21. A. D. Hancy, J. B. Antcliffé, Anoxia can increase the rate of decay for cnidarian tissue:
Using *Actinia equina* to understand the early fossil record. *Geobiology* **18**, 167–184
(2020).
- 22. D. J. Murdock, S. E. Gabbott, G. Mayer, M. A. Purnell, Decay of velvet worms
(*Onychophora*), and bias in the fossil record of lobopodians. *BMC Evolutionary Biology*
**14**, 222 (2014).
- 23. P. A. Allison, Soft-bodied animals in the fossil record: The role of decay in fragmentation
during transport. *Geol* **14**, 979 (1986).
- 24. R. S. Sansom, Experimental Decay of Soft Tissues. *Paleontol. Soc. pap.* **20**, 259–274
(2014).
- 25. R. S. Sansom, Preservation and phylogeny of Cambrian ecdysozoans tested by
experimental decay of *Priapulid*. *Sci Rep* **6**, 32817 (2016).
- 26. I. González-Castellano, J. Pons, E. González-Ortegón, A. Martínez-Lage, Mitogenome
phylogenetics in the genus *Palaemon* (Crustacea: Decapoda) sheds light on species
crypticism in the rockpool shrimp *P. elegans*. *PLOS ONE* **15**, e0237037 (2020).
- 27. A. Jabłońska, T. Mamos, P. Gruszka, A. Szlauer-Łukaszewska, M. Grabowski, First record
and DNA barcodes of the aquarium shrimp, *Neocaridina davidi*, in Central Europe from
thermally polluted River Oder canal, Poland. *Knowl. Manag. Aquat. Ecosyst.* **14** (2018).
<https://doi.org/10.1051/kmae/2018004>.
- 28. K. Ovenbeck, A. Dürr, H. Meenke, D. Brandis, C. Ewers, A shrimp between two worlds:
the genetic differentiation of the brackish water shrimp *Palaemon varians* Leach, 1813 in
the Baltic and the North Sea. *Hydrobiologia* **850**, 97–108 (2023).
- 29. J. A. F. Pantaleão, R. A. Gregati, R. C. da Costa, L. S. López-Greco, M. L. Negreiros-
Fransozo, Post-hatching development of the ornamental ‘Red Cherry Shrimp’ *Neocaridina*
*davidi* (Bouvier, 1904) (Crustacea, Caridea, Atyidae) under laboratorial conditions.
*Aquaculture Research* **48**, 553–569 (2017).

- 30. J. Park, Y. Kim, W. Kwon, H. Xi, J. Park, The complete mitochondrial genome of
Neocaridina heteropoda koreana Kubo, 1938 (Decapoda: Atyidae). *Mitochondrial DNA*
*Part B* **4**, 2332–2334 (2019).
- 31. M. Ivanić, N. Vdović, de Baretto, V. Bermanec, I. Sondi, Mineralogy, surface properties
and electrokinetic behaviour of kaolin clays from the naturally occurring pegmatite
deposits. *Geologia Croatica* **68**, 139–145 (2015).
- 32. J. Du, R. A. Pushkarova, R. St. C. Smart, A cryo-SEM study of aggregate and floc structure
changes during clay settling and raking processes. *International Journal of Mineral*
*Processing* **93**, 66–72 (2009).
- 33. S. B. Bullen, D. F. Sibley, Dolomite selectivity and mimic replacement. *Geol* **12**, 655
(1984).
- 34. C. Detellier, Functional Kaolinite. *The Chemical Record* **18**, 868–877 (2018).
- 35. D. E. G. Briggs, Extraordinary Fossils. *American Scientist* **79**, 130–141 (1991).
- 36. M. A. Purnell, *et al.*, Experimental analysis of soft-tissue fossilization: opening the black
box. *Palaeontology* **61**, 317–323 (2018).
- 37. F. Saleh, *et al.*, Taphonomic bias in exceptionally preserved biotas. *Earth and Planetary*
*Science Letters* **529**, 115873 (2020).
- 38. F. Saleh, *et al.*, A novel tool to untangle the ecology and fossil preservation knot in
exceptionally preserved biotas. *Earth and Planetary Science Letters* **569**, 117061 (2021).
- 39. R. R. Gaines, Burgess Shale-type Preservation and its Distribution in Space and Time. *The*
*Paleontological Society Papers* **20**, 123–146 (2014).
- 40. R. R. Gaines, *et al.*, Mechanism for Burgess Shale-type preservation. *Proceedings of the*
*National Academy of Sciences* **109**, 5180–5184 (2012).
- 41. F. Saleh, *et al.*, Insights into soft-part preservation from the Early Ordovician Fezouata
Biota. *Earth-Science Reviews* **213**, 103464 (2021).
- 42. F. Saleh, *et al.*, The Chengjiang Biota inhabited a deltaic environment. *Nat Commun* **13**,
1569 (2022).
- 43. F. Saleh, B. Pittet, J.-P. Perrillat, B. Lefebvre, Orbital control on exceptional fossil
preservation. *Geology* **47**, 103–106 (2019).
- 44. F. Saleh, *et al.*, Skeletal elements controlled soft-tissue preservation in echinoderms from
the Early Ordovician Fezouata Biota. *Geobios* (2023).
<https://doi.org/10.1016/j.geobios.2023.08.001>.
- 45. R. P. Anderson, N. J. Tosca, R. R. Gaines, N. Mongiardino Koch, D. E. G. Briggs, A
mineralogical signature for Burgess Shale-type fossilization. *Geology* **46**, 347–350
(2018).

46. S. McMahon, R. P. Anderson, E. E. Saupe, D. E. G. Briggs, Experimental evidence that
clay inhibits bacterial decomposers: Implications for preservation of organic fossils.
*Geology* **44**, 867–870 (2016).

First, we would like to thank you for the decision to revise the paper. We have carefully addressed all editorial and reviewer comments as highlighted in this file. A tracked change version of the manuscript is provided in addition to the requested checklists. The current version of the text adheres to the formatting requirements of Communications Earth and Environment.

1] Please fully justify the definition and use of 'soft tissue' as a descriptor of the studied exoskeletons, in response to the criticisms raised by reviewer #1.

We agree with Reviewer 1 that a shrimp cuticle is not exactly a soft tissue, however, animals having the same body constituents as shrimps are not found in all fossiliferous sites, and are almost solely found in Lagerstätten. In other words, this is simply a terminology change, that does not alter the scope of the paper. To avoid confusion, whenever necessary we replaced "soft tissues" with "lightly biomineralized structures" or other descriptive terms starting from the title of the paper which currently reads: "Kaolinite induces rapid authigenic mineralisation in unburied shrimps".

2] The revision should include a more extended presentation and discussion of the impact of the results on considerations of time averaging, including a more thorough referencing of previous work and explanation of how this current work demonstrates new findings.

Per the comments of both Reviewers 1 and 2, we have re-shaped the time-averaging discussion, and we have included the mentioned references in their review. More details on this point are provided in the point-by-point answer to their comments below, and the following text was added to the current version of the manuscript: "This finding has broader implications for understanding community preservation in Konservat-Lagerstätten recording the Cambrian Explosion. In these sites, a separation is usually made between carcasses decaying on the seafloor, termed "time-averaged assemblage," and those killed during the burial event, termed "census assemblage"^{51,52}. Our experimental observations suggest that census assemblages might represent previously dead communities that were stabilized by aluminosilicates instead of freshly killed ones. This process might be operational in hundreds of Cambrian fossiliferous sedimentary layers where kaolinite has been identified in the sedimentary matrix²⁸, and the high preservation potential of arthropods in these sites does not necessarily imply the exclusive preservation of freshly killed organisms."

3] The specific requests concerning improvements in the description of methods (e.g., Reviewer 3 comments) should be met.

We have now included all the additional information requested by Reviewer 3 in the main manuscript text and the supplementary material. More details were added to the Method section, especially about the experimental setup (i.e., preparation of the experimental boxes, dimensions of boxes and shrimps) and the quantification of the decay: "Shrimps were individually left to decay in sterilized acrylic boxes (5 x 3 x 2 cm), closed but not sealed, containing 5g of sediment with one of three compositions: kaolinite, bentonite, or montmorillonite. The clays used have a purity between 80-90%, with the remaining 10-20% phase consisting dominantly of quartz minerals. Each shrimp deposited on the surface of the sediment was covered with 35g of water: reverse osmosis deionized water for freshwater shrimps and artificial seawater (ASW) for marine shrimps, prepared to 1.024 psu with reverse osmosis deionized water and Aquarium Systems Reef Crystals."

The attribution of the taphonomic scores for each organism was described in more detail, so that the methods are reproducible: "Three different taphonomic scores, numbered from 0 to 6 (0 = intact, 6 = completely degraded), were defined based on the degradation aspect of the cephalothoracic carapace, the appendages and the eyes of the animals in this study (see Tab. S1 & Fig. S1 in Supplementary Materials). To determine the global taphonomic score of each individual, the median of the carapace, appendages and eyes taphonomic scores was calculated (see Tab. S1 & Fig. S1 in Supplementary Materials). For score 0, the shrimp is intact and transparent. Score 1 corresponds to the carapace and appendages becoming pink and opaque, and the eyes turning black. When biofilms and/or other coatings are observed on the decaying carcass, score 2 is attributed. Score 3 is observed when the carapace detaches from the cephalothorax, at least 50% of the appendages are disarticulated, and the eyes detach from the cephalothorax. Score 4 is attributed when the carapace is completely detached from the cephalothorax so the gills and internal organs are exposed, at least 50% of the appendages are detached, and the eyes are detached. Score 5 is assigned when the cephalothorax and carapace are separated from the abdomen, at least 50% of the appendages are broken into several small pieces, and the eyes show advanced deterioration. Finally, for score 6, the individual is completely degraded."

In addition, please check and confirm that you have adhered to all required ethical oversight regarding the use of these shrimp and include a statement to that effect in the Acknowledgements, even if it is just to say that no ethical approval was required.

We have indeed adhered to all required ethical elements regarding shrimp experiments in Switzerland, where the experiments were made. The following text was added to the acknowledgements: "No ethical approval was required to do these experiments, which were funded by the Swiss National Science Foundation. The taxa used in these experiments are not protected in Switzerland. Individuals were euthanised ethically, and the experiments adhered to the 3Rs principle of research (reduce, reuse and recycle)."

Reviewer #1 (Remarks to the Author):

This manuscript deals with the results of a taphonomic experiment where the carcasses of two species of caridean shrimp, one marine, one fresh water, were left to decay on substrates composed of various clay minerals. The results of these experiments can be summarized as follows:

- Carcasses associated with a kaolinite substrate decayed more slowly than on other types of clay; while interesting, this finding is not new (Wilson & Butterfield 2014; McMahan et al. 2016; Naimark et al. 2016), as per the authors own admission.

Indeed one of the first observations of the paper is that kaolinite slows down decay. We agree with Reviewer 1 that this idea is not new. However, our experimental setup, which involves unburied carcasses is very dissimilar from other previously made experiments, and the implications of our work are very different from previous studies. This is now clearer in the text: "This result is consistent with previous studies that suggest that kaolinite slows down decay. Earlier work showed that kaolinite can limit bacterial growth and reduce bacterial diversity surrounding a carcass, thus, limiting its degradation and recycling^{34,35}. It has also been shown that carcasses buried in kaolinite decay slower than when buried in other minerals^{16,17,36}. The implication of our study is different from previous works because it involves unburied shrimp carcasses deposited on the surface of three mineralogical beds."

- In marine shrimp decomposing in artificial seawater, growth of likely authigenic aluminosilicates occurs on the exoskeletal parts of the carcass. This finding is of significant interest regarding the debate on whether aluminosilicates in Burgess Shale-type preservation are early diagenetic, and aided in preservation, or are merely a late diagenetic feature, and hence of no preservational consequence. The current results suggest that the former is the case.

This is a summary of the direct implications of the study.

- No aluminosilicates form on carcasses in fresh water, or under reduced salinities. This, again, is not quite unexpected, as the role of salinity in the formation and its influence on the type of authigenic clays that are formed is well established (see e.g. Calvo et al. 1995).

The observation that no aluminosilicates form on carcasses in fresh water and under reduced salinities is not the main story in the manuscript. However, it is mentioned because it is an interesting observation. We are aware that we are not the first researchers to study clay precipitation. However, it is unclear to us how the cited reference by Reviewer 1, which focuses on continental evaporitic rocks and Mg-rich clays is directly relevant to the current discussion on processes that might be happening in siliciclastic marine systems with Al-rich clays. However, we have added a general statement to account for this citation: "It has been previously observed that high salinities can drive the precipitation of certain clays in natural systems³⁸, which could explain why this film did not precipitate at low salinities and in freshwater."

Unfortunately, while some of the experimental results are of interest, the paper over-inflates the significance of these findings through several mischaracterizations and misrepresentations.

We have accounted for all comments left by Reviewers and edited to discussion accordingly.

First, the authors' work focusses solely on the exoskeletal elements of their caridean test subjects. They incorrectly claim that these exoskeletal elements equate to "soft tissues", from which they continue to extrapolate their findings to present a major advance of our understanding of "soft tissue preservation". Euarthropod exoskeletons are cuticles secreted by specialized epithelial cells in the epidermis – hence, they are cellular secretions, not tissues themselves. Euarthropod cuticle is composed of a matrix of intertwined polysaccharides (predominantly chitin) and proteins. Through localized de-watering, and to a lesser extent through cross-linking, these originally soft and flexible cuticles become hard and rigid in a process that is referred to as tanning. In addition, in decapods, like the shrimp used in these experiments, these cuticles are further reinforced through biomineralization with calcium salts. While the extent of biomineralization differs between different decapod taxa, all are biomineralizing to at least some degree. This is actually attested by the authors' own EDS data which show the presence of Ca in the cuticle – something they avoid commenting on (in fact, the apparent absence of Ca in the spectrum for the specimen covered in aluminosilicates suggests that the limited biomineralization may have dissolved as a consequence of acidification associated with decay). Consequently, while shrimp exoskeletons are relatively delicate and poorly biomineralized compared to e.g. crabs or lobsters, they nevertheless are quite definitely neither "soft" nor "tissues", but in fact fall on the more recalcitrant side of the spectrum of preservation. So, the authors' description of the exoskeletal elements of the shrimp as "soft tissues" is doubly wrong, and as a result their characterization of the significance of their findings for our understanding of real soft tissue preservation is misleading. In fact, the authors implicitly admit as much themselves when they write (p. 11, lines 328 – 331): "These results could suggest that the abundance of arthropods in the Cambrian may not solely indicate ecological dominance but could also be the result of a taphonomic process like the black film formation that stabilized their carcasses shortly after death and favored their preservation over other taxa." So, with this sentence, the authors actually admit themselves that their results are not applicable to "soft-bodied" animals or "soft-tissues", but are only relevant to euarthropods which by definition possess rigid, hardened exoskeletons. And it is not exactly a novel idea that even a poorly sclerotized euarthropod generally still has a better preservation potential than a genuinely soft-bodied organism like e.g. a polychaete or a chaetognath.

We agree with Reviewer 1 that a shrimp cuticle is not exactly a soft tissue, however, animals having the same body constituents as shrimps are not found in all fossiliferous sites, and are almost solely found in Lagerstätten. In other words, this is simply a terminology change, that does not alter the scope of the paper. To avoid confusion, and avoid the usage of the words "tissue" and "soft", whenever necessary we replaced "soft tissues" with "lightly biomineralized structures". "Soft-tissue preservation" was replaced by "exceptional preservation" whenever necessary because as indicated earlier, sedimentary levels yielding fossilized shrimps in exquisite details are considered Lagerstätten (see Kimmig and Schiffbauer, 2014). We have also included the following text regarding Ca in the cuticle: "Calcium can also be detected in the analysed shrimps because it is one of the constituent elements of the cuticle (Fig 4i, n). Some of this calcium might have been mobilized away from the cuticle owing to low pH values that form around a decaying carcass 37."

Next, the authors set up a classical straw man argument, by implying that previous workers assumed "exclusive preservation of freshly killed organisms" (p. 12, lines 372-373) which they then use to construct the claim they have now identified a previously-ignored potential source of time averaging in obrution deposits which should be taken into consideration in future ecological studies of exceptionally preserved faunas – thereby falsely implying that this is a novel insight with major previously unrealized implications for the study of the ecology of exceptionally preserved faunas. In fact, the late Harry Whittington in 1971 already commented on a significant presence of decayed specimens in the Burgess Shale. Back in 1977, Conway Morris noted the decay sequence of *Ottoia prolifica* as observed in the specimens, observed decay in other Burgess Shale priapulids, and further explicitly remarked that a considerable number of specimens of *Selkirkia columbia* must have been dead before transportation and burial. The same author (Conway Morris 1986) in his famous paper on the community structure of the Burgess Shale Phyllopod Bed again commented on the topic of carcasses being present among the preserved biota, and the need to distinguish between specimens that were alive and dead at time of burial for community studies. Caron & Jackson (2006) specifically looked at high resolution at the presence of carcasses vs individuals transported and buried alive in the Greater Phyllopod Bed and explicitly stated (Caron & Jackson 2006, p. 458): "The relative ratio of species in different states of preservation varies through time, and in some instances the proportion of the time-averaged assemblage is much higher than the proportion of the census assemblage. This demonstrates that the event of burial is not always the main cause of death for many organisms." The same authors followed up this study of the taphonomy of the

Greater Phyllopod Bed with a work on the ecology (Caron & Jackson 2008). Here, they stated (Caron & Jackson 2008, p. 224): “Rarefaction curves suggest that preburial decay had no effect on species richness (Caron, 2005; Caron and Jackson, 2006). Therefore the effect of time-averaging through preburial decay is not an important controlling factor in the community.” These are only a couple of very well-known references that immediately come to mind, but this list is far from exhaustive, and it is difficult to understand how the authors could not have been aware of these works. Suffice to say that researchers working on exceptional preservation generally have been cognizant of the presence of carcasses and the potential for time averaging in exceptionally preserved faunas for some 40+ years at the very least, and have taken this into consideration when studying the ecology of ancient biotas; indeed, it stands to reason that it is almost inevitable that some carcasses will be included in an exceptionally preserved assemblage, considering that even extremely labile groups like e.g. polychaetes will not instantaneously vanish upon death. These two very major issues, which entirely undercut the premises of the manuscript, should in my opinion be enough to disqualify it from publication in its current form. However, there are also further shortcomings that, although of a lesser magnitude, still need to be noted.

To account for the comments raised by Reviewers 1 and 2, we have reshaped our discussion, and parts of the text the reviewer indicated as problematic were removed. We have also added the following text to the paper: “This finding has broader implications for understanding community preservation in Konservat-Lagerstätten recording the Cambrian Explosion. In these sites, a separation is usually made between carcasses decaying on the seafloor, termed “time-averaged assemblage,” and those killed during the burial event, termed “census assemblage”^{51,52}. Our experimental observations suggest that census assemblages might represent previously dead communities that were stabilized by aluminosilicates instead of freshly killed ones. This process might be operational in hundreds of Cambrian fossiliferous sedimentary layers where kaolinite has been identified in the sedimentary matrix²⁸, and the high preservation potential of arthropods in these sites does not necessarily imply the exclusive preservation of freshly killed organisms.”

In their methods section (p. 3, line 98-100, and supplementary text p. 1), the authors state:

“We opted to use pure ASW and deionized water to limit the number of variables in the experiment by excluding bacteria, plankton, and other organisms found in aquaria and natural aquatic environments in highly variable proportions.” So, apparently, they believe that de-ionized water is essentially sterile. This is incorrect. De-ionized water is exactly what the name says – it is water from which the ions have been removed through adsorption on a resin matrix. However, this process does nothing to remove neutrally charged molecules or microbiota. To assure that the water is sterile, it would need to be distilled, or possibly filtered through a dedicated nanofiltration systems. Given this, the authors’ assertion that the majority of the bacteria present in their experimental system originated from the carcasses is tenuous at best.

This is an accidental omission from the text of the first version of our manuscript, because in fact reverse osmosis water was used. Reverse osmosis water ensures no environmental bacteria is present in the system. This has been corrected in the current version of the manuscript: “We opted to use pure reverse osmosis deionized water to limit the number of variables in the experiment by excluding bacteria, plankton, and other organisms found in aquaria and natural aquatic environments in highly variable proportions.”

In the text, the authors also constantly refer to “lagerstätten”. First, this is a German noun, so it needs to be capitalized. Second, Seilacher originally distinguished two types of Lagerstätten – Konservat-Lagerstätten (exceptionally preserved biotas) and Konzentrat Lagerstätten (concentration deposits with an abundance of material); later authors sometimes added further types of Lagerstätten, e.g. “Echinoderm-Lagerstätten”. So, for clarity, it is important to indicate which type of Lagerstätte one is talking about – and in this case, the authors should use “Konservat-Lagerstätten”.

This has been corrected throughout the paper and “Lagerstätten” has been replaced with “Konservat-Lagerstätten” whenever necessary.

Further remarks and comments are included in the attached annotated manuscript.

These comments were accounted for.

Finally, there is an ethical dimension to this entire paper. Under European legislation, the authors were not required to obtain any approval for their research from an ethical committee. However, there is an increasing body of research that suggests that decapods are sentient and capable of feeling pain (e.g. Elwood 2012; Passantino et al. 2021), leading countries like Switzerland (where the authors are based) and New Zealand to outlaw the live boiling of lobsters. As a consequence, there have also been increasing calls to regulate the use of crustaceans in research and put their use under ethical oversight (e.g. Mather 2019; Passantino et al. 2021; Rowe 2018). Considering that none of the work presented here has any practical value whatsoever, but was carried out simply to satisfy the intellectual curiosity of the authors, I would like them (and, indeed, the editor) to ponder the following question: do you feel that it is ethically and morally justifiable to kill 43 potentially sentient individuals merely to quell your curiosity?

We have adhered to all required ethical elements regarding shrimp experiments in Switzerland, where the experiments were made. Per the editorial suggestion, the following text was added to the acknowledgements: “No ethical approval was required to do these experiments, which were funded by the Swiss National Science Foundation. The taxa used in these experiments are not protected in Switzerland. Individuals were euthanised ethically, and the experiments adhered to the 3Rs principle of research (reduce, reuse and recycle).”

Ethics aside, this paper does present some interesting data that merit publication. However, the manuscript suffers from such significant inaccuracies and misrepresentations, which are used to construct an inflated narrative of the importance of the authors’ work, that it really cannot be considered for publication anywhere in its current form.

We would like to thank Peter for his helpful and thoughtful comments that helped us improve the manuscript.

Peter Van Roy
04 July 2024

Reviewer #2 (Remarks to the Author):

This manuscript by Nora Corthésy and colleagues firmly demonstrates, from experiments, that decay of arthropod carcasses is retarded to different degrees when lying on three investigated clay types (kaolinite, bentonite and montmorillonite), and argue how this may influence exceptional preservation in mainly Burgess Shale-type Lagerstätten.

The experiments are notable because they show that kaolinite is the most effective clay to retard decay, and stabilise tissues, even without burial, and that this stabilisation is associated with aluminosilicate clay templating on the carcasses surfaces. While tissue stabilisation and aluminosilicate clay templating has previously been demonstrated to occur in association with kaolinite in the decay experiments by e.g. Naimark et al. (2016a <https://doi.org/10.1111/pala.12246> and 2016b <https://doi.org/10.1017/jpa.2016.23>) and Newman et al. (2019 <http://dx.doi.org/10.2110/palo.2019.030>), the core strength of the manuscript is that it, through a large and well-documented dataset, adds compelling evidence that these processes start rapidly after death and without burial. Therefore, the main implication of this manuscript is that it convincingly shows that the fossilisation process already starts shortly after death and irrespective of burial. The manuscript is timely since there currently is a lot of work on understanding the preservational biases in fossil Lagerstätten and how these biases shape our interpretation of the fossil record. The chosen methods are well-argued and well-described, and the results are generally well-presented and support these conclusions.

We would like to thank Reviewer 2 for this summary that beautifully showcases the significance of the paper. We have also noticed that the reference Newman et al. (2019) was not cited in the previous version of the manuscript. We have included it in the current version.

As an ecological implication, the authors use this to argue that arthropod death assemblages on kaolinite-rich seabeds (purportedly the Burgess Shale Lagerstätten) may be subjected to a more extended time-averaging if carcasses can lie seemingly undisturbed for ~weeks. This section is the manuscript's weakest part. First, in my opinion, time-averaging at this temporal scale is not overly important for ecological interpretations as populations and communities mostly shift at a higher temporal scale (months/years). In fact, at the monthly scale, another decay experiment with shrimp larvae on kaolinite-surfaces show that the carcasses had dissolved after 3 to 4 months (Naimark et al., 2016b <https://doi.org/10.1017/jpa.2016.23>). Second, the authors implies that it is generally assumed that fossil assemblages in single-bed burial events only contain freshly killed animals, such as in the Burgess Shale. However, it is well established that e.g. Burgess Shale beds contain mixtures between freshly killed (census) assemblages as well as the accumulating death assemblages (Caron & Jackson 2006 <https://doi.org/10.2110/palo.2003.P05-070R> and others since them). However, the authors could still use their findings to make the argument that the completely preserved specimens interpreted to represent census assemblages in Caron & Jackson (2006) could alternatively represent recently deceased death assemblages.

We removed the time-averaged part from the manuscript following the comments of both Reviewers 1 and 2. Currently the text reads: “This finding has broader implications for understanding community preservation in Konservat-Lagerstätten recording the Cambrian Explosion. In these sites, a separation is usually made between carcasses decaying on the seafloor, termed “time-averaged assemblage,” and those killed during the burial event, termed “census assemblage”^{51,52}. Our experimental observations suggest that census assemblages might represent previously dead communities that were stabilized by aluminosilicates instead of freshly killed ones. This process might be operational in hundreds of Cambrian fossiliferous sedimentary layers where kaolinite has been identified in the sedimentary matrix²⁸, and the high preservation potential of arthropods in these sites does not necessarily imply the exclusive preservation of freshly killed organisms.”

Overall, the manuscript is good and deserves publication. The main results will predictably shape future experiments to investigate the timing and processes of exceptional preservation in clay systems. However, the text could be clarified in some sections and certain auxiliary implications scaled back a bit. The authors will be able to do this without changing anything to their experiments or data presentation.

We would like to thank Reviewer 2 for their detailed comments which we have carefully addressed. Sections were clarified and certain auxiliary implications were scaled back per the recommendations of all reviewers.

Line 16-18: “This is the first experimental evidence for the replication of soft tissues in aluminosilicates shortly after death in Burgess Shale-type fossil lagerstätten recording the Cambrian Explosion.” Unless I misunderstand the sentence, it should be clarified a bit since a) the experiments do not mimic the environmental conditions of Burgess Shale beds, and b) is not the first experimental evidence of aluminosilicate replication of soft tissues (see references above).

This has been edited to focus on the main finding of this paper: “This is the first experimental evidence for the replication of arthropod lightly biomineralized structures in aluminosilicates shortly after death, while carcasses are not buried by sediments.”

Line 19-20: “The preservation of morphology through aluminosilicates could frequently result in carcasses persisting on the seafloor for extended durations [...]” In my opinion, this sentence should define the length of “extended durations” since these can vary from days to years, depending on the chosen scale of investigation.

This has been edited to: “The preservation of morphology through aluminosilicates could result in carcasses persisting on the seafloor for weeks without losing much external anatomical information.”

Line 20-21: “meaning that instantaneous or near-instantaneous burial may not be a prerequisite for soft tissue preservation as usually thought.” Again, “instantaneous” and “near-instantaneous” would benefit from being defined to avoid confusion since researchers may work at very different time scales. I think it is commonly assumed that many fossils in exceptionally preserved beds may represent decayed death assemblages. Alternatively, in my opinion (and therefore just a subjective suggestion), the authors could strengthen the manuscript if they instead highlight that their results indicate that the fossilisation process starts before burial.

The time-averaged part was removed from the manuscript as indicated earlier in our response. Precision was added to better define “instantaneous burial”. The last sentence of the current abstract reads: “In this context, instantaneous burial capturing living animals may not be a prerequisite for exceptional preservation as usually thought.”

Line 22-25: As I argued above, I think it is uncertain how much this cryptic time-averaging (albeit convincingly demonstrated) will affect palaeoecological interpretations.

We have removed this part from the manuscript following the comments of Reviewers 1 and 2.

Line 70-71: “Slower degradation is accompanied by the precipitation of newly formed aluminosilicates on the marine shrimp carcasses, providing the first experimental evidence for this mode of authigenic mineralization.” It should be clarified that while aluminosilicate clay templating/mineralization has been shown in other experiments (see refs for Naimark et al., 2016a,b and Newman 2019 above), the manuscript provides the first convincing evidence that the mineralization is authigenic.

The text here was edited to focus on the difference between our experiment and previously made ones: “Slower degradation is accompanied by the precipitation of newly formed aluminosilicates on the marine shrimp carcasses, providing the first experimental evidence that this mode of authigenic mineralization can occur in the absence of burial.”

Line 76-78: “This observation holds particular significance in the context of the Cambrian Explosion, as kaolinite is frequently observed in association with soft tissues in marine rock from the Cambrian Period.” This argument would be strengthened if the authors provided references showing the frequent observation of soft tissues associated with kaolinite-rich sediments.

We have added Anderson et al. (2018) which was present in the discussion here as well. Anderson et al. (2018) investigated hundreds of fossiliferous levels and have shown a positive correlation between exceptional fossil preservation and kaolinite concentrations.

Line 114: “Seven different taphonomic scores, numbered from 0 to 6 (0 = intact, 6 = completely degraded), were defined based on the global degradation aspect of the animal (see Tab. S1 & 114 Fig. S1 in Supplementary Materials).” I am uncertain what “global degradation aspect” refers to; does it refer to the general degradation aspects observed in this study only or in all decay experiments collectively?

Precision is added: “Three different taphonomic scores, numbered from 0 to 6 (0 = intact, 6 = completely degraded), were defined based on the degradation aspect of the cephalothoracic carapace, the appendages and the eyes of the animals in this study (see Tab. S1 & Fig. S1 in Supplementary Materials). To determine the global taphonomic score of each individual, the median of the carapace, appendages and eyes taphonomic scores was calculated (see Tab. S1 & Fig. S1 in Supplementary Materials). For score 0, the shrimp is intact and transparent. Score 1 corresponds to the carapace and appendages becoming pink and opaque, and the eyes turning black. When biofilms and/or other coatings are observed on the decaying carcass, score 2 is attributed. Score 3 is observed when the carapace detaches from the cephalothorax, at least 50% of the appendages are disarticulated, and the eyes detach from the cephalothorax. Score 4 is attributed when the carapace is completely detached from the cephalothorax so the gills and internal organs are exposed, at least 50% of the appendages are detached, and the eyes are detached. Score 5 is assigned when the cephalothorax and carapace are separated from the abdomen, at least 50% of the appendages are broken into several small pieces, and the eyes show advanced deterioration. Finally, for score 6, the individual is completely degraded.”

Line 214-215: “The black film (Fig. 2) was formed uniquely in the experimental condition of marine shrimp decaying on kaolinite at 1.024 psu.” Did the black film form on all carcasses decaying on kaolinite at 1.024 psu or only some?

On all of them. Precision is added: “The black film (Fig. 2) was formed uniquely on all marine shrimps decaying on kaolinite at 1.024 psu. The black film did not form in the single marine shrimp left to decay on kaolinite at a slightly lower salinity (1.019 psu, prepared for Cryo-SEM).”

Line 224-225: “This black film has not been reported in previous publications on decay experiments.” Could the black films (“clay veneers”) in Newman et al. (2019 <http://dx.doi.org/10.2110/palo.2019.030>) possibly be similar? Albeit from a different experimental setup.

This is very hard to say considering differences in the experimental setup. However, we still added the following sentence to the manuscript: “This black film forming on decaying marine arthropod has not been reported in previous publications on decay experiments although, possibly similar aluminosilicate veneers have been observed on scallop muscle tissues buried in kaolinite 20 and on anemone decaying on kaolinite substrates 21. Directly comparing the results of Newman et al.20 and this work is a challenging task considering major differences in experimental design. Also, Slagter et al.21 did not work on arthropods and obtained their results with other substrates in the system, such as quartz sand, iron oxide, and microbial biofilm in addition to the kaolinite substrate, which is a different experimental design from our own.”

Line 236-239: “For the specimens on which no black film formed, the remains of the cuticle, observed with the Cryo-SEM, show an organic surface (Fig. 4A), covered by bacteria, mainly coccoids, ranging in size from 0.6 to 238 1.5µm (Fig. 4B).” This is a bit unclear to me. According to the methods (lines 120-130), three carcasses decaying on kaolinite were prepared for Cryo-SEM. How many of these did not form black films? The text on lines 236-239 indicates that it did not form on multiple carcasses and not just the single carcass on kaolinite at 1.019 psu.

Precision was added to the text to indicate clearly that only one shrimp without black film was analysed: “For specimens on which no black film formed, the analysed shrimp under Cryo-SEM, show an organic surface (Fig. 4a), covered by bacteria, mainly coccoids, ranging in size from 0.6 to 1.5µm (Fig. 4b). These bacteria are likely to correspond to the white filamentous layer that was observed on all the samples except for the shrimps placed on kaolinite (Fig. 1b, c). For the two marine shrimps on which a black film formed (Fig. 4c), the cuticle is irregular (Fig. 4d), showing more relief than in the sample where a black film did not form (Fig. 4a)”

Line 271: “Importantly, the distribution of aluminosilicates in this film is non-uniform” Would it be possible to provide some kind of meaningful general density/frequency intervals of the aluminosilicate sheets to estimate their prevalence within the black films? Unfortunately, this is not possible because the samples were not polished to ensure that meaningful quantifications using SEM-EDX could be made.

Line 273-276: “The absence of these elements in some areas could mean that aluminosilicate nuclei are too small to be detected and that specific regions of the shrimp are more conducive to mineralization than others, although no clear differences could be seen while investigating the shrimp photographs at a macro scale.” Have these nuclei been observed or are they assumed to be present? If they have not been observed, other possibilities could be discussed. Is it, for example, a possibility that the aluminosilicates just have not precipitated everywhere despite the black film being present at macro scale?

This is also indeed a possibility that we include in the current version of the text: “The absence of these elements in some areas could mean that aluminosilicate nuclei are too small to be detected. It could also mean that specific regions of the shrimp are more conducive to mineralization than others, although no heterogeneity of aluminosilicates is visible at a macro scale.”

Line 306-310: “These colour differences, and the scarcity of large, ordered aluminosilicate sheets covering the shrimp cuticle (Fig. 4F-H) suggest rapid in-situ nucleation (32). When rapid nuclei formation outpaces mineral growth, it results in an enrichment of aluminium and silicon in small nuclei rather than well-structured sheets, which may develop later over time (33)” Again, it is unclear if these nuclei have been observed. If not, I think it should be clearly stated in the text.

The following sentence was added to the manuscript: “The absence of these elements within the samples could mean that aluminosilicate nuclei are too small to be detected. It could also mean that specific regions of the shrimp are more conducive to mineralisation than others, although no heterogeneity of aluminosilicates is visible at a macro-scale.”

Line 327-322: “The formation of a black film has, thus far, been exclusively observed on the carcasses of marine arthropods. These results could suggest that the abundance of arthropods in the Cambrian may not solely indicate ecological dominance but could also be the result of a taphonomic process like the black film formation that stabilized their carcasses shortly after death and favored their preservation over other taxa.” It is my understanding that these experiments have only been performed on arthropods. Unless I am mistaken, I think it is a stretch to attribute the rich arthropod record in Cambrian Lagerstätten to this process, especially since kaolinite has not, to my knowledge, been observed in all other Lagerstätten. Additionally, if the black clay veneers in the scallops of Newman et al. (2019 <http://dx.doi.org/10.2110/palo.2019.030>) is similar to the black films shown in this manuscript, the taphonomic process may extend to other phyla than just arthropods.

Anderson et al. (2018) found a positive correlation between chamosite/berthierine (representing original kaolinite in the environment) and soft tissue preservation in hundreds of exceptionally preserved assemblages. However, nuance was added to this part of the discussion: “However, since authigenic clay were also observed on unburied decaying anemone 21 although placed in different environmental conditions, this hypothesis is yet to be fully proven by similar experiments on other animal groups. This would investigate whether aluminosilicate precipitation on decaying carcasses other than arthropods can happen without burial or whether non-arthropod carcasses need to be buried for this process to occur (see for instance Newman et al.20). In addition, it is important to remember that the black film only forms with standard marine salinity (1.024 psu) and not with a lower salinity (1.019 psu). Therefore, the authigenic mineralisation of the arthropod carcasses may be limited to the marine realm though the range of viable salinities has not been constrained in detail and no upper bound was defined.”

Line 336: “[...] driven by the precipitation of authigenic aluminosilicates on the carcass (Fig. 6 A, B).” Has it been fully established that the driver of preservation at this timing is the precipitation of authigenic aluminosilicates rather than kaolinites inhibition of decay-inducing bacteria with authigenic precipitation as a side product?

More precision was added to the manuscript: “This can be due to the inhibition of certain bacterial strains by kaolinite^{34,35} and by the precipitation of authigenic aluminosilicates on the carcass in the presence of kaolinite.”

Line 344-345: “As such, instantaneous burial may not always have been necessary in marine fossil lagerstätten containing decapod crustaceans, and perhaps other arthropods.” and Line 351-256: “Note that the suggested lack of instantaneous burial does not contradict observations of rapid obrution events in Burgess Shale-type deposits (37–42). Obrution events are fast, but they do not always occur regularly in depositional environments and they depend on numerous parameters such as seasonality (43). As such, animals could be decaying for days or weeks on the seafloor before their burial by a rapid sedimentary event.“ As stated above, Caron & Jackson (2006 <https://doi.org/10.2110/palo.2003.P05-070R>) already argued that Burgess Shale-type burial events contain both census assemblages as well as accumulated death assemblages, implying that instantaneous burial is not necessary for exceptional preservation and that some fossils may have accumulated (and decayed) on the seafloor for prolonged periods. However, Caron & Jackson do assume that completely preserved fossils represent freshly killed census assemblages.

Indeed, we have edited the manuscript to account for these studies as indicated earlier in our response. Here, we simply wanted to provide sedimentary context for non-sedimentologists, that our suggestion that instantaneous burial might not be necessary for exceptional preservation does not contradict the observation of rapid burial events in Konservat-Lagerstätten.

Line 357-377: This section argues for cryptic time-averaging since the black film on kaolinite beds can retard decay for weeks. However, since the composition of communities and populations in a given environment does not vary significantly at the daily/weekly-scale, but rather at a monthly/yearly scale, it is uncertain what the impact of this geologically very short time-averaging is. Perhaps it could be speculated that the black films, in combination with other processes, could potentially induce time-averaging at a larger scale. Like for the section above, Caron & Jackson (2006) (and several other publications since) have already argued that Burgess Shales-type burial events already consist of both census and death assemblages.

We removed the time averaging part from the discussion as suggested by Reviewers 1 and 2. The current text reads: “This finding has broader implications for understanding community preservation in Konservat-Lagerstätten recording the Cambrian Explosion. In these sites, a separation is usually made between carcasses decaying on the seafloor, termed “time-averaged assemblage,” and those killed during the burial event, termed “census assemblage”^{51,52}. Our experimental observations suggest that census assemblages might represent previously dead communities that were stabilized by aluminosilicates instead of freshly killed ones. This process might be operational in hundreds of Cambrian fossiliferous sedimentary layers where kaolinite has been identified in the sedimentary matrix²⁸, and the high preservation potential of arthropods in these sites does not necessarily imply the exclusive preservation of freshly killed organisms.”

Line 383-386: “Any part of the organism that has not been replicated by aluminosilicates would, in turn, be preserved in carbon (8, 40)(Fig. 6D), which would result in a complex fossil preserved in both carbonaceous compressions and aluminosilicates owing to their flattening over geological times (3, 9)(Fig. 6D).” It could be clarified whether aluminosilicate coats or replaces the carbon. As I read it right now, the text suggests, to me, that the aluminosilicates replace the carbon and that carbon only occurs where there are no aluminosilicate sheets, rather than aluminosilicates co-occurring with the carbon. Additionally, the text currently appears to suggest that the flattening is the driver of the complexity and not the authigenic aluminosilification.

More precision is added to the text: “These processes result in organisms preserved in aluminosilicates^{3,9} in association with organic material^{8,47}. Due to the flattening of this organic material over geological times, the resulting fossils would consist of carbonaceous compressions with accessory minerals.”

Congratulations on the manuscript, which I hope to see published.

We would like to thank Morten for his helpful and thoughtful comments that helped us improve the manuscript.

Best wishes,
Morten Lunde Nielsen

Reviewer #3 (Remarks to the Author):

Comments for Author

This is a really interesting manuscript that explores the interactions and effects of substrate on decay in shrimps and investigates Burgess Shale-type preservation pathways. The work is well-written, and the figures complement the text. The approach and methods used are sensible and highly appropriate for this field of work, and they provide a nice framework for future workers.

The resulting novel evidence of the rapid formation of an aluminosilicate covering and the disruption of the decay process on a Kaolinite substrate is very exciting. I agree with the authors that, as well as answering a long-standing question surrounding the timing of Burgess Shale-type preservation, the findings have wide-ranging implications for the interpretation of fossils and fossil-bearing sites and subsequent ecological reconstructions. I think that as this work aims to quantify taphonomic processes and identify pathways that underpin the fossil record, it will be of interest to a broad range of palaeontologists and also to adjacent fields such as archaeology and forensics.

While I think this is an important piece of work, I feel that some minor revisions are needed to improve the repeatability of the methods and clarify some areas of the manuscript. These do not require additional analysis or data collection and are largely minor edits, so I am confident the authors can quickly address them.

Overall, this is an excellent piece of work, and I recommend it for publication once the revisions have been made.

We would like to thank Reviewer 3 for the comments and suggestions, which have helped us to improve our manuscript.

General comments:

- My primary concern is that I don't think the methods are detailed enough to support the reader, especially with the taphonomic scoring, but also with the general setup. As it is, the manuscript gives a good general explanation, but I think it would be difficult to replicate. I think the addition of extra text, largely in the extended methods, would quickly rectify this.

We have edited the manuscript as suggested by Reviewer 3 to give more details in the Method section:

“Shrimps were individually left to decay in sterilized acrylic boxes (5 x 3 x 2 cm), closed but not sealed, containing 5g of sediment with one of three compositions: kaolinite, bentonite, or montmorillonite. The clays used have a purity between 80-90%, with the remaining 10-20% phase consisting dominantly of quartz minerals. Each shrimp deposited on the surface of the sediment was covered with 35g of water: reverse osmosis deionized water for freshwater shrimps and artificial seawater (ASW) for marine shrimps, prepared to 1.024 psu with reverse osmosis deionized water and Aquarium Systems Reef Crystals.”

“Three different taphonomic scores, numbered from 0 to 6 (0 = intact, 6 = completely degraded), were defined based on the degradation aspect of the cephalothoracic carapace, the appendages and the eyes of the animals in this study (see Tab. S1 & Fig. S1 in Supplementary Materials). To determine the global taphonomic score of each individual, the median of the carapace, appendages and eyes taphonomic scores was calculated (see Tab. S1 & Fig. S1 in Supplementary Materials). For score 0, the shrimp is intact and transparent. Score 1 corresponds to the carapace and appendages becoming pink and opaque, and the eyes turning black. When biofilms and/or other coatings are observed on the decaying carcass, score 2 is attributed. Score 3 is observed when the carapace detaches from the cephalothorax, at least 50% of the appendages are disarticulated, and the eyes detach from the cephalothorax. Score 4 is attributed when the carapace is completely detached from the cephalothorax so the gills and internal organs are exposed, at least 50% of the appendages are detached, and the eyes are detached. Score 5 is assigned when the cephalothorax and carapace are separated from the abdomen, at least 50% of the appendages are broken into several small pieces, and the eyes show advanced deterioration. Finally, for score 6, the individual is completely degraded.”

“For each shrimp, SEM images, elemental spectra, and an elemental map of eight different random areas of the cephalothorax and the abdomen were acquired using Quanta FEG-250 Scanning Electron Microscope at 10 keV.”

- Taphonomic Scoring: I think it is a sensible system, and I can broadly understand what was done, but as scoring is the key method for quantifying the decay data for statistical comparison, more explicate information on how the system was established and applied would be appropriate. The methods provide a short statement that the scoring system was derived through observation, but I don't think it is clear enough to understand how or why the boundaries were established. The addition of information such as what was observed and for how long along with why those markers were chosen as a boundary (I assume it was due to major or distinct morphological changes, but I don't know) would help the reader understand and also help future workers adapt the method. Similarly, while Fig S1 is very helpful for communicating morphological states, I would like more information on how scores are

assigned and the thresholds for scores. As it is, I think I would struggle to replicate the scoring system confidently and get the same results for each specimen. For example, in the case of appendages, how many are needed to be disarticulated to score a 3? Is it all, or is just one enough? How do I score intermediate or mixes of categories? If I had a shrimp that had a carapace cuticle detaching from internal organs (3) but no disarticulated appendages (2) or detached eyes (2), would I score that as a 3 or a 2? I think this is especially important to help support the finding of a decay plateau/delay in the Kaolinite specimens – is it truly a pause, or do they decay in a different order, or is it an artefact of the scoring system? It is hard to distinguish this as a reader, so including the details of the system in the methods would provide more evident support for the findings.

- Related to this, I think it would also be useful to have a breakdown of the scores or morphology per specimen, but I appreciate that Fig 3 & S3 do give a sense of this information so it is less important than the detail of the system itself.

We would like to thank Reviewer 3 for pointing out the lack of detail concerning the quantification of decay. We added more details in the manuscript to better explain how the taphonomic scores were assigned to the decaying organisms.

“Three different taphonomic scores, numbered from 0 to 6 (0 = intact, 6 = completely degraded), were defined based on the degradation aspect of the cephalothoracic carapace, the appendages and the eyes of the animals in this study (see Tab. S1 & Fig. S1 in Supplementary Materials). To determine the global taphonomic score of each individual, the median of the carapace, appendages and eyes taphonomic scores was calculated (see Tab. S1 & Fig. S1 in Supplementary Materials). For score 0, the shrimp is intact and transparent. Score 1 corresponds to the carapace and appendages becoming pink and opaque, and the eyes turning black. When biofilms and/or other coatings are observed on the decaying carcass, score 2 is attributed. Score 3 is observed when the carapace detaches from the cephalothorax, at least 50% of the appendages are disarticulated, and the eyes detach from the cephalothorax. Score 4 is attributed when the carapace is completely detached from the cephalothorax so the gills and internal organs are exposed, at least 50% of the appendages are detached, and the eyes are detached. Score 5 is assigned when the cephalothorax and carapace are separated from the abdomen, at least 50% of the appendages are broken into several small pieces, and the eyes show advanced deterioration. Finally, for score 6, the individual is completely degraded.”

- There could also be some more detail given to the physical set-up (albeit it constrained in the supplementary extended section rather than in the main body of the manuscript). Inclusion of a figure or description of the setup of the acrylic boxes, including the dimensions, number of shrimps per box etc. would be very helpful. It would help give a sense of the size of the animals in relation to the surrounding clay surfaces, and help future workers who wish to use your protocols.

The method section was edited as suggested by Reviewer 3: “Adult freshwater shrimps (*Neocaridina davidi*; 3 cm long) and marine shrimps (*Palaemon varians*; 1.5 cm long) [...]” “Shrimps were individually left to decay in sterilized acrylic boxes (5 x 3 x 2 cm), closed but not sealed, containing 5g of sediment with one of three compositions: kaolinite, bentonite, or montmorillonite. The clays used have a purity between 80-90%, with the remaining 10-20% phase consisting dominantly of quartz minerals. Each shrimp deposited on the surface of the sediment was covered with 35g of water: reverse osmosis deionized water for freshwater shrimps and artificial seawater (ASW) for marine shrimps, prepared to 1.024 psu with reverse osmosis deionized water and Aquarium Systems Reef Crystals.”

- Additionally, with the cryo-SEM and the EDS, indication of where on the shrimp the scans were taken from would help the reader understand if differences in the distribution of the aluminosilicate considered in the discussion are related to the location or not.

We have edited this section: “For each shrimp, SEM images, elemental spectra, and an elemental map of eight different random areas of the cephalothorax and the abdomen were acquired using Quanta FEG-250 Scanning Electron Microscope at 10 keV.”

- Black film vs White biofilm – This is a slightly pedantic question but how have you established that the white film is a biofilm and not something else? Perhaps reword to white film?

We would like to thank Reviewer 3 for pointing out this question. Bacteria were observed with SEM images on the cuticle of shrimps where the black film did not form (Fig. 4b). We have observed filamentous and coccoids forms corresponding to a biofilm. On the other hand, no bacteria accumulation, corresponding to a biofilm, was observed with SEM images on the other shrimps where the black film formed. As mentioned in the manuscript, at taphonomic score 2, we have observed a white filamentous layer (white film) on all samples, except for the shrimps decaying on kaolinite and thus where the black film formed. Therefore, the white film, that is absent from the samples with the black film, is likely to correspond to a bacterial film. We have edited the manuscript to explain this hypothesis: “For specimens on which no black film formed, the analysed shrimp under Cryo-SEM, show an organic surface (Fig. 4a), covered by bacteria, mainly coccoids, ranging in size from 0.6 to 1.5µm (Fig. 4b). These bacteria are likely to correspond to the white filamentous layer that was observed on all the samples except for the shrimps placed on kaolinite (Fig. 1b, c).”

- Some consideration of the sample size would be appropriate in the discussion.

The discussion was edited as suggested: “Despite the small sample size, the decay rate of marine shrimps placed on kaolinite is significantly different from the decay rate of the other specimens (Fig. 3).”

- Finding that the black film does not form in lower salinity is really interesting, and I think it is worth a few sentences of consideration in the Taphonomic and Ecological implications section. Do you have a sense of what this could present like over salinity gradients in costal/inlet regions? Alongside the temporal complexity that this preservation style might create, it seems like it might capture some additional localised environmental biases too, adding to the difficulties of understanding lagerstätten.

We have edited the Discussion section as suggested by Reviewer 3: “In addition, it is important to remember that the black film only forms with standard marine salinity (1.024 psu) and not with a lower salinity (1.019 psu). Therefore, the authigenic mineralisation of the arthropod carcasses may be limited to the marine realm though the range of viable salinities has not been constrained in detail and no upper bound was defined.”

- Fig 3 E -This is a lovely figure that really complements the descriptive sections and helps communicate the variation in decay seen. I assume the pink objects and black blobs in the small circles are meant to represent bacteria or the films forming? A key or explanation in the caption would help to clarify this.

We added some details in the caption as suggested by Reviewer 3: “For the shrimps placed on bentonite, montmorillonite and in the absence of clay, the decay pattern is the same, with the formation of a white filamentous layer that is likely to correspond to a biofilm represented with the white colour and the little pink bacteria in the circle. The black film only forms in the presence of kaolinite hence the representation of black shrimps in this specific experimental condition and the poorly mineralized aluminosilicate crystals are represented in the circle.”

- Fig 4 B – Highlighting an example of a bacteria on this would help orientate the reader.

The Figure 4B was edited to highlight some bacteria.

- There is some inconsistency throughout on the use of shrimp/shrimps as a plural, e.g., L134 marine shrimps is used, then on L134-5 marine shrimp is used. It's largely not a problem for understanding the work, but is worth checking. The only place I think it causes some ambiguity is L216 where I could read the “shrimps” as marine or fresh water shrimp which does change the narrative of the paragraph.

We would like to thank Reviewer 3 for pointing this out. We edited the manuscript as suggested.

Comments on the manuscript:

Lines 22-24 – Consider rewording this sentence. I feel it gives the impression that heavily decayed morphology would be similar to ‘fresh’ morphology, rather than there being a delay in the decay process so the older specimens are less decayed than expected impacting temporal reconstructions (which I think is what this section is indicating?)

The abstract has been greatly revised, considering the comments of Reviewer 3 and in order to fit the formatting style of Communications Earth & Environment.

Lines 51-53 - “If their association...”, “..unclear whether these aluminosilicates...” - In this section I find the use of them/these slightly hides the subject of the sentences and disrupts the readability. It could be replaced both with “the” and keep the meaning.

We have edited the manuscript and replaced “their” and “this” by “the”.

Line 63 – “... for a range of animals,...” please include some reference examples here, such as Sansom et al 2011 and Briggs & Kear 1993 (or similar)

We have added references as suggested by Reviewer 3.

Line 65-78 – I think a small direct sentence on why this work was done (e.g., in order to test/investigate...etc) would create a stronger link between the introduction and this paragraph.

We have edited the manuscript as follow: “In order to test the effects of clay minerals on decaying carcasses without burial, euthanised specimens of the marine shrimp *Palaemon varians* and the freshwater shrimp *Neocaridina davidi* were placed atop beds of three different clay compositions and underwater to simulate conditions at the sediment-water interface.”

Line 94-96 – This is an important point but in this form reads like a discussion point. Consider editing it to highlight that consistent use of clove oil is the control a variable as done with the following sentences about ASW (and maybe moved to the area above about euthanasia)

The manuscript was edited: “All shrimps were euthanized using clove oil, to ensure that the method of euthanasia cannot explain why some decay patterns occurred only in the presence of specific clays.”

Lines 100-101 – Consider removing this sentence, I think it is repeating the previous sentence in a less informative way. It might be that some of the nuance of it has been lost in the transition from extended method to main body?

This comment has been taken into account and this sentence was removed from the manuscript.

Lines 109-111 – I understand this sentence to mean that after 10 days photos were only taken if there were major taphonomic changes – is that correct? – or was a photo taken every week while also checking for major changes regardless of whether or not any changes had happened? I could read it both ways so is worth checking to remove ambiguity. Additionally, this is slightly different from the method in the supplementary where the check is “periodically” not weekly.

The manuscript was edited to avoid ambiguity: “Following the ten days, specimens were allowed to continue to decay for up to 21 days, during which time they were checked daily and additional photos were taken in the event of major taphonomic state changes.”

Line 110 – weekly checked → checked weekly

The manuscript was edited.

L164&166 Fig 1 Caption - consider rewording “is detaching” maybe to ‘detaches’ or ‘beings to detach’

The caption of Figure 1 was edited as suggested.

Line 211 – “...organs are removed..” Consider rewording, ‘removed’ sounds intentional, ‘displaced’ maybe?

The word “removed” was replaced by “displaced”.

Line 213 – Can you elaborate on the one difference? I don't think it needs a lot of discussion, just stating what it is would be fine. We edited the manuscript to give more details: “Generally, no significant differences are observed between freshwater shrimps decaying on different substrates (Tab. S3), other than a single significant difference occurring at 48 hours when the carapace detaches faster for the shrimps placed on montmorillonite than in the other conditions (Fig. S3).”

Lines 271&273 – “certain areas” and “in some areas” – Where are the certain areas and are they consistent through the shrimp? The elemental variation within the body maybe linked to analysis artefacts as suggested, but it's quite hard to judge that as a reader without some understanding of the areas mentioned. If it's not random then something else might be happening that is related to the organism. This might be fixed by changing the word choice – deleting certain and some – but I think an idea of the distribution of these areas would be helpful to support the point, even if constrained to the supplementary material.

The manuscript was edited: “Importantly, the distribution of aluminosilicates in this film is non-uniform, as within the samples exhibiting the black film, aluminium, silicon, and potassium are not distributed on the entire surface of the samples (Fig. 5I-K). The absence of these elements within the samples could mean that aluminosilicate nuclei are too small to be detected.”

We would like to thank Reviewer 3 for the interesting and constructive comments that helped us improving the manuscript.

REVIEWERS' COMMENTS:

Reviewer #2 (Remarks to the Author):

The authors' replies to my own and reviewer 1's comments are satisfactory. The authors also satisfactorily handled the three outlined editorial thresholds.

I therefore consider the manuscript ready for publication.

Best wishes,
Morten Lunde Nielsen

We would like to thank Reviewer 2 for his comments and support throughout the revisions of this paper.

Reviewer #3 (Remarks to the Author):

I am grateful for the authors' thoughtful consideration and thorough approach to revisions. I am happy that the authors have fully addressed my comments and concerns from the original review. I think the changes have improved the work's clarity, removed ambiguity and strengthened the overall message of the paper.

For the specific editor comments:

1] Please fully justify the definition and use of 'soft tissue' as a descriptor of the studied exoskeletons, in response to the criticisms raised by reviewer #1.

I agree with reviewer #1 that shrimp cuticle is technically not a "soft-tissue". However, I agree with the authors that as shrimp cuticle largely exhibits the same preservation patterns as soft-tissues the use of "soft-tissue" did indicate the inherent preservation potential of the cuticle. While it is just a terminology change and largely doesn't impact the message of the paper, I think the authors' response and related changes are appropriate and increase the precision of the work.

2] The revision should include a more extended presentation and discussion of the impact of the results on considerations of time averaging, including a more thorough referencing of previous work and explanation of how this current work demonstrates new findings.

I think the changes to the discussion, including expansion and revision of the text throughout have addressed these concerns. There is an increased consideration of the place of the work within a wider context, and relation to other works, along with more explicate description of the novelty of the presented work.

3] The specific requests concerning improvements in the description of methods (e.g., Reviewer 3 comments) should be met.

I think this response to this has been detailed and well thought through. I think the additional information has improved the repeatability of the protocols and clarified what exactly was done. Improvements to the taphonomic scoring descriptions greatly improve the readers' understanding and subsequent interpretation of the results

Overall I think the revisions have improved the message and the manuscript greatly, I have no hesitation in recommending this work for publication.

We would like to thank Reviewer 3 for their comments and support throughout the revisions of this paper.